# Caveolae-mediated Tie2 signaling contributes to CCM pathogenesis in a brain endothelial cell-specific *Pdcd10*-deficient mouse model

Huanjiao Jenny Zhou [1✉], Lingfeng Qin[1,4], Quan Jiang [1,4], Katie N. Murray[2,4], Haifeng Zhang[1,4], Busu Li[1], Qun Lin[1], Morven Graham [3], Xinran Liu[3], Jaime Grutzendler [2] & Wang Min [1✉]

Cerebral cavernous malformations (CCMs) are vascular abnormalities that primarily occur in adulthood and cause cerebral hemorrhage, stroke, and seizures. CCMs are thought to be initiated by endothelial cell (EC) loss of any one of the three *Ccm* genes: *CCM1 (KRIT1), CCM2 (OSM), or CCM3 (PDCD10)*. Here we report that mice with a brain EC-specific deletion of *Pdcd10* (*Pdcd10*BECKO) survive up to 6-12 months and develop bona fide CCM lesions in all regions of brain, allowing us to visualize the vascular dynamics of CCM lesions using transcranial two-photon microscopy. This approach reveals that CCMs initiate from protrusion at the level of capillary and post-capillary venules with gradual dissociation of pericytes. Microvascular beds in lesions are hyper-permeable, and these disorganized structures present endomucin-positive ECs and α-smooth muscle actin-positive pericytes. Caveolae in the endothelium of *Pdcd10*BECKO lesions are drastically increased, enhancing Tie2 signaling in Ccm3-deficient ECs. Moreover, genetic deletion of caveolin-1 or pharmacological blockade of Tie2 signaling effectively normalizes microvascular structure and barrier function with attenuated EC-pericyte disassociation and CCM lesion formation in *Pdcd10*BECKO mice. Our study establishes a chronic CCM model and uncovers a mechanism by which CCM3 mutation-induced caveolae-Tie2 signaling contributes to CCM pathogenesis.

[1] Interdepartmental Program in Vascular Biology and Therapeutics, Department of Pathology, Yale University School of Medicine, New Haven, CT, USA.
[2] Department of Neurobiology, Yale University School of Medicine, New Haven, CT, USA. [3] Department of Cell Biology, Yale University School of Medicine, New Haven, CT, USA. [4] These authors contributed equally: Lingfeng Qin, Quan Jiang, Katie N. Murray, Haifeng Zhang. ✉email: huanjiao.zhou@yale.edu; mike.wang388@gmail.com

Cerebral vascular malformations affect 0.1–4% of the general population and can be classified into four categories: arteriovenous malformations, venous malformations, capillary telangiectasia, and cerebral cavernous malformations (CCMs). In particular, CCMs are clusters of enlarged endothelial channels ('caverns') arranged back-to-back. These caverns form densely packed sinusoids containing a single layer of endothelial cells (EC) with little or no intervening brain parenchyma. CCMs lack smooth muscle cells/pericytes, sub-endothelial support, and an intact basal lamina; therefore, vessel walls are thin and prone to leakage[1–3]. CCMs are primarily found within the vasculature of the central nervous system (CNS, i.e., brain, spinal cord, and retina) where they increase the risk of stroke, seizures, and focal neurological deficits[3]. Currently, the only treatment for CCM is surgical resection.

CCMs are associated with loss-of-function mutations in one of the three Ccm genes: CCM1 (or Krev/Rap1 Interacting Trapped 1, KRIT1)[4], CCM2 (or malcavernin or osmosensing scaffold for mitogen-activated protein kinase kinase-3, OSM)[5], and CCM3 (or programmed cell death 10, PDCD10)[6]. Specifically, the primary defects underlying CCMs are Ccm gene deficiencies in vascular ECs[7,8] but mural cells also contribute to CCM pathogenesis[9]. Indeed, current mouse models inducing global EC deletions in any one of the three Ccm genes (Ccm1, Ccm2, or Ccm3) cause CCM lesions in mouse brains and retinas that resemble human lesions[10–15]. Importantly, these deletions induce CCM lesions in neonatal but not adult mice; however, the reason for this is unknown. Moreover, neonatal mice with global EC-specific Ccm gene deletions do not survive to adulthood. Therefore, a more reliable mouse model is needed to explore the underlying mechanisms and potential therapies for CCM.

Ccm genes encode three CCM proteins: CCM1, CCM2, and CCM3, respectively. These proteins are found in the same complex within the cell. Recent studies have suggested that the CCM proteins regulate common pathways, including RhoA-dependent EC stress fiber formation, TGF-β/Smad/BMP-mediated endothelial–mesenchymal transition (EndMT) signaling, and MEKK3-ERK5-KLF2/4-mediated matrix remodeling. Importantly, all of these pathways contribute to the onset and progression of CCMs[10–15]. EndMT, a process defined as the acquisition of mesenchymal- and stem-cell-like characteristics by the endothelium[16,17]. Dr. Dejana's lab first reported that endothelial-specific disruption of Ccm1 gene in mice induced the TGF-β and BMP-mediated EndMT, which contributed to the development of vascular malformations[14]. By contrast, Dr. Kahn's lab found no evidence of EndMT or increased SMAD signaling during early CCM formation[15]. Therefore, the role of EndMT in CCM pathogenesis remains controversial. Recently, studies using multi-color fluorescent reporter mice indicated that CCM lesions originate from a single clonal expansion of mesenchymal- and stem-cell-like Ccm-mutant ECs within CCM lesions, followed by the incorporation of wild-type ECs into the growing malformation[18,19]. Of note, CCM3 may also act separately from CCM1 and CCM2, as CCM3 mutations in humans often result in a more severe form of the disease[20]. Consistent with human studies, mice with CCM3 loss exhibit a more severe phenotype than those with loss of CCM1 or CCM2[10–15]. Earlier studies suggested that CCM3, but not CCM1 or CCM2, specifically interacts with the germinal center kinases (GCKs) STK24 and STK25[21]. Our recent work showed that by interacting with GCK in ECs, CCM3 suppresses exocytosis-mediated secretion of angiopoietin-2 (Angpt2). Angpts are endothelial growth factors that bind to the tyrosine kinase receptor Tie2, cooperatively regulating EC adherens junction-dependent vascular stability and blood vessel formation during angiogenesis[22]. Angpt2, secreted by ECs, is classically considered as an Angpt1 antagonist,

counteracting the stabilizing action of Angpt1 to increase EC permeability[23,24]. More and more studies have now indicated that Angpt2, by activating Tie2, is required for angiogenesis[25,26]. Our early work has demonstrated that CCM3 loss-augmented Angpt2 release disrupts the blood–brain barrier (BBB) and promotes CCM3 disease pathogenesis in the brain which can be blocked by Angpt2 neutralization antibody[27]. However, whether and how Angpt2 through its receptor Tie2 drives the CCM pathogenesis is unclear.

Why human CCM lesions are primarily confined to the brain vasculature and retina, despite ubiquitous expression of CCM proteins, remains unclear. The unique feature of the brain vasculature is the BBB formed by the brain neurovascular unit consisting of endothelial cells (EC), pericytes, astrocytes, microglias, and neurons[28]. Caveolae are flask-shaped invaginations of the plasma membrane with diverse function, including transcytosis and receptor signaling. The formation of caveolae vesicles requires the integral membrane protein caveolin-1 (Cav1) and cytosolic adaptor cavin proteins[29]. Despite that caveolae are particularly abundant in vascular endothelium, they are limited in highly restrictive microvascular ECs of brain and retina[29]. Interestingly, increased caveolae has been associated with increased BBB disruption[30,31].

Here, we show that an inducible Pdcd10 deletion using a brain EC (BEC)-specific Cre line (Pdcd10[BECKO]) promotes CCM lesions in the brain and retina without causing vascular defects in other tissues. Importantly, Pdcd10[BECKO] mice survive up to 6–12 months, allowing us to visualize vascular lesion formation by live imaging, to define the CCM pathogenesis, and to test therapeutics in adulthood. Using this newly established mouse CCM model, we observe CCM lesion is generated by initially bulging from capillary/post-capillary venule with a single dilated blinding end (cavern), and rapidly grows into disorganized venule beds with disrupted microvascular network. We detect dramatically increased caveolae vesicles in the brain microvascular ECs, and the increased caveolae augment Tie2 receptor signaling contributing to CCM lesion progression.

## Results

**Inducible BEC-specific Ccm3 deletion (Pdcd10[BECKO]) promotes CCM lesions in adult mice that resemble human disease.** Current CCM models have employed Ccm gene deletions with either a global EC- or a brain EC-specific promoter[32–34]. We previously created inducible Pdcd10 knockout mice (Pdcd10[ECKO]) with a global EC Cdh5-CreER[T2] activated upon tamoxifen feeding of pups at postnatal days (P) 1–3[27]. Pdcd10[ECKO] pups develop severe CCM lesions in the brain and retina by P10, but do not survive beyond P15. To identify the causality of Pdcd10[ECKO] perinatal lethality, we examined peripheral organ structures and morphologies. This revealed that Pdcd10[ECKO] pups had severe vascular damage in several other tissues, particularly the spleen. Like the cerebellums and retinas, spleens contained enlarged and irregular blood vessels as visualized by fresh tissue imaging (Supplementary Fig. 1a-c), hematoxylin and eosin (H&E) staining (Supplementary Fig. 1d-f), and CD31 immunostaining (Supplementary Fig. 1g-h). Moreover, splenectomy at P6 prolonged Pdcd10[ECKO] mouse survival to 1 month, and these mice exhibited seizures with severe CCM lesions and brain hemorrhaging (Supplementary Fig. 1i-j). Thus, splenic damage is the primary cause of neonatal lethality in Pdcd10[ECKO] mice.

To circumvent perinatal lethality from splenic hemorrhage, we next created a tamoxifen-inducible BEC-specific Cre line driven by the Mfsd2a promoter (Mfsd2aCreER[T2])[35]. We then generated BEC-specific Ccm3 knockout mice (Pdcd10[BECKO]) by crossing

Ccm3[lox/lox] with Mfsd2aCreER[T2] mice. We next evaluated the BEC deletion efficiency and CCM lesion incidence by feeding mT/mG: Mfsd2aCreER[T2]; Pdcd10[fl/fl] pups with tamoxifen from P1 to P3. Tamoxifen-induced mG (membrane-bound EGFP) expression occurred specifically in the brain and retinal vasculatures (Supplementary Fig. 2a-b). Unlike Cdh5-CreER[T2] mice[27], Mfsd2aCreER[T2] mice did not exhibit mG expression in other tissues such as the heart, liver, kidney, and spleen (Supplementary Fig. 2c). Moreover, EGFP expression in these mice co-localized with the EC marker CD31 in the brain vasculature (Supplementary Fig. 2d). We further isolated mouse brain microvascular ECs (mBMVECs) and mouse brain microvascular pericytes (mBMVPCs) as we described previously[9]. Immunostaining confirmed that isolated mBMVECs expressed CD31 but not PDGFRb; conversely, mBMVPCs expressed PDGFRb but not CD31 (Supplementary Fig. 2e). Ccm3 deletion was specifically detected in isolated mBMVECs, but not in mBMVPCs, as verified by qRT-PCR and western blotting (Supplementary Fig. 2f, g).

We then compared the location, number, and size of CCM lesions between wild-type (WT), Pdcd10[BECKO], and Pdcd10[ECKO] pups. H&E staining revealed that by P10, Pdcd10[BECKO] mice rapidly developed CCM lesions with dilated capillaries in both cerebrum and cerebellum comparable with Pdcd10[ECKO] pups (Fig. 1a with quantifications in 1b, c). However, while all Pdcd10[ECKO] pups died by P15, >90% of Pdcd10[BECKO] mice survived until they were at least 2 months old. Most Pdcd10[BECKO] mice died of brain hemorrhage, with severe cerebral edema and attenuated body-weight gain, between 6 and 9 months old (Fig. 1d, e). At these timepoints, Pdcd10[BECKO] mice had drastically enlarged CCM lesions (Fig. 1f–i with quantifications in 1j, k). Furthermore, these late-stage CCM lesions exhibited hemosiderin deposits, collagen deposition, and increased EC proliferation (Fig. 1g, i). Staining with venous endothelium and capillary marker endomucin indicated that venous malformations were detected at the periphery of the retinal vascular plexus at both P15 and 2-month ages (Supplementary Fig. 3a-b). In contrast to Pdcd10[ECKO] mice, Pdcd10[BECKO] mice did not exhibit lesions in the spleen or other tissues at various ages (Supplementary Fig. 4).

**CCM lesions bulged from the capillary/post-capillary venules visualized by live confocal microscopy in adult Pdcd10[BECKO] mice.** All previous CCM mouse models with whole-body EC-specific deletion induce CCM lesions only in the cerebellum as the mice died around P15[10–15]. It is still unknown why CCM lesions progress more rapidly in cerebellum compared to cerebrum. This could be related to the unique structure and hemodynamics of cerebellum[13,36] or spatial expression of the Angpt-Tie2 signaling as reported previously[37,38]. CCM lesions in Pdcd10[BECKO] mice were not only detected in the cerebellum but also in the cerebrum, allowing us to perform experiments visualizing the vascular dynamics of CCM lesions using transcranial two-photon microscopy. Our data indicated that Pdcd10[BECKO] mice are a viable model for visualizing CCM lesions in adult mice. We visualized vascular abnormalities in Pdcd10[BECKO] mice using confocal microscopy after injection of Evans blue dye (EBD). The initial tamoxifen feeding protocol (P1 deletion model as described in Fig. 1) caused massive lesions with severe hemorrhaging that made imaging the vasculature difficult. Therefore, we modified the protocol by injecting tamoxifen at different times (P1, P5, P10, and P21; Fig. 2a) and obtained adult mice (>2-month old) with severe (feeding at P1), moderate (P5), mild (P10), or no lesions (P21), respectively (Fig. 2b, c). In Pdcd10[BECKO] mice, mild CCM lesions appeared as protruding

blood-filled (and EBD-perfused) balloons (caverns) that contained turbulent flow (Supplementary movies 1–4). By contrast, moderate and severe lesions had increased cavern numbers and sizes, leading to disrupted microvascular network (as indicated by asterisks in Fig. 2c).

We next used confocal microscopy to visualize the CCM microvasculature. To this end, Pdcd10[BECKO] mice were crossed under the background of mT/mG reporter mice, in which ECs expressed mG (membrane-bound EGFP). Of note, mG expressed mainly in the veins and capillaries but not in the arteries, consistent with the Mfsd2a gene expression pattern[30,31,39]. These mice were injected with EBD and confocal microscopy was used to visualize CCM lesions and perfusion. In Pdcd10[BECKO]:mT/mG mice, we observed that mild lesions (P10 deletion) bulged primarily from the capillary/post-capillary venules with dilated blinding ends (caverns; indicated by arrows). Moderate lesions (P5 deletion) were multi-cavernous and arranged back-to-back to form densely packed sinusoids throughout the venous beds and surrounding capillaries (indicated by asterisks) (Fig. 2c with quantification in 2e). Thus, moderate lesions had drastically larger total vascular and EBD-perfused areas than mild ones (Fig. 2d with quantification in 2f, g). Of note, the imaging of EBD indicated that most caverns were perfused, but some of the newly formed sprouts were non-perfused (Fig. 2d with quantifications in 2h). The moderate lesions In Pdcd10[BECKO]:mT/mG mice also had enlarged veins with a vein diameter approximately three times higher than that of control Mfsd2aCreER[T2]:mT/mG mice (Fig. 2i with quantifications in 2j).

**Pericytes dissociated from microvessels with increased α-SMA expression during CCM lesion formation.** Pericytes are an integral component of the neurovascular unit that play fundamental roles in the development and maintenance of the brain vascular network[28,40]. We specifically labeled brain pericytes using the fluorescent Nissl dye NeuroTrace 500/525 and then performed high-resolution optical imaging in live animals[41]. Topical application of this dye through a cranial window labeled a distinct population of cells that line cerebral blood vessels at cortical depths up to 400 μm. Dye labeling was very bright and concentrated both in cell soma and throughout the processes where it displayed a punctate pattern. Co-labeling of perfused vessels with Texas Red IV dye showed that in WT mice, NeuroTrace-labeled cells lined the smallest cerebral vessels and exhibited the morphology of capillary pericytes. Specifically, these cells had multiple slender processes extending longitudinally and spanning several vessel branches. By comparison, pericytes in Pdcd10[BECKO] mice had shorter processes (Fig. 2k; pericyte cell body by arrowheads and processes by brackets); pericytes dissociated from the vessel are indicated by arrows). In Pdcd10[BECKO] mice, pericyte-free caverns were dramatically increased (Fig. 2l). Pericyte density (pericytes/vascular area) in mild lesions was not altered but was dramatically reduced in moderate lesions (Fig. 2l with quantifications in 2m).

We also examined pericyte coverage and EC junctions using co-immunostaining with the EC marker CD31, pericyte marker NG2, and junctional proteins VE-cadherin and claudin-5. Consistent with the two-photon microscope imaging showing shorted processes in pericytes, moderate CCM lesions (P5 deletion and P60 analyses) had more looser association of pericyte with EC and had a much lower rate of pericytes coverage relative to normal vessels (Fig. 3a with quantification in 3b). Immunostaining also revealed that moderate lesions in Pdcd10[BECKO] mice demonstrated a single layer of dilated endothelium with significantly reduced endothelial adherens and tight junction coverage (Fig. 3a with quantification in 3b).

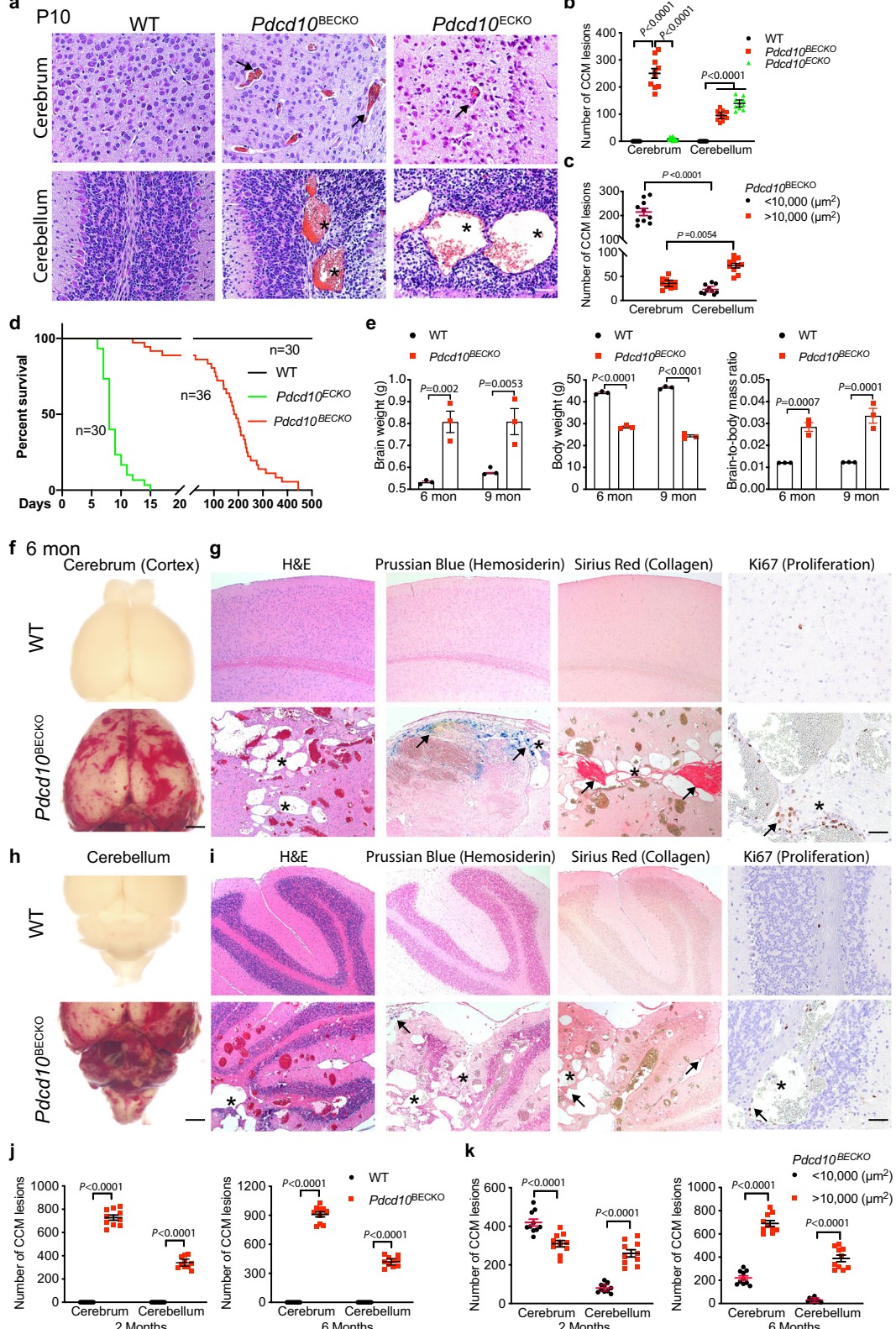

It has been previously reported that EC in the CCM lesion undergoes EndMT with gain of α-SMA expression[14]. While this is disputed subsequently by another study[15], detailed analyses were not provided. To this end, we employed stimulated emission depletion (STED) microscopy, a super-resolution microscopy technique, to visualized if ECs expressed α-SMA. We chose early CCM lesions in the P1 deletion model to examine initial cellular

changes (Fig. 3c for the protocol). CD31+ EC and NG2+ pericyte were tightly associated with each other in normal brain microvessels without detectable α-SMA. CD31+ ECs and NG2+ pericytes were clearly separated into two layers in the CCM lesion. Moreover, multiple α-SMA+ cells were co-localized with NG2+ pericyte but not with CD31+ EC within a lesion (Fig. 3d). Similar results were obtained in the retinas: α-SMA in WT retina

**Fig. 1 Pdcd10[BECKO] mice develop CCM lesions but survive to adulthood. a–c** CCM lesion quantification in WT, Pdcd10[BECKO], and Pdcd10[ECKO] mice at P10. Representative images of H&E staining and lesion quantifications in the cerebrum (arrows) and cerebellum (asterisks) are shown. The numbers of total lesions and lesions with different sizes were quantified as # of lesions per 10 coronal sections, which were 100 μm apart. n = 10 mice per group. **d** Cumulative survival curve for WT (Pdcd10[fl/fl]), Pdcd10[BECKO] (Mfsd2aCreER[T2]; Pdcd10[fl/fl]), and Pdcd10[ECKO] mice (Cdh5-CreER[T2]; Pdcd10[fl/fl]). N for each group is indicated. P < 0.0001 (Log-rank test). **e** Brain/body-weight ratios. Brain weight, body weight, and the brain/body-weight ratios are presented for WT and Pdcd10[BECKO] mice at 6–9 months old. n = 3 mice per group. **f–k** CCM lesion characterization and quantification in WT and Pdcd10[BECKO] at 2–6 months. Representative images of fresh tissues, H&E staining (lesions indicated by asterisks), iron deposits (blue patches identified by Prussian blue), collagen deposition (red patches identified by Sirius Red), and EC proliferation (Ki67[+]) in the cerebrum (**f, g**) and cerebellum (**h, i**). Arrows indicate positive staining. The numbers of total lesions and lesions with different sizes were quantified as # of lesions per 10 coronal sections, which were 200 μm apart. n = 10 mice per group. Data are means ± SEM. P values are indicated, two-way ANOVA followed by Sidak's multiple comparisons test (**b, c, e, j, k**). Scale bars: 25 μm (**a**); 100 μm (**g, i**); 1 mm (**f, h**). Source data are provided as a Source data file.

was detected only in artery and weakly in vein, but not in the microvascular plexus. However, α-SMA was highly upregulated in the CCM lesions of Pdcd10[BECKO] retinas (Fig. 3e). Furthermore, high-power confocal imaging indicated that α-SMA in the CCM lesion of Pdcd10[BECKO] mice was expressed in NG2[+] pericytes but not in CD31[+] ECs (Fig. 3f). Taken together, these data indicate that pericytes may gain α-SMA expression during CCM lesion formation.

**CCM lesions exhibit increased caveolae vesicles and the Cav1-Tie2 signaling.** Mouse brain lesions (from the P1 deletion/P15 analyses model) were further examined at the ultrastructural level using transmission electron microscopy (EM). The normal brain capillaries formed tight junctions (TJ) between EC-EC. However, 65% EC-EC tight junctions (65 out of 100 junctions examined) were weakened with a discontinuous pattern in Pdcd10[BECKO] mice (Supplementary Fig. 5a-b). Detailed analyses on EM images indicated that the intracellular vesicles increased in Pdcd10[BECKO] brain ECs exhibited typical caveolae structure. Consistent with previous findings that the mouse brain microvasculature has limited caveolae[29], very few caveolae were detected in WT brain venous/capillary EC. However, the number of caveolae was dramatically increased in Pdcd10[BECKO] brain lesion ECs. Moreover, caveolae were detected in intracellular and junctional regions on both luminal and abluminal sides with formation of rosettes (clusters) in Pdcd10[BECKO] brain which were further confirmed by electron tomography which visualized 3D structure of caveolae in cells (Fig. 4a, b with quantification in 4c, d; Supplementary Fig. 5c-d for EM tomography; Supplementary Movies 5-6). These data suggest an active caveolae-mediated trafficking in Pdcd10[BECKO] lesion EC[29]. Consistently, increased caveolae structural protein Cav1 was detected in Pdcd10[BECKO] ECs compared to WT as determined by immunostaining. Interestingly, we detected increased total and phosphorylation of Tie2 (p-Tie2), the Angpt2 receptor in CCM lesions where it was co-localized with Cav1 (Fig. 4e, g). One report indicated that Cav1 regulated Tie2 nuclear translocation in tumor cells[42]. We employed STED to visualize Tie2 localization and we found no or a minimal nuclear translocation of total Tie2 or p-Tie2 in ECs of WT and Pdcd10[BECKO] brain tissues (Fig. 4h, i).

We further determined the Cav1-Tie2 signaling by western blotting in mouse brain tissues. Since we observed CCM lesion progressed quickly in cerebellum compared to cerebrum, we compared if the Cav1-Tie2 signaling was different between cerebellum and cerebrum. We found that CCM3 protein was expressed at a lower level in cerebellum than cerebrum (Supplementary Fig. 6a) (Note: Other brain cells also expressed CCM3 so overall CCM3 levels were not reduced in Pdcd10[BECKO] mice unless we isolated brain ECs as presented in Supplementary Fig. 2). Interestingly, Angpt2, Cav1, Tie2, p-Tie2, and downstream phosphor-Akt (p-Akt) were highly upregulated in cerebellum compared to cerebrum whereas other known

CCM3-regulated signaling such as phosphor-MLC (indicative of the RhoA signaling) was increased more profoundly in cerebrum than cerebellum (Supplementary Fig. 6a). These results further support a critical role of the Cav1-Tie2 signaling in CCM lesion progression. Like all other mouse models, deletion of Ccm3 gene in adult mice (at P21) did not generate CCM lesions. It has been suggested that ongoing angiogenesis and brain vascular remodeling are required for CCM lesion formation[13]. Interestingly, signaling proteins (Angpt2, Tie2, p-Akt, p-MLC, and VEGFR2 in particular) declined dramatically by P21 in both cerebrum and cerebellum of WT brain (Supplementary Fig. 6a). It has been reported that VEGF signaling enhances the CCM lesion burden in the Ccm1-deficient mice; inhibition of VEGFR2 using a specific inhibitor SU5416 significantly decreased the number of lesions formed and slightly lowered the average lesion size in Ccm1-knockout mice[43]. Taken together, these data are consistent with the notion that VEGFR2 expression and ongoing angiogenesis are required for CCM lesion formation.

In parallel, we also examined the vascular structural proteins such as EC junctional proteins (Cldn5, ZO-1, VE-cadherin, and N-cadherin), and SMC/pericyte marker proteins (PGDFRb and SMA). Of note, the overall levels of pericyte marker PDGFRb and EC junctional proteins were lower in cerebellum compared to cerebrum, reciprocal to the Cav1-Angpt2 levels. Consistent with the immunostaining results, pericyte marker PDGFRb was slightly reduced (in cerebrum) whereas α-SMA was increased in Pdcd10[BECKO] mice. While junctional proteins were clearly reduced in CCM lesions by immunostaining, their total protein levels (e.g., Cldn5, ZO-1, and VE-cadherin) detected by Western blot were slightly increased (Supplementary Fig. 6b). This could be related to increased total vascular areas in Pdcd10[BECKO] mice as indicated by a weak increased CD31 levels. Alternatively, regulation at a post-translational modification and intracellular endocytosis reduced the junctional localization of these proteins[44].

To determine if Cav1-Tie2 upregulation has clinical relevance, we examined expression of Cav1-Tie2 in human CCM3 lesions. We obtained paraffin blocks of 8 cases of human CCM3 lesions[27]. Cav1 staining was absent in normal brain areas, but was significantly upregulated in CCM lesions where it was co-localized with CD31. Similar patterns of upregulation were observed for total and p-Tie2 in CCM lesions where they both were co-localized with Cav1 in endothelium (Fig. 4j, k). These data indicated that Cav1-Tie2 signaling was increased in both mouse and human CCM lesions.

**Cav1 mediates CCM3 loss-augmented Tie2 signaling and lumen enlargement in in vitro models.** We further determined the functional significance of the co-localization of Cav1 with Tie2 in CCM3-knockdown (KD) ECs. As previous reported[29], Cav1 was restricted to the membrane-proximal region in control EC where Tie2 was localized with Cav1 (arrow) as determined by

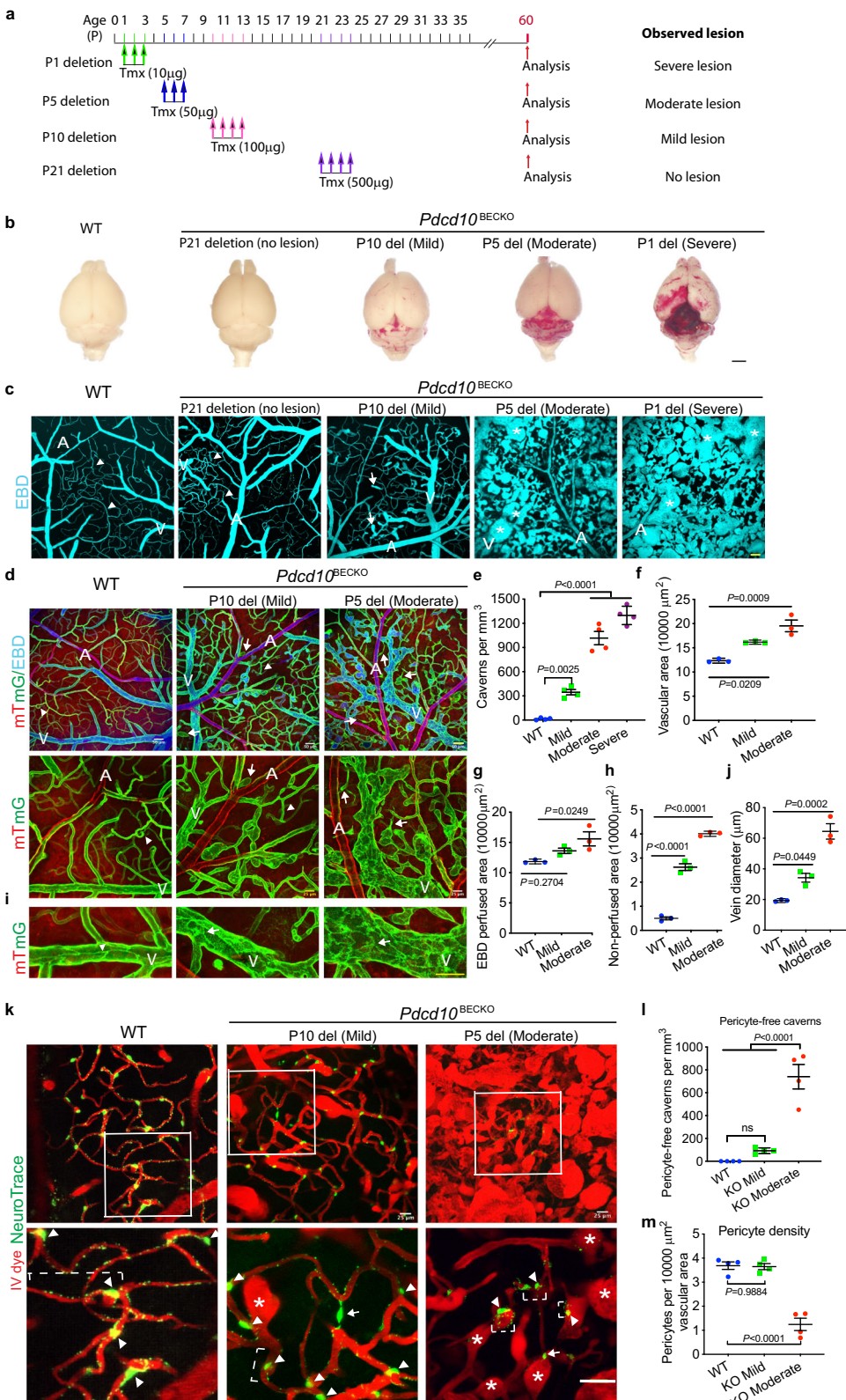

immunofluorescence staining. Silencing of CCM3 in HBMVEC significantly increased Cav1 and Tie2 co-localization both in the membrane-proximal region (arrow) and in intracellular vesicles (arrowhead) (Fig. 5a for high-power images; Supplementary Fig. 7a for lower power images). Cav1 siRNA diminished the surface Tie2 whereas Tie2 siRNA removed both surface and intracellular Tie2 staining (Fig. 5a). By contrast, increased Tie2 in

CCM3-KD EC was barely detected in clathrin-positive endo-somes (Supplementary Fig. 7b). Western blot showed that total- and p-Tie2 were increased in CCM3-KD EC. The high levels of the p-Tie2 is a result of high levels of the total Tie2 expression, and the ratios of p-Tie/Tie2 were similar between WT and Ccm3-KD ECs. Furthermore, co-silencing of Cav1 diminished CCM3 loss-augmented Tie2 expression in ECs (Fig. 5b, c). We have

**Fig. 2 CCM lesions bulged from the capillary/post-capillary venules with blinding ends. a** Tamoxifen feeding protocols. WT (*Pdcd10*fl/fl) and *Pdcd10*BECKO (*Mfsd2a*CreERT2; *Pdcd10*fl/fl) pups were fed tamoxifen at different times to induce Ccm3 deletion at P1, P5, P10, and P21, respectively. Lesion phenotypes were analyzed at P60. **b** CCM lesions visualized by whole-brain imaging. **c** Live confocal microscopy of vascular malformations after injection of Evans blue dye (EBD in cyan color). Arrowheads and arrows indicate normal capillary and cavern, respectively. Asterisks indicate large lesions. A; artery/arterioles; V: vein/venules. **d–j** Characterization of vascular structure and perfusion. CCM lesions and perfusion in 2-month-old WT (*Mfsd2a*CreERT2:mT/mG) and *Pdcd10*BECKO:mT/mG mice were visualized by confocal microscopy after injection with EBD. Blue: EBD; green: mG, primarily expressed in veins/venules/ capillaries/arterioles; red: mT, all blood vessels. **d** Representative images of normal vessels in WT mice and mild to moderate lesions in *Pdcd10*BECKO mice are shown, with higher magnifications without EBD color at the lower panels. Arrowheads indicate normal capillaries/post-capillary venules in WT and *Pdcd10*BECKO mice, whereas arrows indicate non-perfused newly formed sprouts in *Pdcd10*BECKO mice. **e** The number of lesions (caverns-like structure/ vascular area) are quantified. n = 4. **f, g** Total vascular area (mG) and EBD-perfused area are quantified. **h** Non-perfused (mG+/EBD−) area is quantified. **i** Comparison of vein junctions in WT (arrowhead) and *Pdcd10*BECKO (arrow). Quantifications of vein diameters (**j**). n = 3. **k–m** Two-month-old WT and *Pdcd10*BECKO mice were perfused with a Texas Red intravenous (IV) dye and NeuroTrace (pericytes) followed by confocal microscopy. **k** Representative images of normal vessels in WT mice and mild to moderate lesions in *Pdcd10*BECKO mice are shown. **l, m** Pericyte-free caverns and pericyte density (number/vascular area) were quantified. n = 4. Arrowheads indicate normal pericyte processes in WT mice, whereas arrows indicate pericytes with shortened processes in *Pdcd10*BECKO mice. Asterisks indicate CCM lesions in *Pdcd10*BECKO mice. Data are means ± SEM. *P* values are indicated, one-way ANOVA followed by Sidak's multiple comparisons test (**e**); one-way ANOVA followed by Tukey's multiple comparisons test (**f, g, h, j, l, m**). Scale bar: 2 mm (**b**); 50 μm (**c**); 2 μm (**d, k**). Source data are provided as a Source data file.

previously shown that silencing of CCM3 caused a significant increase in exosome activity and associated secretion of Tie2 ligand Angpt2 from ECs, and Angpt2 release augmented by CCM3 silencing could be suppressed by co-silencing Unc13b[27]. However, we found that co-silencing of Cav1 had no effects on Angpt2 release (Supplementary Fig. 7c). Conversely, co-silencing of Unc13b did not attenuate Tie2 expression in CCM3-KD ECs (Supplementary Fig. 7d-f). These data suggest that CCM3 regulates Cav1-Tie2 independently from Unc13b-mediated exocytosis and Angpt2 secretion.

Since CCM3 did not alter Tie2 mRNA (Supplementary Fig. 7d), we examined if Cav1 regulates Tie2 protein stability and cellular trafficking[29]. Biotinylation assays[45] showed that surface and endocytic Tie2 proteins were increased in CCM3-KD ECs, but both were attenuated by co-silencing Cav1 (Fig. 5d, e), consistent with the immunostaining results. We then examined the half-life of Tie2 in ECs in the presence of a protein synthesis inhibitor cycloheximide (CHX), and results showed that Cav1 deletion reduced the half-life of Tie2 (Fig. 5f, g). Notably, Tie2 was relative stable ($t_{1/2}$ ~ 3 h, similar to VEGFR3) compared to VEGFR2 ($t_{1/2}$ < 30 min) as we previously reported[46]. Of note, expression of Abl kinase, a kinase involved in Tie2 signaling and vascular malformation, was not altered by CCM3- or Cav1 silencing in ECs. Consistent with the Tie2-clathrin localization, decreased Tie2 localization in the Lamp1+ lysosome was detected in Ccm3-KD ECs. However, Cav1 deletion augmented localization of Tie2 with the lysosomal marker Lamp1 (Fig. 5h). Taken together, these data suggest that Cav1 increases Tie2 protein in CCM3-deficient ECs by enhancing Tie2 intracellular trafficking and stabilization.

To determine if Cav1 mediates CCM3 loss-induced EC junction disruption as observed in the *Pdcd10*BECKO brains, we examined the co-silencing effect of Cav1 on CCM3 loss-induced disruptions of EC junctions and EC integrity. CCM3 knockdown drastically disrupted both adherens and tight junctions as detected by immunostaining for VE-cadherin and ZO-1, respectively. However, co-silencing Cav1 significantly restored normal adherens junctions and tight junctions (Fig. 5i with quantifications in 5j). Co-silencing Cav1 also restored barrier function in the CCM3-ablated ECs as assessed for transendothelial electric resistance (TEER) by electrical cell-substrate impedance sensing (Fig. 5k).

To determine if Cav1 mediates CCM3 loss-induced vessel lumen enlargement as observed in the *Pdcd10*BECKO brains, we finally analyzed EC sprouting and lumen formation using an optimized in vitro EC sprouting and tube formation model[27]. To this end, EGFP-expressing and/or siRNAs transfected HBMVECs

were seeded onto a confluent layer of fibroblasts, and lumen formation occurred by multicellular ECs at the sites of lateral EC-EC contacts and peaked between day 7 and 14 of co-culture as we described previously[27]. EC sprouts and lumens visualized by VE-cadherin and collagen-IV staining were greatly enhanced by CCM3 deletion (Fig. 5l). Quantitative analyses indicated that the number of branch points, mean lumen diameters and lumenized (collagen-IV-covered) areas[47] in the CCM3-knockdown group were significantly increased. Increased EC sprouting and enlarged lumen formation induced by CCM3 knockdown were blunted by either co-repression of Cav1 with its siRNA or by Tie2 inhibitor Rebastinib (Fig. 5l with quantifications in 5m). These data suggest that increased Cav1 in CCM3-deficient EC is critical for Tie2 signaling and vessel lumen enlargement in the in vitro models.

**Cav1 deficiency reduces CCM lesions and normalizes EC-pericyte associations in *Pdcd10*BECKO mice.** *Cav1-deficient* (*Cav1*−/− or Cav1-KO) mice are viable and fertile, and abnormal brain vascular phenotypes have not been reported although *Cav1*-KO mice exhibited pulmonary fibrosis, hypertension, or cardiac hypertrophy as well as retinal vascular permeability[29]. To test if *Cav1* deficiency could rescue CCM phenotypes in the mouse CCM models, *Pdcd10*BECKO:*Cav1*−/− (DKO) mice were obtained by mating with *Cav1*−/− mice with Mfsd2a-CreERT2: *Pdcd10*fl/fl mice, followed by feeding pups with tamoxifen at P1 to P3 for induction of severe lesions. Brain tissues were harvested at P15 when full bloom of the lesion phenotypes in *Pdcd10*BECKO could be detected. No obvious lesions or very limited vessel dilation was found in the *Cav1*-KO brain or retina. Moreover, Cav1 deletion in *Pdcd10*BECKO mice (DKO) significantly reduced the number of lesions and capillary dilation of CCM lesions induced by *Ccm3* deletion as visualized by whole-brain imaging and H&E staining in the cerebellar sections from the DKO mice compared to *Pdcd10*BECKO mice (Fig. 6a–c; Supplementary Fig. 8 for retina). We then examined caveolae and Cav1-Tie2 signaling in these mice. Consistently, caveolae were significantly increased in *Pdcd10*BECKO brain ECs compared to WT mice. However, caveolae were absent in both Cav1-KO and the DKO mice (Fig. 6d, e). Tie2 and p-Tie2 staining surrounding dilated vessels were increased in the *Pdcd10*BECKO mice as compared to the WT mice or *Cav1*-KO mice. p-Tie2-positive ECs increased in the diseased brains of the *Pdcd10*BECKO was much more subdued in the DKO mice (Fig. 6f, g). The NG2+ pericyte coverage of CD31+ microvessels was enhanced from 22% in *Pdcd10*BECKO mice to over 60% in the DKO mice (Fig. 6f, h).

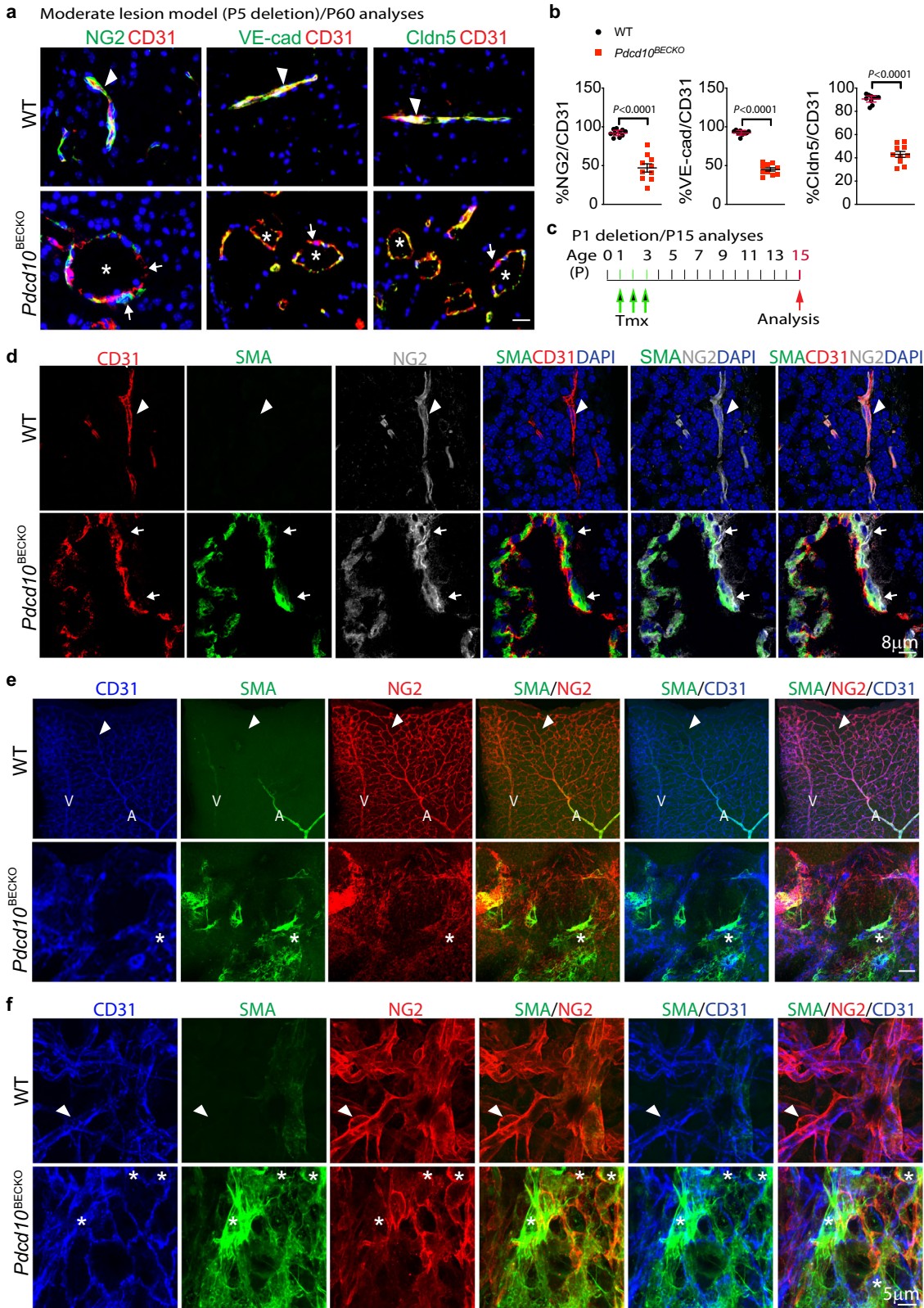

To further evaluate the rescue effect of *Cav1* deletion on CCM lesion formation and progression, we employed two-photon microscope imaging for the dynamic association of EC-pericyte and vascular structures. To this end, Ccm3 in mice was deleted at P5 followed by imaging of CCM lesions at age of 2-month old upon co-injection of NeuroTrace 500/525 and Texas Red IV dye (Note: The moderate but not the severe CCM model was suitable

for two-photon microscopy imaging). Cav1 deficiency dramatically reduced lesion sizes and numbers even at 2-month age as visualized by whole-brain imaging (Fig. 6i). Texas Red IV dye-perfused capillaries and post-capillary venules exhibited dilated blinding ends (caverns) with few pericytes in *Pdcd10*BECKO mice compared to WT and Cav1-KO mice. However, pericyte density was nearly normalized with much reduced number of pericyte-

**Fig. 3 Pericytes dissociated from microvessels with increased α-SMA expression during CCM lesion formation. a, b** P60 cerebrum sections from WT and the moderate CCM model of $Pdcd10^{BECKO}$ mice were stained for CD31 with NG2, VE-cadherin (VE-cad), or claudin-5 (Cldn5). Representative images of normal EC junction/EC-pericyte association (arrowheads) and disrupted EC junction/EC-pericyte association (arrows) within CCM lesions (asterisks) are shown (**a**). % of NG2, VE-cadherin, and claudin-5 coverage on CD31 vessels, i.e., % green (NG2, VE-cad, or Cldn5)-conjugated area/total red (CD31) areas was quantified by Image J (**b**), $n = 10$. $P < 0.0001$ (unpaired two-sided Student's $t$-test). Data are presented as mean ± SEM. **c–f** Pericyte phenotype characterization in early CCM lesions. **c** An early lesion model. $Pdcd10^{BECKO}$ pups were fed with tamoxifen at P1 to P3 and tissues were harvested at P15. **d** Brain sections from WT and $Pdcd10^{BECKO}$ mice were co-stained with CD31, α-SMA, and NG2. Images were captured under SP8 STED microscopy. Representative images for WT normal vessels and CCM lesions in $Pdcd10^{BECKO}$ mice are shown. $n = 6$. α-SMA was undetectable in EC or pericyte of WT vessels (arrowhead). α-SMA$^+$ cells were co-localized with NG2$^+$ pericyte but not with CD31$^+$ EC within a lesion (arrows). **e, f** Retinas from WT and $Pdcd10^{BECKO}$ pups at P15 were whole-mount stained with CD31, α-SMA, and NG2. Images were captured under confocal microscopy. Representative low power (**e**) and high-power (**f**) images for WT and $Pdcd10^{BECKO}$ mice are shown. $n = 6$. α-SMA was expressed in artery and weakly in vein, but undetectable in WT microvessels (arrowhead). α-SMA was dramatically increased in CCM lesions of $Pdcd10^{BECKO}$ retina (asterisks). Scale bar: 25 μm (**a**); 8 μm (**d**); 100 μm (**e**); 5 μm (**f**). Source data are provided as a Source data file.

---

free caverns in the DKO mice (Fig. 6i with quantifications in 6j–l). These results strongly suggest that $Cav1$ deficiency suppresses the Tie2 signaling in ECs and attenuates the vascular malformation caused by $Pdcd10$ inactivation.

**Tie2 inhibition rescued EC-pericyte associations with a therapeutic effect on CCM lesion progression.** Angpt2, a protein secreted by ECs, is classically considered to be an Angpt1 antagonist that counteracts the stabilizing action of Angpt1. Thus, Angpt2 increases EC permeability[23,24]. Previously, we reported that global EC deletion of $CCM3$ increases Angpt2 and, unexpectedly, p-Tie2 in CCM3 lesions[27]. Similarly, the $Pdcd10^{BECKO}$ mice had significantly upregulated levels of both Angpt2 and pTie2 (Supplementary Figs. 6 and 9a–d). It has been reported that endothelial Tie1 is essential for the agonist activity of autocrine Angpt2 by directly interacting with Tie2 to enhance Angpt2-induced p-Tie2[48–51] and Angpt2-induced the enlargement of post-capillary venules and capillaries in the trachea[48,49]. These studies prompted us to examine Tie1 expression in CCM lesions and CCM3-KO ECs. We observed highly upregulated Tie1 expression in CCM lesions (Supplementary Fig. 9a with quantifications in 9c). Moreover, Tie1 was upregulated in cultured ECs by CCM3 deletion and co-silencing Tie1 attenuated p-Tie2 (but not total Tie2) in CCM3-KO ECs (Supplementary Fig. 9e with quantifications in 9f). These data suggest that increased Tie1 in CCM3-KO EC may attribute to Angpt2-induced Tie2 activation.

Importantly, Angpt2 neutralization antibody treatment significantly reduced CCM lesion progression in the new $Pdcd10^{BECKO}$ mouse model. Moreover, we observed a long-term effect of Angpt2 neutralization antibody CCM lesion progression. Specifically, the antibody was injected intraperitoneally into the CCM $Pdcd10^{BECKO}$ P1-deletion model daily from P4-P14 followed by weekly from P15-P60, and CCM lesion was examined at P60 (Supplementary Fig. 9g). Angpt2-neutralizing antibody delivered with this regimen did not affect normal vessels in WT mice. However, whole-mount and H&E staining demonstrated that the antibody abrogated CCM lesion formation in the 2-month-old $Pdcd10^{BECKO}$ mice (Supplementary Fig. 9h-j), suggesting a long-term effect of Angpt2-neutralizing antibody on CCM lesion progression.

To determine if inhibition of Tie2 could be used as therapeutics for CCM, we tested the effects of Tie2 inhibitor Rebastinib (DCC-2036), a phase 1b/phase 2 clinical trial drug, and selective Tie2 kinase inhibitor, in the Ccm3 mouse models. To this end, we subcutaneously injected Rebastinib into WT and $Pdcd10^{BECKO}$ mice after the time of Ccm3 deletion. We first test the P1-deletion severe CCM model (Ccm3 deletion at P1-P3) followed by Rebastinib on P4-P14 (Fig. 7a for the protocol). Brain and retina tissues were harvested at P15, and slightly decreased angiogenesis on brain vascular development or retinal angiogenesis were

observed in Rebastinib-treated WT. However, Rebastinib almost completely diminished the CCM lesions induced by $Ccm3$ deletion as visualized by whole-brain imaging and H&E staining compared to vehicle treatment in $Pdcd10^{BECKO}$ mice (Fig. 7b with quantification in 7c). Rebastinib diminished p-Tie2 staining surrounding dilated vessels in both WT and $Pdcd10^{BECKO}$ mice (Fig. 7d with quantification in 7e). The NG2$^+$ pericyte coverage of CD31$^+$ microvessels in $Pdcd10^{BECKO}$ mice was normalized by Rebastinib treatment (from 22% in $Pdcd10^{BECKO}$ mice to over 86% as WT mice) (Fig. 7d with quantification in 7e). Rebastinib had similar effects on p-Tie2 and NG2$^+$ pericyte coverage in the retinas of $Pdcd10^{BECKO}$ mice (Supplementary Fig. 10a-b). Moreover, α-SMA expression in brain pericytes was diminished upon Tie2 inhibition (Supplementary Fig. 10b). We further examined if Rebastinib had a long-term effect on CCM progression. To this end, Rebastinib was injected subcutaneously into the CCM $Pdcd10^{BECKO}$ P1-deletion model daily from P4 to P14 followed by weekly from P15 to P60, and CCM lesion was examined at P60 (Fig. 7f). Very fewer CCM lesions were detected in Rebastinib-treated $Pdcd10^{BECKO}$ brain compared to vehicle group even at age of 2 months (Fig. 7g, h). These data suggest that Tie2 inhibitor had long-term inhibitory effects on CCM lesion progression.

We further evaluated the therapeutic effects of Tie2 inhibitor on the dynamic association of EC-pericyte and vascular structures during CCM lesion formation and progression by two-photon microscope imaging. To this end, Rebastinib was applied to the moderate CCM model (P5 deletion) from P8 to P60 (Fig. 7i for the Protocol). EC-pericyte association was imaged with Neuro-Trace 500/525 and Texas Red IV dye. Rebastinib did not affect pericyte number and density in WT mice. Vehicle-treated $Pdcd10^{BECKO}$ mice exhibited dilated blinding ends (caverns) with few pericytes at capillaries and post-capillary venules. However, Tie2 inhibitor completely normalized the pericyte-EC association as determined by increased pericyte density and the diminished pericyte-free caverns in the Rebastinib-treated mice (Fig. 7j with quantifications in Fig. 7k). We also evaluated the effects of Rebastinib on the vascular structure using confocal microscopy in $Mfsd2aCreER^{T2}$:mT/mG and $Pdcd10^{BECKO}$:mT/mG mice (with the same protocol as Fig. 7i. Rebastinib did not alter the vascular structures in $Mfsd2aCreER^{T2}$:mT/mG (WT) animals. However, Rebastinib completely reversed the enlarged vascular areas resulting from increased vessel sprouting and dilation in $Pdcd10^{BECKO}$:mT/mG mice (Fig. 7l, m). Taken together, these results suggest that inhibition of Tie2 signaling by Tie2 inhibitor achieved therapeutic effects on CCM lesions.

Since Rebastinib inhibits the kinase activity of Tie2, Abl, and VEGFR2 at similar efficacies[52], effects of Rebastinib on these kinases were examined in the brain tissues from the Rebastinib therapeutic model (Supplementary Fig. 11a). Phosphorylations of Tie2, Abl, and VEGFR2 were all increased in $Pdcd10^{BECKO}$ brain

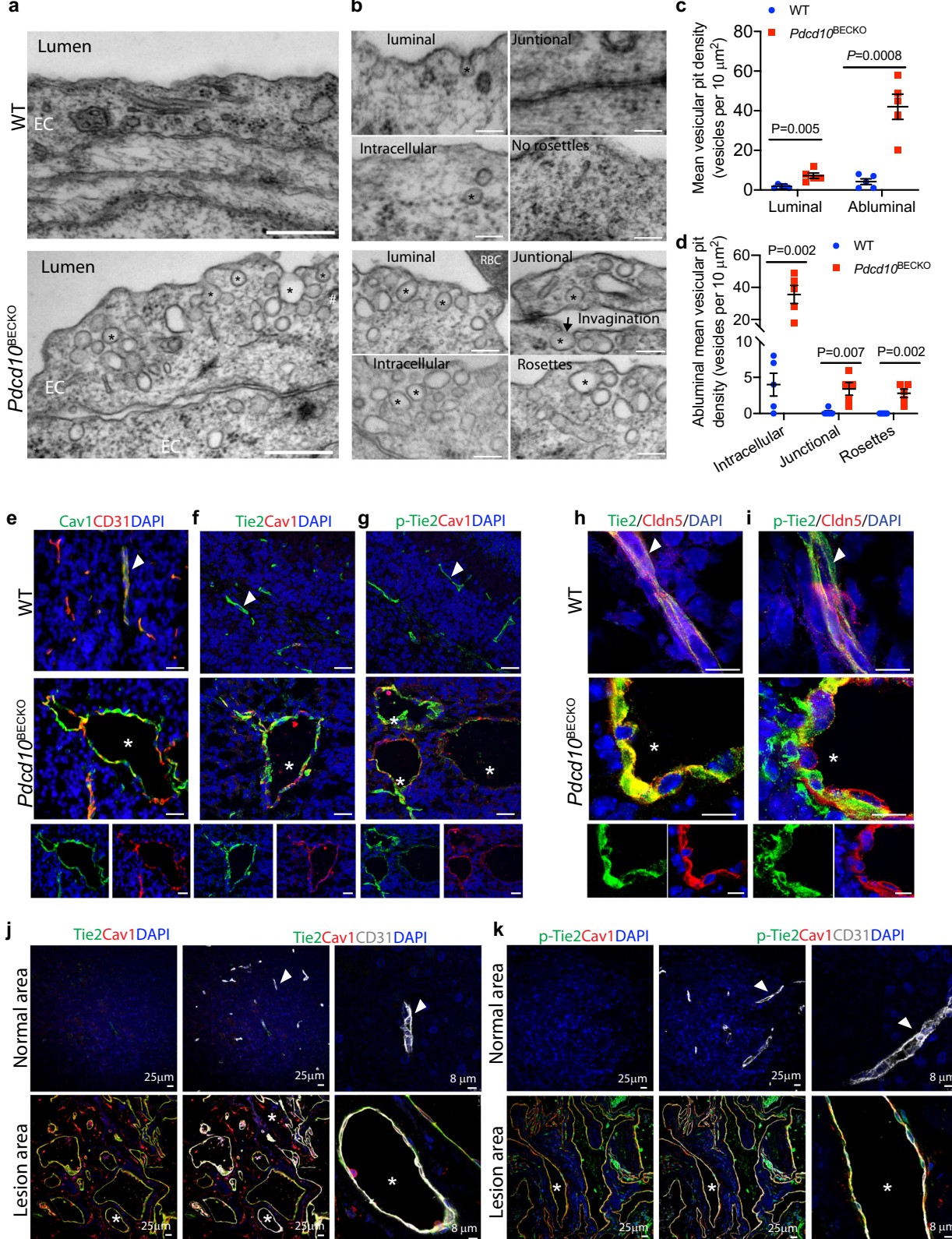

tissues. However, p-Tie2 and p-Abl, but not p-VEGFR2, were evidently attenuated by Rebastinib. Interestingly, Rebastinib had no effects on expression of Tie2 ligand Angpt2 and Tie2 regulator Cav1, but reduced the Tie2 downstream Akt (Supplementary Fig. 11b). The correlation between Abl and Tie2 in *Pdcd10*BECKO mice prompted us to determine the role of Abl in Tie2 signaling in Ccm3-deficient ECs. Consistent with the in vivo observations,

silencing of Ccm3 increased the total protein levels of Tie2 and Cav1, but not Abl or VEGFR2. Interestingly, Abl siRNA, but not Kdr (VEGFR2) siRNA, similar to Cav1 siRNA, attenuated increased total Tie2 in CCM3-deficient ECs. However, Abl expression was not altered by CCM3 deletion in mice or cultured ECs (Supplementary Fig. 11b-c). It is plausible that Cav1 and the non-receptor kinase Abl cooperatively regulate Tie2 expression

**Fig. 4 CCM lesions exhibit increased caveolae vesicles and Cav1-Tie2 signaling. a–i** Caveolae and Cav1-Tie2 signaling characterization in the early lesion model. *Pdcd10*BECKO pups were fed with tamoxifen at P1 to P3 and tissues were harvested at P15. **a–d** Caveolae characterization in CCM lesions by EM. Cerebral sections from WT and *Pdcd10*BECKO mice were subjected to EM. Representative images from 5 mice are shown. **a** Caveolae was indicated by asterisks while clathrin-coated pit by #. **b** Caveolae localization and forms. Luminal invagination, intracellular caveolae, and caveolae rosettes (clusters) are indicated. **c** Mean vesicular pit density at luminal and abluminal sides are quantified per 10 mm². **d** Mean vesicular pit density at intracellular, junctional, and rosettes are quantified per 10 mm². 20 EM section per mouse and $n = 5$. Data are means ± SEM. *P* values are indicated, unpaired two-sided Student's *t*-test. **e–g** Cav1-Tie2 signaling was upregulated in mouse CCM lesions. Cerebral sections from WT and *Pdcd10*BECKO mice were immunostained for Cav1 with CD31 (**e**), Tie2 (**f**), or p-Tie2 (**g**). Representative merged images of normal vessels (arrowheads) and CCM lesions (asterisks) are shown with individual color images of *Pdcd10*BECKO at the bottom. $n = 6$. **h, i** Tie2 was not localized in EC nucleus. Brain sections from WT and *Pdcd10*BECKO mice were co-stained with Tie2 and Cldn5 (**h**) or p-Tie2 and Cldn5 (**i**). Images were captured under SP8 STED microscopy. Representative images for WT normal vessels (arrowheads) and CCM lesions (asterisks) in *Pdcd10*BECKO mice are shown with individual color images of *Pdcd10*BECKO at the bottom. $n = 6$. **j, k** Cav1-Tie2 signaling was upregulated in human CCM lesions. Human CCM specimens were co-immunostained for CD31 and Cav1 with Tie2 (**j**) or p-Tie2 (**k**). Representative images of human CCM3 samples captured under SP8 STED microscopy are shown with high-power merged 4-color images on the right. $n = 6$. Arrowheads indicate normal vessels whereas asterisks indicate for lesions. Scale bar: 500 nm (**a**); 150 nm (**b**); 25 μm (**e**; left and middle panels in (**j**) and (**k**)); 8 μm (**h, i**; right panels in (**j**) and (**k**)). Source data are provided as a Source data file.

and signaling in Ccm3-deficient ECs, and it needs to be further investigated.

**Tie2 inhibition normalizes EC barrier function.** The unique feature of the brain vasculature is the BBB formed by the brain neurovascular unit. Since Ccm3 deletion increased caveolae in brain ECs which has been associated with increased BBB disruption[30,31], an important question was whether or not Tie2 inhibition could rescue BBB integrity and function, preventing CCM lesion progression. Two-photon microscopy was not applicable to visualize and quantify vascular permeability. Therefore, we assessed brain vascular leakages and BBB function using various assays. Vascular leakage was first visualized by TMR-dextran perfusion followed by whole-mount staining in the Rebastinib therapeutic model, i.e., Ccm3 deletion at P1–P3 followed by Rebastinib on P4–P14, and the permeability assay was performed on P15 (Fig. 8a). The *Pdcd10*BECKO mice exhibited local brain vascular leakage, which was indicated by accumulation of TMR-dextran dye outside the vasculatures (Fig. 8b with quantification in 8c). The vascular leakage in the brain was prevented by Rebastinib (Fig. 8b, c). The same effects of Rebastinib were obtained on the retinal hyper-permeability (Fig. 8d with quantification in 8e). The vascular leakage in *Pdcd10*BECKO mice was consistent with the pericyte dissociation and EC junctional disruption observed by immunostaining for junctional protein and EM for tight junctions (see Figs. 2 and 3 and Supplementary Fig. 5). Consistent with the restored barrier function, EC–PC interaction (Fig. 7) and EC tight junctions (TJ) (Fig. 8f, g) in *Pdcd10*BECKO brain were restored by Rebastinib.

In addition to junctional disruptions, increased transcytosis by caveolae vesicles in brain EC contributes to vascular permeability[30,31,39]. To determine if EC transcytosis was augmented by *Ccm3* deletion and if Rebastinib could diminish this effect, we performed horseradish peroxidase (HRP)/DAB-based transcytosis assay followed by EM ultrastructural analyses[30,31,39] in the Rebastinib therapeutic model (see Fig. 8a). Indeed, caveolae-mediated transcytosis (DAB-positive dots) was dramatically enhanced in *Pdcd10*BECKO mice compared to WT brain, and this augment was diminished by the Rebastinib treatment (Fig. 8h with quantification in 8i). Inhibition of Rebastinib on the Cav1-mediated EC transcytosis suggested a reciprocal regulation between Tie2 and Cav1/caveolae.

Based on these analyses, we concluded that both junctional disruption and enhanced transcytosis contributed to increased vascular hyper-permeability in *Pdcd10*BECKO mice, and Tie2 inhibition normalized EC barrier function in CCM mouse models.

**Tie2 inhibition prevents progression of pre-existing lesions.** We finally tested the therapeutic effects of Rebastinib on pre-existing CCMs in a severe model, i.e., Ccm3 deletion at P1 followed by administration of the inhibitor at P15 when the CCM lesion fully bloomed (Fig. 9a). Results showed that Rebastinib prevented further CCM lesion progression of pre-existing lesions compared with the vehicle-treated group. However, the inhibitor could not completely reverse the pre-existing lesions (Fig. 9b with quantifications in 9c). More experiments need to test the therapeutic effects with more numbers of mice and in less-progressive models, such as the moderate and mild CCM lesion models with delayed injection of tamoxifen for Ccm3 deletions.

## Discussion

Although CCMs can have severe clinical consequences, their pathogenesis is still poorly understood. This is in part because animal models that fully recapitulate human CCM pathogenesis are lacking. Currently, EC-specific *Ccm* deletions in whole body are common CCM models. However, the CCM lesion-carrier mice from whole-body EC-specific *Ccm* ablation do not survive to adulthood, limiting all previous studies to perinatal mice[10–15,27]. Critically, here we discover that *Pdcd10*ECKO pups with whole-body EC deletion of *Pdcd10* (*Ccm3*) exhibit severe splenic hemorrhage and rupture, leading to perinatal death. In contrast, BEC-specific Ccm3 deletion (*Pdcd10*BECKO) does not develop lesions in the spleen or other tissues and is able to survive to adulthood. *Pdcd10*BECKO mice have CCMs ranging in size from early-stage, isolated caverns to large, multi-cavernous lesions. Late-stage CCM lesions in old *Pdcd10*BECKO mice (6 months) display many characteristics of human CCM lesions, including hemosiderin deposits, increased EC proliferation, and pericyte loss. Thus, *Pdcd10*BECKO mice are a reliable model that closely resembles human CCM pathogenesis. A similar model has been established for a *Ccm2* or *Ccm3* deletion with the Slcoc1-CreERT2 system[34,53]. We then employ two-photon microscopy imaging technique and visualize the dynamic changes of the vasculature and EC-pericyte interactions during CCM lesion formation. We observe enlarged diameter of the brain vasculatures with a gradual loss of pericytes from capillary/post-capillary venules in *Pdcd10*BECKO mice. These phenotypes are correlated with an increased number of caveolae by EM and caveolae structural protein Cav1 protein by immunofluorescence staining and immunoblotting in *Pdcd10*BECKO brain ECs. Moreover, we detect increased total- and p-Tie2 which are co-localized with Cav1 in mouse and human CCM lesions. In vitro studies elucidate that CCM3 deficiency in ECs causes enhanced Cav1-Tie2 signaling, EC junctional disruption, and lumen enlargement which could be significantly attenuated by

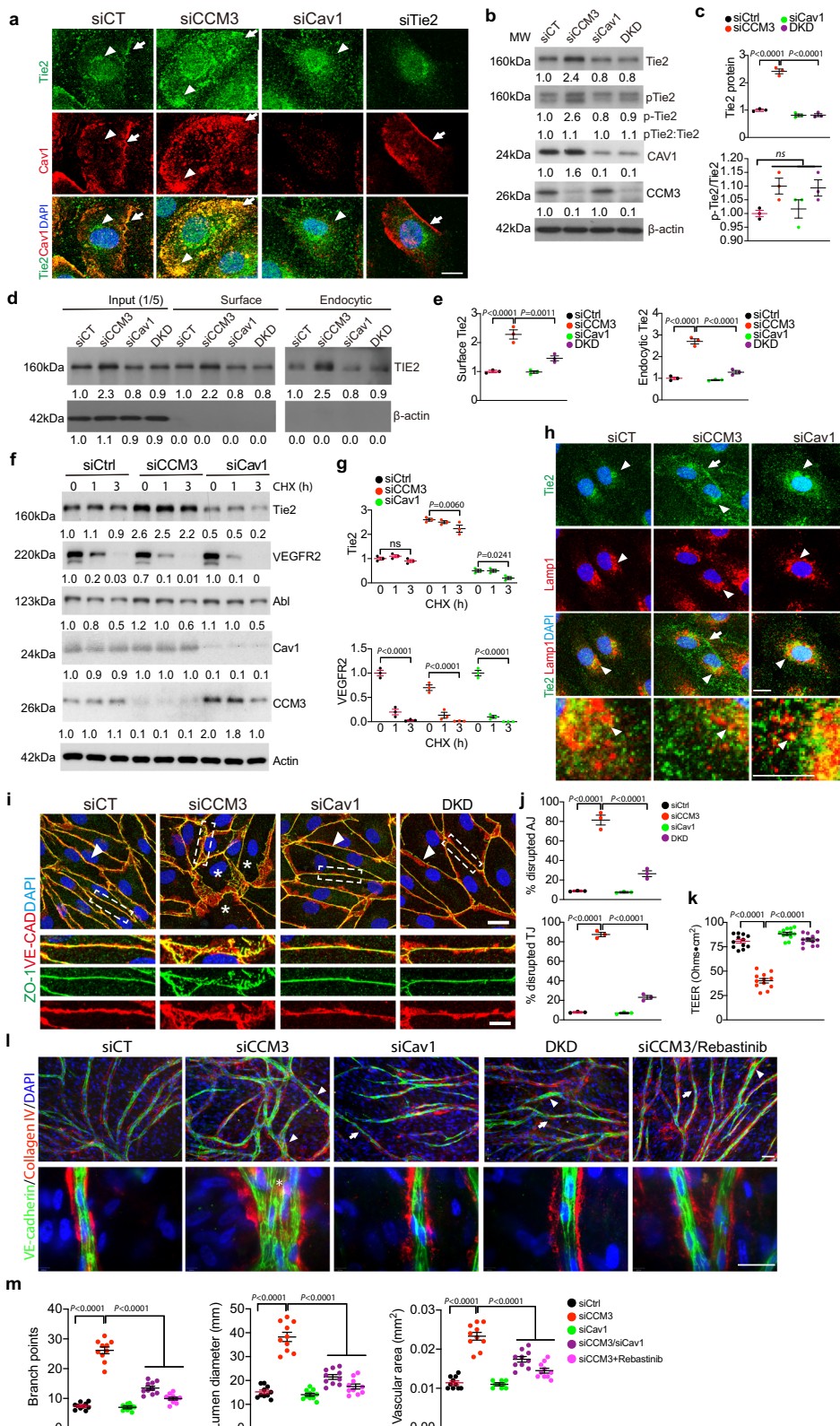

Cav1 silencing. Using mouse genetic approaches, we show that Cav1 deficiency prevents Tie2 signaling and blocks CCM lesion development caused by brain EC-specific *Ccm3* inactivation. Importantly, our results demonstrate that a Tie2 inhibitor Rebastinib (a phase 1b/phase 2 clinical trial drug) ameliorates CCM lesion progression in the mouse CCM models, correlating with normalized EC junction, EC-pericyte interaction, and

vascular integrity. Our study reveals a mechanism by which CCM3 mutation-induced Cav1-Tie2 signaling contributes to the progression of the CCM disease (Fig. 10).

A key finding of the current study is the visualization of the vasculature and EC-pericyte interactions during CCM lesion progression using confocal and two-photon microscopy. Previous studies from human samples[1–3] and CCM2-deficient mouse

**Fig. 5 Cav1 silencing attenuates CCM3 loss-augmented Tie2 signaling and lumen enlargement in in vitro models.** HBMVECs were transfected with indicated siRNAs for 72 h. **a** Co-immunofluorescence staining with Tie2 and Cav1. Arrows and arrowheads indicate membrane and intracellular Tie2/Cav1, respectively. **b, c** Cav1 co-silencing attenuates Tie2 expression. Western blotting for Cav1-Tie2 signaling. Relative protein levels and p-Tie2/Tie2 ratios were quantified by taking Ctrl siRNA as 1.0. $n = 3$. **d, e** Biotinylation assay for cell surface and endocytic Tie2. Aliquot of input (1/5) was loaded as controls. Relative surface and endocytic Tie2 were quantified by taking Ctrl siRNA as 1.0. $n = 3$. **f, g** Tie2 stability was determined by cycloheximide (CHX) treatment followed by western blot with respective antibodies. The samples were derived from the same experiment and that gels/blots were processed in parallel. Relative protein levels were quantified by taking Ctrl siRNA as 1.0. $n = 3$. **h** Immunofluorescence staining with Tie2 and Lamp1. Arrows and arrowheads membrane-localized and lysosomal Tie2, respectively. High-power images of Tie2/Lamp1 co-localization are shown at bottom. **i, j** Adherens junctions (AJ) and tight junctions (TJ) were visualized by immunostaining, and percentages of disrupted (discontinuous) junctions are quantified in panel (**j**) (10 microscope fields in each group), $n = 3$. **k** Barrier function of ECs cultured on fibronectin-coated ECIS was assessed for transendothelial electric resistance (TEER; expressed as Ohms multiplied by $cm^2$). $n = 12$ wells per group. **l, m** Organotypic angiogenesis assay in the absence or presence of Tie2 inhibitor Rebastinib (1 μM). EC sprouts and lumens were visualized by VE-cadherin and extracellular matrix collagen-IV staining. Arrows and arrowheads indicate normal lumen and dilated lumen, respectively. Number of branch points, mean lumen diameter, and lumen areas are quantified (**m**). All experiments were repeated at least three times. $n = 10$. Data are means ± SEM. $P$ values are indicated, one-way ANOVA followed by Tukey's multiple comparisons test (**b, c, d, e, j, k, m**); two-way ANOVA followed by Tukey's multiple comparisons test (**g**). Scale bar: 20 μm (**a, h, i**); 100 μm (**l**). Source data are provided as a Source data file.

models have suggested that CCM lesions affect the venous bed, but not the arterial compartment[54]. All these studies have been based on histology, immunostaining, EM, and micro-CT analyses, providing important information regarding vessel structural changes and EC-pericyte interactions[27,55]. However, dynamic vascular changes during CCM development is not determined. We have developed reproducible methods to track individual microvascular structures over time using transcranial two-photon microscopy and has used these methods to follow the development of vessels in EC-specific reporter mice[56]. Unlike previous models, $Pdcd10^{BECKO}$ mice develop CCM lesions not only in the cerebellum but also in the cerebrum. These cerebral lesions allow us to visualize vascular lesion formation and progression at various stages using transcranial two-photon microscopy with live imaging in these mice under the background of mT/mG reporter mice ($Pdcd10^{BECKO}$:mT/mG). We reveal that lesions in $Pdcd10^{BECKO}$:mT/mG mice have significantly greater capillary branches and vessel diameters with turbulent blood flow compared to control $Mfsd2a$CreER$^{T2}$:mT/mG mice. These capillaries consist of clusters of enlarged endothelial channels ('caverns') that are arranged back-to-back to form densely packed sinusoids as described for human lesions[1–3]. Using high-resolution optical imaging in live mice labeled with the Fluorescent Nissl dye NeuroTrace 500/525[41], we observe that pericytes have multiple slender processes that usually extend longitudinally to span several branches within the cerebral vessels. We reveal that pericytes shorten their processes and gradually dissociate from dilated vessels within CCM lesions, resulting in lack of pericytes surrounding the single layer of endothelium in the late CCM lesions. This is further confirmed by immunofluorescence staining of brain sections that pericytes in moderate CCM lesions exhibit loose associations with EC with lower rate of vessel coverage compared to normal brain microvessels. Interestingly, different from the previous report on the endothelial gain of α-SMA expression (EndMT) within the CCM lesion[14], we observe that α-SMA expression is partially in pericytes but not in ECs during early CCM formation in both brain and retinas of $Pdcd10^{BECKO}$ mice. This is based on our analyses using the super-resolution STED microscopy of brain sections and confocal imaging of whole-mount retina staining. It has been shown that brain and retinal pericyte basally express α-SMA[57]. Interestingly, α-SMA expression in brain pericytes can be specifically upregulated by TGF-β[58], a cytokine increased in CCM lesions[14]. The contribution of the SMA expression in pericytes and potentially in other cells to CCM pathogenesis is unknown, and further investigation is needed.

Caveolae is limited in the mouse brain vasculatures and increased caveolae is associated with increased vascular permeability[30,31]. We observe a dramatic increase in the number of caveolae in $Pdcd10^{BECKO}$ brain ECs as visualized by EM, and increased caveolae structural protein Cav1 protein as detected by immunofluorescence staining and immunoblotting. Caveolae are detected in intracellular, luminal, and abluminal compartments with "rosettes" patterns. Similarly, Cav1 protein is distributed throughout the cells in CCM3-deficient ECs compared to more membrane-association in WT cells. These results indicate an active caveolae-mediated trafficking in $Pdcd10^{BECKO}$ brain EC[29]. Indeed, mechanistic studies support that Cav1 mediates Tie2 surface expression and trafficking in CCM3-deficinet ECs. Importantly, Cav1 and Tie2 protein and Tie2 activity are highly upregulated in the lesions of human CCM specimen. Moreover, genetic rescue study with $Cav1$-deficient mice supports that increased caveolae is required for CCM3 loss-induced Tie2 signaling, and subsequently vascular permeability and CCM lesion formation. It has been reported that Tie2 activation by ligand Angpt1 or Angpt2 results in co-localization of Tie2 with clathrin-coated vesicles followed by Tie2 endocytosis and degradation[59]. In contrast, we observe that colocalizations of Tie2 with clathrin-positive endosome and subsequently with lysosomes were dramatically reduced in CCM3-KO cells, concomitantly increased co-localization of Tie2 with Cav1. We propose that Cav1/caveolae mediate Tie2 intracellular trafficking, leading to increased Tie2 protein stabilization and signaling in CCM3-deficient ECs.

Our study demonstrates CCM3 regulates Angpt2 and Tie2 through distinct mechanisms. We have previously shown that CCM3 tightly associates with GCK kinase STK24, and through this interaction CCM3 suppresses STK25-Unc13B-mediated exocytosis and Angpt2 secretion. Therefore, CCM3 loss release STK25 to facilitate Unc13-driven secretion of Angpt2 ligand[27]. Our current study demonstrates that CCM3 deletion enhances the receptor Tie2 in ECs via caveolae-dependent but Unc13B-mediated exocytosis-independent mechanisms. On the other hand, we show that Cav1-caveolae regulates Tie2 signaling via modulating the Tie2 protein stability. How exactly CCM3 loss increases Cav1-caveolae is unclear. It seems that CCM3 has broad cellular function in controlling vesicle trafficking. This is supported by several studies: (1) CCM3 stabilizes GCK kinases to promote Golgi assembly[60]; (2) Loss-of-function mutants of fly CCM3 or GCK kinase ortholog have dilated tracheal tubes, which resembles dilated blood vessels found in CCM patients[61]. This tube dilation phenotype can be suppressed by the reduction in the expression of N-ethylmaleimide sensitive factor 2 (NSF2), a protein involved in SNARE recycling and the secondary phase of exocytosis. (3) Interestingly, more recent studies suggest that CCM3 via the striatin-interacting phosphatase and kinase

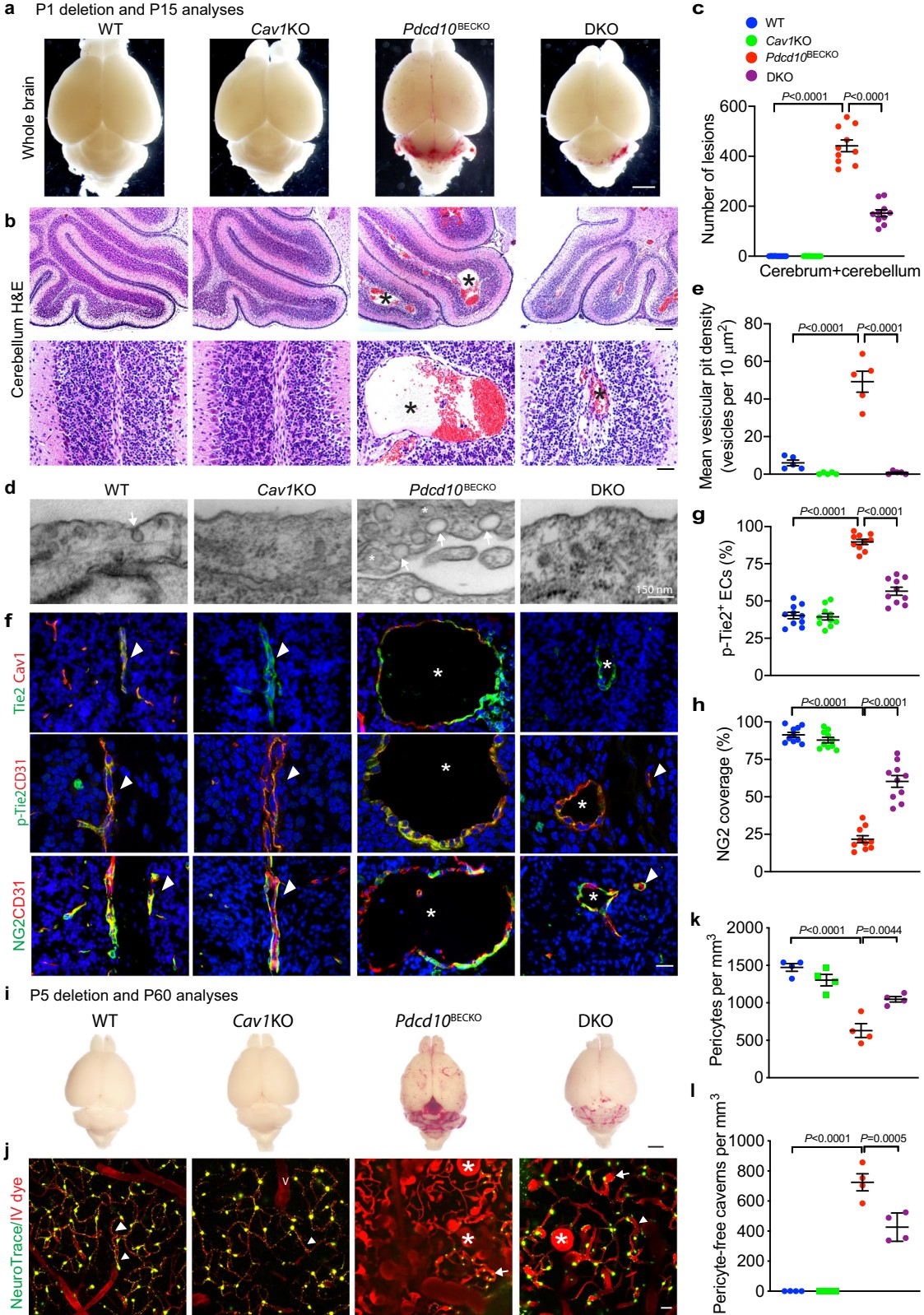

(STRIPAK) complex regulates Golgi stability and CDC42 or Rab11-dependent receptor endocytic recycling[62]. The STRIPAK complex was identified by proteomic analysis which revealed that CCM3 and GCK kinases together with PP2A were present in a striatin family-associated protein complex[63]. Interestingly, striatin family members in STRIPAK complex contain a caveolin scaffolding domain-interaction motif and have been shown to associate with Cav1[64]; and this association could localize the STRIPAK complex to caveolae to regulate membrane signaling, endocytosis, and vesicle trafficking in cells[62]. Given the dynamic association of CCM3 with the STRIPAK complex and that caveolae is dramatically increased upon CCM3 deletion, it worth exploring if the STRIPAK complex is enriched in caveolae in CCM3-defective ECs to mediate Tie2 cellular trafficking and

**Fig. 6 Cav1 deficiency reduces CCM lesions and normalizes PC-EC associations in Pdcd10BECKO mice.** Ccm3lox/lox (WT), Cav−/− (Cav1-KO), Pdcd10BECKO, and Pdcd10BECKO:Cav−/− (DKO) pups were fed with tamoxifen from P1 (**a–h**) or P5 (**i–l**) to induce deletion of Ccm3, and cerebella were harvested at P15 and P60, respectively. **a** Images for fresh brain tissue. **b**, **c** H&E staining and lesion quantifications. Representative images are shown where lesions are indicated asterisks and n = 10. **d**, **e** Caveolae characterization by EM. Cerebral sections from each strain were subjected to EM. Representative images from 5 mice are shown. Luminal invagination and intracellular caveolae are indicated. **e** Mean vesicular pit density at luminal and abluminal sides are quantified per 10 mm². 10 EM section per mouse and n = 5. **f–h** Co-staining for Tie2/Cav1, p-Tie2/CD31, and NG2/CD31. Representative images of normal vessels (arrowheads) and CCM lesions (asterisks) are shown and n = 10. Quantification of p-Tie2-positive ECs and NG2 coverage on CD31 vessels, i.e., % green (p-Tie2 or NG2)-conjugated area/total red (CD31) areas by Image J (**g**, **h**). **i** CCM lesions at P60 were visualized by whole-brain imaging. **j** Characterization of CCM lesions and EC-pericyte interactions by confocal microscopy. Two-month-old mice were perfused with a Texas Red IV dye and NeuroTrace (pericytes) followed by confocal microscopy. **j** Representative images from each group are shown. Arrowheads indicate pericyte processes whereas arrows indicate pericytes with shortened processes in lesions (asterisks). **k**, **l** Pericyte density (number/area) and pericyte-free caverns were quantified. n = 4. Data are means ± SEM. P values are indicated, one-way ANOVA followed by Tukey's multiple comparisons test (**c**, **e**, **g**, **k**, **l**). Scale bars: 2 mm (**a**, **i**); 100 μm (upper panel in (**b**)); 25 μm (lower panel in (**b**)); 150 nm (**d**); 16 μm (**f**); 50 μm (**j**). Source data are provided as a Source data file.

signaling. Another possible link between CCM3 and caveolae is through regulation of Msdf2a. Mfsd2a has been identified as a lipid transporter at the luminal plasma membrane of brain ECs[31]. Mfsd2a is not only specifically expressed in the brain ECs, but also acts as a critical regulator in brain vascular function. Genetic ablation of Mfsd2a in mice caused a leaky blood–brain barrier (BBB)[30,31]. Moreover, increased caveolae is reported to mediate the increased transcytosis and EC permeability in Mfsd2a-deficient mice without disruption of EC tight junctions[30,31]. In supporting this model, Ccm3-deficient mice exhibit increased caveolae, vascular transcytosis, and EC permeability. Of note, CCM3, but not CCM1 or CCM2, specifically interacts with GCKs and the STRIPAK complex. Similarly, the Angpt2-Tie2 signaling is increased in CCM3 knockdown, but not in CCM1 or CCM2-deficient EC[27]. Therefore, our newly discovered Cav1-Tie2 pathways could be unique to CCM3 pathogenesis. It needs to explore whether or the Cav1-Tie2 pathways are related to the other inherited forms and "sporadic" CCMs.

Identifying the Angpt2-Tie2 signaling involved in CCM lesion formation and progression is initially surprising. Classically, Angpt1 activates Tie2 in normal blood EC, promoting EC junction formation and quiescence while Angpt2 acts as an Angpt1 antagonist, counteracting the stabilizing action of Angpt1 to increase EC permeability[23,24]. However, it is now recognized that Angpt2 is able to induce Tie2 phosphorylation in a context-dependent manner[26,48–51,65,66]. Angpt2 can act as a Tie2 agonist in lymphatics[47,65,67–70], and the Angpt2 agonist activity is attributed to a low level of VE-PTP expression[71]. Interestingly, it has been reported that endothelial Tie1 is essential for the agonist activity of autocrine Angpt2 by directly interacting with Tie2 to promote Angpt2-induced p-Tie2[48–51], promoting Angpt2-induced the enlargement of post-capillary venules and capillaries in the trachea[48,49]. It needs further investigation to determine if and how VE-PTP and Tie1 contribute to the Angpt2-Tie2 signaling and CCM lesion formation in the future studies.

It worth mentioning that Tie2 gain-of-function (GOF) mutations in humans lead to hyper-activation of Tie2 and consequently induce venule malformations[72]. Distinct from the CCM lesion in anatomy and gene mutations, venule malformations are composed of ectatic venous channels found usually in the head, neck, limbs, and trunk[72]. GOF mutant of Tie2 receptor, probably through Abl kinase, cause ligand-independent sustained activation of downstream Akt. Moreover, Abl kinase is required for Tie2 expression[73]. Therefore, Abl inhibitors and rapamycin effectively reduced mutant Tie2-induced Akt signaling and reduced venule malformations in mice and humans[74]. These studies suggest that over-activated Tie2 does not tighten the EC junctions, rather destabilizes the vasculature that may cause vascular malformation. Therefore, it is considerable that Tie2 signaling pathway plays distinct roles in both vascular

quiescence and angiogenesis. Our early work has demonstrated that CCM3 loss-augmented Angpt2, likely via an autocrine fashion, activates Tie2 in cultured ECs as neutralization of Angpt2 diminishes Tie2 activation in CCM3-deficient ECs[27]. However, it has been puzzling how "a weak agonist" Angpt2 would be able to induce the level of Tie2 activation as observed in the Tie2 GOF causing vascular malformation mutations. Our current study has addressed this puzzle. We demonstrate that CCM3 deletion not only augments exocytosis-mediated Angpt2 release, but also enhances total and p-Tie2 in ECs via caveolae-dependent mechanisms. Therefore, Angpt2 activates Tie2 in autocrine fashion that could drive vascular malformation. Consistently, either neutralization of Angpt2[27] or Tie2 inhibitor (current study) greatly attenuates, Tie2 activation and CCM disease pathogenesis. A combination therapy needs to be explored in the CCM mouse models. CCM3-defective ECs increased total Tie2 and consequently the p-Tie2 while Tie2 GOF increases p-Tie2 or/and downstream Akt without increasing the total Tie2 levels. Although CCM3-defective ECs also induced the Abl-Akt signaling which could be blocked by Rebastinib, the contribution of Abl to CCM lesion progression need further investigations. On the other hand, it is unclear whether or not the Tie2 GOF-mediated Abl-Akt signaling is cav1/caveolae-dependent. Our two-photon microscope imaging indicates that CCM occurs primarily at the post-capillary venules without pericyte coverage. Moreover, mutant ECs expressing Tie2-L914F, the most frequent mutation identified in venule malformations patients, form enlarged lumens with the paucity of pericytes, mimicking vascular lesions present in an in vitro model[75]. In this regard, increased Tie2 activity and similar vascular anomaly suggest a potential common pathogenesis between CCM and Tie2 GOF-dependent venule malformation.

Taken together, our studies have identified the caveolae-mediated Tie2 receptor signaling in the presence of its ligand Angpt2 released by exosomes from CCM3-deficiency ECs are critical for CCM pathogenesis. Therefore, inhibition of Angpt2-Tie2 signaling offers a therapeutic approach to treat CCM disease.

## Methods

**Generation of tamoxifen-inducible BEC-specific CCM3-deficient mice.** Tomato reporter mice were purchased from The Jackson Laboratory (Stock No. 007576; strain name STOCK Gt (ROSA)26Sortm4(ACTB-tdTomato,-EGFP)Luo/J). The Mfsd2a-CreERT2 mouse line was obtained from Dr. Bin Zhou's lab[35]. Pdcd10fl/fl mice were crossed with Mfsd2aCreERT2 mice, in which CreERT2 recombinase expression is driven by the Mfsd2a promoter to generate Pdcd10fl/fl:Mfsd2aCreERT2 mice. For the in vivo tamoxifen-induced Pdcd10 deletion, tamoxifen (Sigma, T5648) was diluted to 10 mg/ml and fed to mice at a dose of 50 μg/day from P1 to P3 (regular 'severe' model)[76]. For pups, littermates were randomly separated into vehicle (WT) and tamoxifen-fed (Pdcd10BECKO) groups such that the sex ratios of mice in both groups were equal. For adult mice, male, and female animals were used in equal numbers for all experiments. Mice were housed in the animal care facility of Yale University under standard pathogen-free conditions with a 12 h light/dark schedule

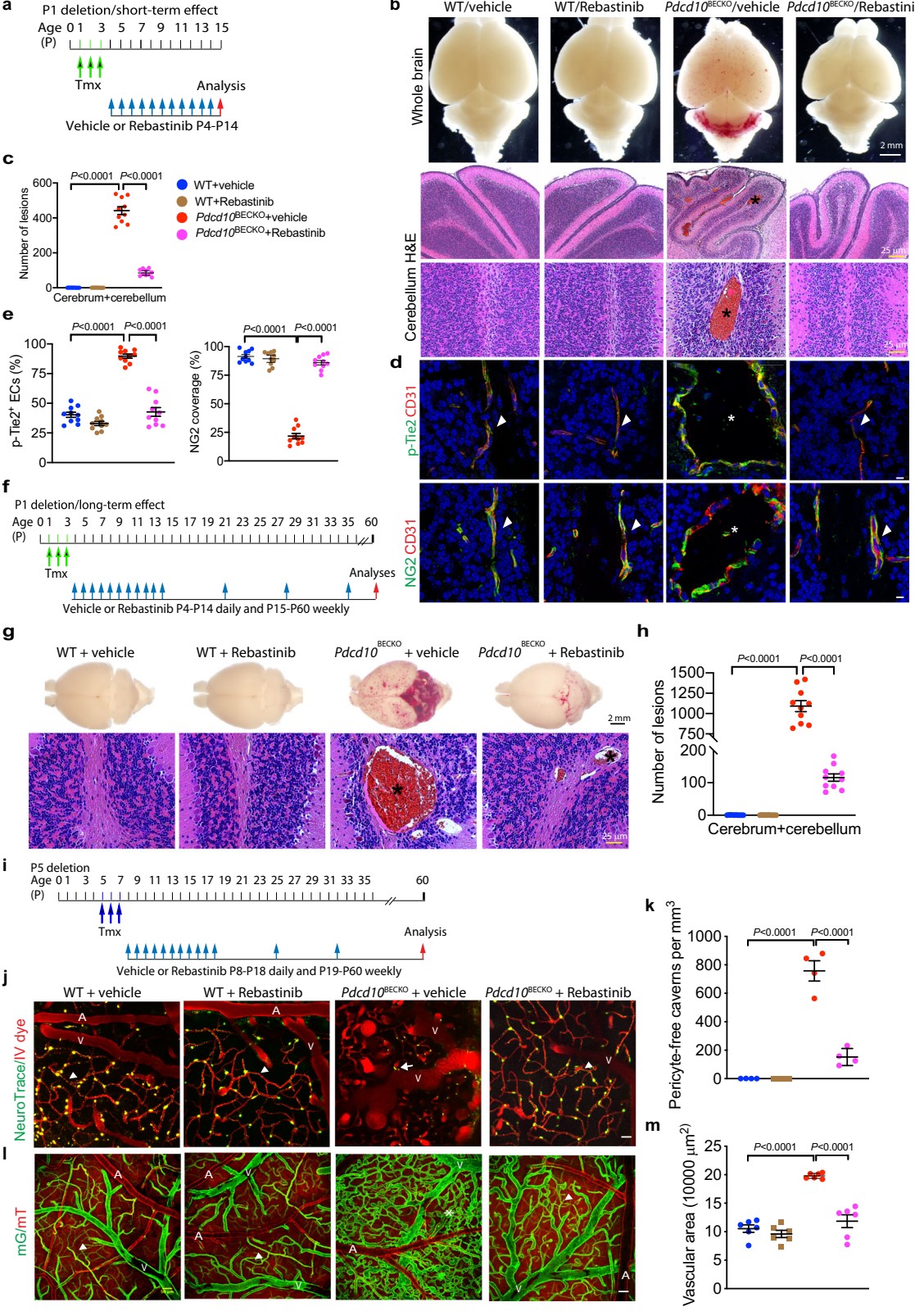

and provided with food and water ad libitum, temperature was between 20 and 24 °C and relative humidity between 45 and 65 rH. Mice were cared for in accordance with the National Institutes of Health guidelines, and all procedures were approved by the Yale University Animal Care and Use Committee.

## Cranial window preparation and in vivo imaging

*Mouse cranial windows.* Windows were prepared as previously described[56]. Briefly, animals were anesthetized via i.p. injections of 100 mg/kg ketamine and 10 mg/kg

xylazine. A 3–4 mm craniotomy was prepared over the somatosensory cortex, and the underlying dura was removed. After dye labeling, a #0 cover glass was placed over the exposed area and sealed with dental cement. For all experiments, mice were imaged whilst anesthetized with a mixture of ketamine and xylazine as stated above. For repeated in vivo imaging, a nut was affixed to the skull and embedded in dental cement for head immobilization and window orientation. Cerebral vessels were visualized by intravenous (i.v.) injection of EBD. In vivo images were acquired using an upright Leica SP5 confocal microscope. A ×20 water immersion objective (1.0 NA, Leica) was used for optimal fluorophore excitation and emission

**Fig. 7 Tie2 inhibition rescued EC-pericyte associations with a therapeutic effect on CCM lesion progression. a–h** Short-term effects of Rebastinib on CCM lesions. **a** A diagram for the protocol. Vehicle or Rebastinib was subcutaneously injected into WT and P1 deletion Pdcd10BECKO mice, and brain tissues were harvested at P15. **b** Images of fresh brain tissue and H&E staining of brain sections. **c** Lesion quantifications from the H&E staining. $n = 10$. **d** Cerebellum sections were stained for CD31 with p-Tie2 or NG2, respectively. Representative images of normal vessels (arrowheads) and CCM lesions (asterisks) are shown captured by SP8 STED microscopy. **e** Quantification of p-Tie2 and NG2 coverage on CD31 vessels, $n = 10$. **f–h** Long-term effects of Rebastinib on CCM lesions. **f** A diagram for the protocol. Vehicle or Rebastinib was subcutaneously injected into WT and P1 deletion Pdcd10BECKO mice, and brain tissues were harvested at P60. **g** Images of fresh brain tissue and H&E staining of brain sections. **i** Lesion quantifications, $n = 10$. **i–m** Characterization of CCM lesions and EC-pericyte interactions by confocal microscopy. **i** A diagram for the protocol. Vehicle or Rebastinib was subcutaneously injected into WT and P5 deletion Pdcd10BECKO mice, and mice were analyzed at P60. **j** Two-month-old mice were perfused with a Texas Red IV dye and NeuroTrace (pericytes) followed by confocal microscopy. Representative images from each group are shown. Arrowheads indicate pericyte processes whereas arrows indicate pericytes dissociating from vessels. **k** Pericyte-free caverns were quantified. $n = 4$. **l, m** Characterization of CCM lesions and perfusion in WT (Mfsd2aCreERT2:mT/mG) and Pdcd10BECKO:mT/mG mice were visualized by confocal microscopy. **l** Representative images of normal or normalized microvessels (arrowheads) and CCM lesions (asterisks) are shown. Veins (V) and artery are indicated. **m** Vascular areas are quantified. $n = 6$. Data are means ± SEM. P values are indicated, one-way ANOVA followed by Tukey's multiple comparisons test (**c, e, h, k, m**). Scale bars: 2 mm (upper panels in (**b**) and (**g**)); 25 μm (lower panels in (**b**) and (**g**)); 8 μm (**d**); 50 μm (**j, l**). Source data are provided as a Source data file.

separation. For confocal imaging, the following wavelengths were used: 488 nm for GFP, 561 nm for tdTomato, and 633 nm for EBD.

*Visualization of pericytes.* Fluorescent Nissl dye NeuroTrace 500/525 was used to specifically label brain pericytes[41]. NeuroTrace 500/525 dye was applied topically (1:25 dilution in PBS) to the cortical surface for 5 min and then thoroughly washed. For topical application bright labeling was evident 3 h after labeling and remained for at least 48 h. Topical application of this dye through a cranial window led to labeling of a distinct population of cells lining cerebral blood vessels up to 400-μm deep in the cortex. In vivo images were acquired using a two-photon microscope (Prairie Technologies) equipped with a mode-locked MaiTai two-photon laser (Spectra Physics) and ×20 water immersion objective (Zeiss 1.0 NA). The two-photon laser was tuned to the following wavelengths for optimal excitation of particular fluorophores: 1000 nm for NeuroTrace 500/525 and Evans Blue Dye.

Quantifications of cavern number, vascular area, and pericyte number. High-resolution fluorescence images at 1024 pixels × 1024 pixels from regions of interest (ROIs) which equals 738.10 mm × 738.10 mm were acquired. Cavern number, vascular areas, and pericyte number were quantified from four separate ROI images from Z-projections of 60 μm depth from the cerebrum cortical surface for each mouse group captured from the cerebrum cortex. For imaged volume (mm³), microns were converted to mm which is 0.7381 mm × 0.7381 mm. To calculate volume, we multiplied 0.7381 × 0.7381 by the depth (e.g., 60 mm therefore 0.06 mm), i.e., mm³ = 0.7381 × 0.7381 × 0.06. Cavern numbers were calculated for total numbers per mm³ imaged volume. For the imaged vasculature, vascular areas instead of vascular volume were quantified. Specifically, GFP⁺ vascular areas were calculated after merging 60-layer images (1 mm per layer). Subsequently, number of pericytes per 10,000 mm² vascular area was quantified.

**TMR-dextran perfusion assays.** Tetramethylrhodamine (TMR)-dextran 10-kDa (Thermo Fisher Scientific, D1817) was dissolved in sterile normal saline at a concentration of 10 mg/ml followed by centrifugation at $10,000 \times g$ for 5 min. The supernatant was collected and protected from light for injection. Littermate of WT or KO pups at P15 were weighed and anesthetized with an intraperitoneal (IP) injection of anesthetic (katamine 100 mg/kg + xylazine 10 mg/kg). The retro-orbital injection was followed as we previously published[27]. Briefly, the lateral canthus of the left orbit was chosen as the injection site. A 31-gauge needle with a 3/10 ml insulin syringe was used to gently pierce into the mouse's orbital venous sinus about 2–3 mm with the bevel of the needle facing upward at a 45° angle. 10 ml per g body weight of TMR-dextran was injected into each pup and waited for 5 min. Brains and retinas were harvested and fixed for sectional (10 mm) or whole-mount fluorescent immunostaining. Mean fluorescence intensity (MFI) of TMR-dextran was quantified by Image J 1.52P.

**Horseradish peroxidase (HRP) perfusion and 3,3-diaminobenzidine (DAB) reaction assay for transcytosis.** P15 pups were briefly anesthetized (katamine 100 mg/kg + xylazine 10 mg/kg) and horseradish peroxidase, type II (Sigma-Aldrich, Cat# P8250-50KU, 0.5 mg/g body weight dissolved in 0.4 mL normal saline) was injected bi-laterally into the retro-orbital sinus. After 30 min of circulation, brains were dissected and cut into hemispheres. Brain hemispheres were fixed in in 2.5% glutaraldehyde and 2% paraformaldehyde in 0.1 M sodium cacodylate buffer pH 7.4 containing 2% sucrose for 1 h and post fixed in 1% osmium tetroxide at 4 °C overnight. Following fixation, brain hemispheres were washed in PBS and vibratome free-floating sections of 150 μm were collected, processed in a standard DAB reaction for 35 min, and mounted on 3% gelatin-coated slides. Slides were dehydrated through an ethanol series and mounted with Permount for brightfield imaging.

**Transmission electron microscopy (TEM) and electron tomography.** Tissues were fixed in 2.5% glutaraldehyde and 2% paraformaldehyde in 0.1 M sodium cacodylate buffer pH 7.4 containing 2% sucrose for 1 h and post fixed in 1% osmium tetroxide for 1 h. The sample was rinsed in buffer and en-bloc stained in aqueous 2% uranyl acetate for 1 h followed by rinsing in distilled water, dehydrated in an ethanol series, and infiltrated with Embed 812 (Electron Microscopy Sciences) resin. The samples were placed in silicone molds and baked at 60 °C for 24 h. Hardened blocked were sectioned using a Leica UltraCut UC7. Sixty-nanometer sections were collected on formvar-coated nickel grids and 250-nm sections on copper slot grids and stained using 2% uranyl acetate and lead citrate. Sixty-nanometer sections on grids were viewed FEI Tencai Biotwin TEM at 80 kV. Images were taken using Morada CCD and iTEM (Olympus) cellSens Dimension software.

For electron tomography, 250-nm-thick sections with 15 nm fiducial gold to aid alignment for tomography was done using FEI Tecnai TF20 at 200 kV. Data were collected using SerialEM (Boulder) on a FEI Eagle 4Kx4K CCD camera using tilt angles −60° to 60° and reconstructed using Imod4.9 (University of Colorado Boulder).

**Treatment with the Tie2 inhibitor Rebastinib in vivo.** Rebastinib (DCC-2036, #HY-13024, from MedChemExpress) was dissolved in 0.5% DMSO with 40% PEG300, 5% Tween-80, and 54.5% saline at 1 mg/ml. Rebastinib or vehicle control were subcutaneously injected from P4 to P14 daily, or P8 to P18 daily, and P19 to P60 weekly (10 μg/g body weight) to WT littermates or Pdcd10BECKO pups. Subsequently, mice were harvested at P15 or P60.

**Treatment with the Angpt2-neutralizing antibody in vivo.** Humanized mouse monoclonal Angpt2-neutralizing antibody (Genentech) was dissolved in sterile PBS at 1 mg/ml. Angpt2-neutralizing antibodies or control IgG1 were injected daily from P4 to P14 and weekly from P15 to P60 (i.p., 10 μg/g body weight) to WT littermates or Pdcd10BECKO pups. Mice were harvested at P60.

**Immunofluorescence analysis.** Mouse brains were harvested, fixed with 4% paraformaldehyde (PFA), embedded in OCT, and sectioned into 5-μm-thick sections. Slides were washed with PBS twice to remove OCT and then blocked in 5% donkey serum in 0.3% TritonX-100 in PBS for 30 min to prevent non-specific staining and to permeabilize the tissues. Slides were then incubated with primary antibodies against CD31, Caveolin-1, Claudin-5, Collagen-IV, Tie2, p-TIE2, NG2, VE-cadherin, ZO-1, SMA, or Angiopoietin-2 overnight at 4 °C. The next day, slides were washed with PBS three times and then incubated with secondary antibodies (1:400) at room temperature for 1 h (Supplementary Table 1). After washing in PBS three more times, slides were mounted with VECTASHIELD mounting medium with DAPI (Vector Laboratory). Images were taken under a confocal Leica SP8 STED microscope (Leica, Germany).

HBMVECs were fixed in 4% PFA for 20 min, washed with PBS, and permeabilized with 0.1% TritonX-100 for 2 min. After blocking for 30 min, primary and secondary antibodies were applied sequentially. Images were taken under a confocal Leica SP5 microscope (Leica, Germany).

**Fluorescent staining of whole-mount retinas.** Eyes were collected from neonatal mice on P6, P15, or P60 and then fixed in 4% PFA for 2 h on ice. For dissection, the cornea, lens, sclera, and hyaloid vessels were removed. Retinas were permeabilized in 0.5%Triton/PBS (5% normal donkey serum) overnight at 4 °C, followed by incubation with primary antibodies diluted 1:100 in 0.1% Triton/PBS (1% normal donkey serum) overnight at 4 °C. The following day, retinas were washed in PBS and then incubated with fluorescent secondary antibodies against Endomucin, CD31, Isolectin B4, SMA, NG2, Caveolin-1, Tie2, p-Tie2, or VE-cadherin

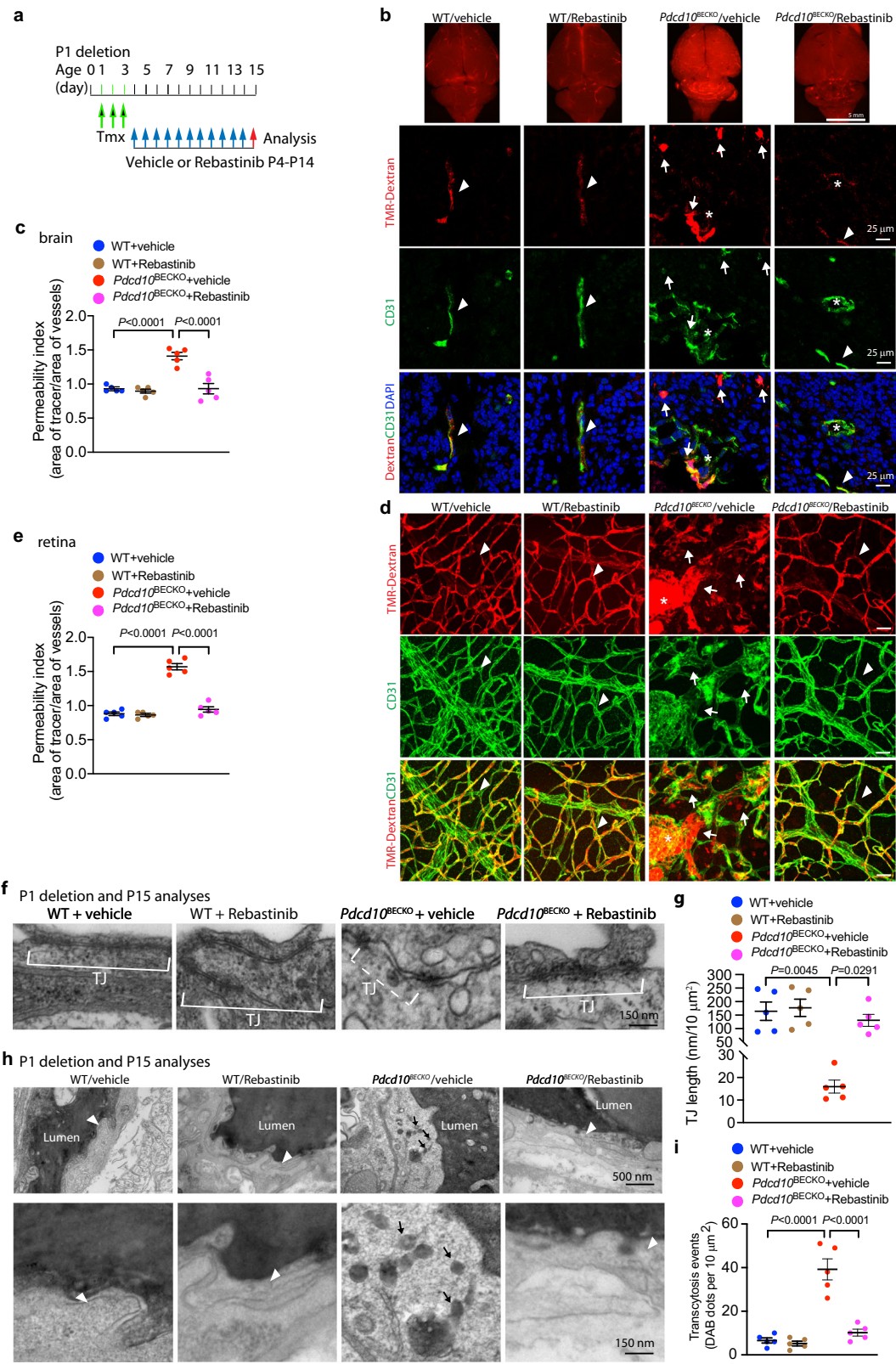

overnight at 4 °C (Supplementary Table 1). Retinas were then washed five times with PBS and mounted by making four incisions in the fluorescent mounting medium. Pictures were taken with the same exposure and gain settings using a confocal Zeiss Airyscan 880 microscope (Zeiss, Germany). Vascular areas were outlined using NIH Image J software and quantified as the percentage of the total area of the retina analyzed.

For immunofluorescence, the secondary antibodies were produced in either donkey or goat and targeted against the appropriate species, with conjugation with

AlexaFlour 488, 555, or 647 (1:400; Invitrogen) or with FITC, Cy3, or Cy5 (1:400; Jackson Laboratories).

**Immunoblotting**. Tissue or cells were lysed in 2x Laemmli buffer and boiled for 5 min at 100 °C followed by centrifugation at $20,000 \times g$ for 15 min at 4 °C. The protein extracts were subjected to standard Western blot analysis which was performed. The following antibodies were used for western blotting: Rabbit polyclonal

**Fig. 8 Tie2 inhibition normalized EC barrier function in CCM mouse models. a** A diagram for the protocols. Vehicle or Rebastinib was subcutaneously injected into Ctrl or P1 deletion *Pdcd10*BECKO mice, and P15 mice were subjected to TMR-dextran perfusion (**b–e**) or transcytosis assays (**f, g**). **b–e** TMR-dextran perfusion was performed at P15 by retro-orbital injection. Local brain and retinal vascular leakages were indicated by accumulation of TMR-dextran dye outside the vasculatures. **b** Whole-brain tissue images (top) and frozen sections co-stained with CD31 (lower 3 panels). Representative images of normal or normalized microvessels (arrowheads) and leaky vessels (arrows) within or near CCM lesions (asterisks) are shown. **c** Quantifications of Permeability index (area of tracer/area of vessels). $n = 5$. **d** Whole-mount staining of retinas with CD31. Representative images of normal or normalized microvessels (arrowheads) and leaky vessels (arrows) within or near CCM lesions (asterisks) are shown. **e** Quantifications of Permeability index (area of tracer/area of vessels). $n = 5$. **f, g** Cerebral sections from WT and *Pdcd10*BECKO mice (P1 deletion/P15 harvest) were subjected to EM. **f** Representative EM images from 5 mice are shown. Tight junction (TJ) was indicated by bracket in WT and dashed bracket in *Pdcd10*BECKO. **g** TJ length was quantified per 10 mm². 20 EM section per mouse and $n = 5$. **h, i** Brain EC transcytosis assay. P15 mice were subjected to HRP perfusion and DAB reaction assay followed by EM imaging of microvessels. Representative images of normal or normalized microvascular ECs (arrowheads) and leaky ECs with increased DAB bots (arrows) are shown. **i** Transcytosis events (DAB dots per 10 mm²) were quantified. $n = 5$. Data are means ± SEM. *P* values are indicated, one-way ANOVA followed by Tukey's multiple comparisons test (**c, e, g, i**). Scale bars: 5 mm (top panel in (**b**)); 25 µm (2nd–4th panels in (**b**) and (**d**)); 500 nm (top panel in (**h**)); and 150 nm (**f**; bottom panel in (**h**)). Source data are provided as a Source data file.

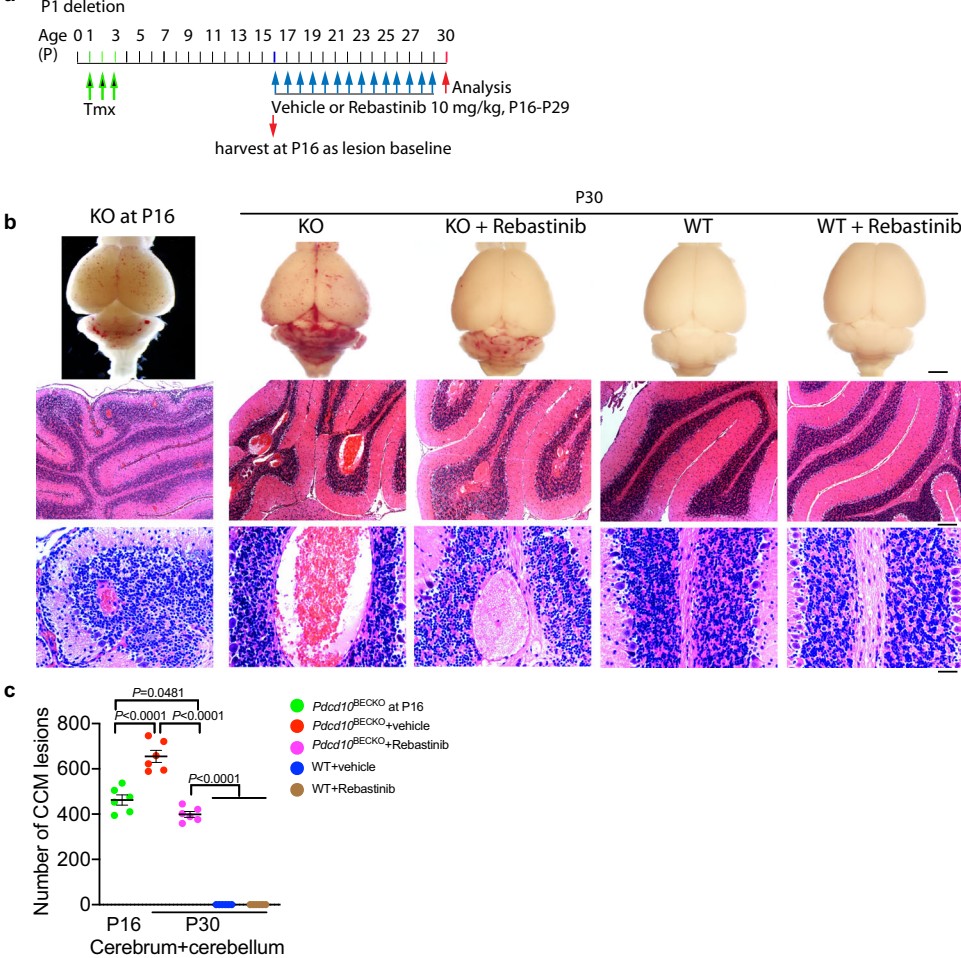

**Fig. 9 Tie2 inhibition prevents progression of pre-existing lesions. a** A diagram for the protocol. WT and P1 deletion *Pdcd10*BECKO mice were subcutaneously administrated with vehicle or Rebastinib daily at P16–P29 and brain tissues were harvested at P30. P16 KO mice were harvested as lesion baselines. **b** Images of fresh brain tissue and H&E staining of brain sections. **c** The numbers of total lesions were quantified as # of lesions per 10 coronal sections, which were 200 µm apart. $n = 6$ mice per group. *P* values are indicated, one-way ANOVA followed by Sidak's multiple comparisons test. Data are means ± SEM. Scale bars: 1 mm (**b** top panel); 100 µm (**b** middle panel); 25 µm (**b** bottom panel). Source data are provided as a Source data file.

antibody against CCM3 was generated (Invitrogen) against full-length recombinant human CCM3 protein expressed and purified from *Escherichia coli*. β-actin (mouse, A1978) and β-actin (mouse, A5441) were from Sigma; p-Caveolin-1 (rabbit, 611339) was from BD Pharmingen; Abl (rabbit, 2862s), p-Akt (rabbit, 9271), Akt (rabbit, 9272), p-MLC2 (rabbit, 3674), PDGFR-β (rabbit, 3168), p-Tie2 (rabbit, 4221), p-Tie2 (rabbit, 4226), Tie2 (rabbit, 4224), and VEGFR2 (rabbit, 2479) were from Cell Signaling Technology. Caveolin-1 (rabbit, sc894) was from Santa Cruz. Angpt2 (rabbit, ab8452), p-Abl (rabbit, ab4717), and N-Cadherin (rabbit, ab76057) were from Abcam; Angpt2 (AF7186) was from R&D Systems. All

primary antibodies were diluted 1:1000. For data presented in the same figure panel, the samples were derived from the same experiment and that gels/blots were processed in parallel. Uncropped blots and gel images were provided in the Source data file.

**Gene expression**. Total RNA was isolated from tissues or cells using the RNeasy kit with DNase I digestion (Qiagen, Valencia, CA). Reverse transcription was performed using standard procedures (Super Script first-strand synthesis system;

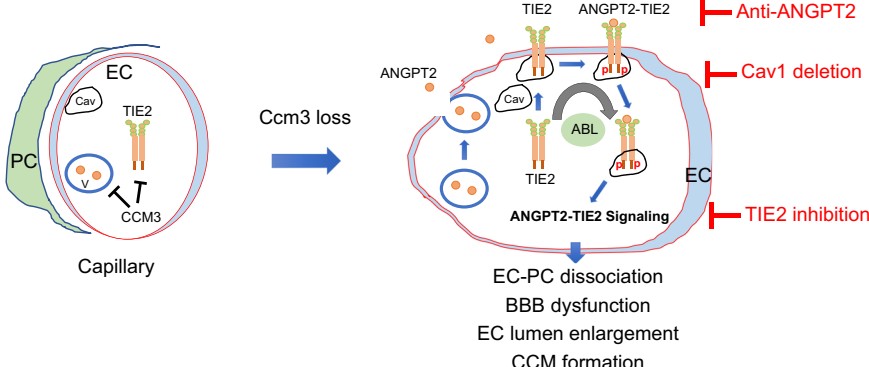

**Fig. 10 A model for Caveolae-mediated Tie2 signaling in CCM progression.** CCM3 in brain ECs normally suppresses caveolae by yet unknown mechanism. CCM3 loss in ECs causes enhanced caveolae-mediated Tie2 surface expression and endocytosis, leading to EC junctional disruption, EC-pericyte dissociation, BBB dysfunction, and lumen enlargement. Non-receptor kinase Abl cooperatively regulates Tie2 expression and activity. Cav1 deficiency or Tie2 inhibition, like neutralization of Angpt2 ligand, ameliorates CCM lesion progression in the mouse CCM models, with normalization of EC-pericyte interaction and vascular integrity.

Qiagen) with 1 μg of total RNA. Quantitative real-time polymerase chain reaction (qPCR) was performed using iQ SYBR Green Supermix on an iCycler real-time detection system (Bio-Rad Laboratories, Inc., Hercules, CA). qRT-PCR with specific primers (e.g., CCM3, Unc13b, and Tie2) was performed (Supplementary Table 2).

**Cell culture and growth factors**. HBMVECs (cAP0002) were purchased from Angio-Proteomie (Boston). ECs were grown in Microvascular Endothelial Cell Growth Medium-2 MV (EGM2, Lonza).

**Angpt2 ELISA**. To measure Angpt2 in human EC supernatants, cells were treated and harvested at the indicated times. Enzyme-linked immunosorbent assay (ELISA) was performed with a standard sandwich protocol using anti-Angpt2 antibodies (R&D Systems, Minneapolis, MN), including recombinant human Angpt2 (625-AN-025), human Angpt2 Mab (MAB098), and human Angpt2 biotinylated Mab (Bam0981).

**Clinical specimens**. CCM clinical specimen were obtained from Pathology Tissue array services (https://medicine.yale.edu/pathology/ypts/tissuemicroarrayfacility/) where they hold archives of human paraffin tissues for research. For the tissue arrays, ethical evaluation for the clinical specimens was reviewed and approved by the Yale Human Investigation Committee (IHC# 0601000969). Written informed consent was provided by all subjects before the tissue collection and the tissue collection was performed in compliance with all ethical regulations. The Section of Neuropathology, Department of Pathology at the Yale University School of Medicine archives human specimens, including cerebral sections, from individuals with PDCD10 mutations from patients or family members giving informed consent. All of the samples were evaluated independently by two pathologists, who were experienced in evaluating immunohistochemistry and had no prior information regarding the clinical outcome of the patients. All investigators used the tissue arrays are requested to agree to abide by (a) the Yale University Federal wide Assurance (FWA) and the specific terms of the Yale University FWA; (b) the relevant Yale University policies and procedures for the protection of human research participants, and (c) HIPAA at Yale, Researcher's Guide to HIPAA. We obtained paraffin blocks for eight cases of cerebral sections from individuals with PDCD10 mutations and serial 5-μm-thick sections (~20) were cut from each block for H&E staining and immunostaining. According to H&E staining, we found that six samples contained typical CCM lesions with normal surrounding tissues. Two cases showed severe vascular fibrosis and were not used for further analyses[27].

**Statistics and reproducibility**. Group sizes were determined by an a priori power analysis for a two-sided, two-sample $t$-test with an α of 0.05 and power of 0.8 to detect a 10% difference in lesion size at the endpoint. Animals were randomized into groups, and investigators were blinded to the groups during the experiments. Male and female animals were used in equal numbers for all experiments. No samples or animals were excluded from analysis. All quantifications (lesion sizes, junctional integrity, sprout length, and lumen) were blinded. All figures are representative of at least three experiments unless otherwise noted. All graphs report mean ± standard error of mean (SEM) of the biological replicates. Comparisons between two groups were performed by unpaired, two-sided $t$-tests, while comparisons between more than two groups were performed by one-way ANOVA followed by Bonferroni's post hoc test or by two-way ANOVA using Prism 6.0 and 8.0 software (GraphPad). All statistical tests used to examine statistical significance were two-sided, confidence interval: 95%. Exact $p$-values and the respective test/

analysis are listed in the figure and legends. $P$ values <0.05 were considered significant.

**Reporting summary**. Further information on research design is available in the Nature Research Reporting Summary linked to this article.

## Data availability
All data, including data associated with main figures and supplementary figures, are available within the article or in the online-only Data Supplement or from the corresponding author on request. Source data are provided with this paper.

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

## Acknowledgements

This work was partly supported by NIH grants HL136507, HL157019, and GM109487. H.J.Z. is supported by AHA career development award (19CDA34760284). K.N.M. is supported by AHA postdoctoral fellowship (17POST33670779). We thank Dr. Bin Zhou provided the Mdsf2aCre mice; we appreciate Al Mennone's help in acquiring images at the Yale University Center for Cellular and Molecular Imaging (CCMI); we appreciate Morven Graham's effort in acquiring tomography images at the Yale University CCMI Electron Microscopy Core Facility.

## Author contributions

H.J.Z. and W.M. provided conceptual of the study and designed all the experiments; H.J.Z. executed and organized team members; H.Z. for mouse breeding, genotyping, in vivo western blotting and half of the in vitro studies; H.J.Z., L.Q., B.L., and Q.L. for in vivo sample staining; Q.J. for half of the in vitro study; H.J.Z. for organotypic angiogenesis assays; K.N.M. collected the two-photon images guided by J.G.; M.G. performed EM guided by X.L.; H.J.Z. and W.M. analyzed the data, wrote the manuscript, and provided funding sources.

## Competing interests

The authors declare no competing interests.
