## [Peer Review File · Nature Communications]

REVIEWER COMMENTS

Reviewer #1 (Remarks to the Author):

This manuscript reports several novel discoveries related to the pathogenesis of Cerebral cavernous malformations.

1. First is their new model of disease, using a brain-specific endothelial cell Cre driver to delete the Ccm3 gene. These animals live well past the former pre-weaning bottleneck and exhibit lesions in the cerebrum that are large, and show evidence of chronic hemorrhage. These are much more clinically relevant than the smaller and less hemorrhagic CCMs observed in the cerebellum in other mouse models of the disease. This itself is a major advance.

2. A second novel discovery is the connection between Ccm3 gene/protein loss and increased caveolae vesicles, and with increased CAV1-Tie2 signaling. This data supporting this connection form a large part of the manuscript and figures.

3. Given activation of tie2 as per the discovery discussed above, the authors show that Rebastinib, a selective inhibitor of tie-2, has a profound effect on reducing CCM lesions if given at the same time as lesion genesis begins (deletion of the Ccm3 gene). (please note that the drug is misspelled in one place on page 12).

4. The use of transcranial two-photon microscopy shows that CCM lesions bulge from capillaries or post-capillary venues and pericytes become dissociated from the vessels.

5. The importance of the endothelial-mesenchymal transition (EndMT) in CCM pathogenesis has published evidence for and against. In this manuscript, using stimulated emission depletion (STED) microscopy, they show that alpha-smooth muscle actin expression occurs in vascular malformation associated (and aberrant) pericytes and not endothelial cells. The data looks relatively compelling, at least in this particular gene (Ccm3) mouse model of disease. This suggests that some of the data in support of an EndMT in CCM pathogenesis may not occur as such. Instead these changes may be due to pericytes that dissociate from ECs during CCM formation.

All of these are fundamental discoveries and elevate this paper to a high level.

The huge question, and one that has yet to be solved for any drug, is whether Rebastinib treatment given late in life can reverse / shrink pre-existing CCMs? I would not however insist that this experiment be done, given the current lockdown most universities are experiencing.

Questions:

1. Please discuss the link between the STRIPAK signaling complex and caveolae.
2. Since their data suggests that CCM3 regulates CAV1-Tie2 independently from UNC13B-mediated exocytosis and ANGPT2 secretion, please discuss how CCM3 loss separately effects these independent effects.
3. In light of the above, would a combination therapy of anti-ANGPT2 neutralizing antibody and Rebastinib have a more profound effect on CCM pathogenesis?
4. These newly discovered pathways appear to be unique to CCM3 pathogenesis, and possibly have no relevance to the other inherited forms, nor possibly to "sporadic" CCMs. The authors should discuss this limitation, or if it is not a limitation, discuss why not.
5. Venous malformations are caused by specific somatic and germline mutations in tie-2 that have been shown to activate downstream signaling. In light of the current data on tie-2 signaling in CCM pathogenesis, the venous malformation studies would be quite informative to include in the Discussion, comparing and contrasting both the vascular anatomy and also the signaling.

The Discussion requires some drastic editing for style and sentence structure. On the bottom of page 14 deficient is misspelled.

Reviewer #2 (Remarks to the Author):

Revision report on the manuscript NCOMMS-20-16750-T.

The current manuscript describes a novel genetic mice model for studying cerebral cavernous malformations (CCMs), which are vascular abnormalities causing cerebral hemorrhage, stroke and seizures in adult humans. CCMs have been previously shown to be associated with loss-of-function mutations in one of the three *Ccm* genes. Here, the authors presented a novel mice model (i.e. *Pdcd10BECKO*), in which the inducible *Ccm3* gene deletion occurs specifically in the brain and retinal endothelial cells. The *Pdcd10BECKO* mice, in contrast to the previously reported global *Ccm3* knock-out mice, survive to adulthood and resemble human CCM pathogenesis. Moreover, unlike the global *ccm3* KO mice, in the newly described *Pdcd10BECKO* mice the CCM lesions occur in the cerebrum. Thus, the authors used this phenotype to their advantage and succeeded in visualizing and characterizing the vascular dynamics of the CCM lesions employing transcranial live imaging. This approach revealed that CCMs lesions are initiated mainly at the capillary/post-capillary venules and exhibit pericytes dissociation.

In their earlier work, the authors have demonstrated that high levels of *Angpt2*, resulting from the global *ccm3* deletion, promote CCM pathogenesis in the brain. However, the role of the *Angpt2* receptor *Tie2* in the CCM pathogenesis was not clear thus far. In the current manuscript, the authors extended their analysis on the receptor *Tie2* and its' role in the CCM pathogenesis. They have found that *Tie2* protein is upregulated upon the induced *ccm3* deletion in the brain and retinal endothelial cells. Furthermore, the authors have demonstrated by employing pharmacological and genetic tools that *Tie2* upregulation is critical for the development of vascular malformations and is mediated via the caveolae-dependent mechanism.

The establishment of the novel *Pdcd10BECKO* mice model, presented in this manuscript, definitely advances the field and allows further investigations of the CCM pathogenesis and would be of interest to others in the vascular biology community. That being said, the data presented here is not sufficient to support all conclusions the authors have made.

Major concerns:

First, the authors demonstrate that during CCM lesion formation, pericytes gain elevated expression levels of alpha-SMA and dissociate from the endothelial cells. However, the deficiency of pericytes at the CCM sites has been shown already by others (Schulz, Wieland et al. 2015). Furthermore, in their previous work (Jenny Zhou, Qin et al. 2016), authors already have shown similar pericytic dissociation in CCM lesions in the brain. The only novelty of this work is that there's an increase in alpha-SMA protein expression in the pericytes located in the CCM lesions of the mutant mice. The authors suggest that the increase in the alpha-SMA expression may contribute to pericytes dissociation from the endothelial cells; however, the data provided here is not sufficient to support this suggestion. It is not clear from the main text how the upregulation of alpha-SMA in the pericytes may contribute to the CCM progression. Are there any other studies supporting that idea, published examples in the field? If yes, the authors should discuss this. If not, authors should provide data, which would demonstrate that downregulation of alpha-SMA in pericytes can attenuate lesion progression.

Second, the authors demonstrate elevated levels of both total *Tie2* and phospho-*Tie2* protein in the brain and retinal endothelial cells in the *Pdcd10BECKO* mice. They suggest that *ccm3* deletion enhances the activation of the *Tie2* receptor, which in turn, promotes vascular malformation. The quantification analyses provided here are not sufficient to support higher activity state of the *Tie2*, because high levels of the phospho-*Tie2* might be a result of high levels of the total *Tie2* expression (see specific comment 10.1).

In general, in the present work, authors demonstrate that *ccm3* deletion in the brain and retinal

vasculature affects both endothelial cells and pericytes. However, it is not clear from the manuscript how and if the changes in the molecular signature of both pericytes and endothelia are connected? Is there any connection between the pericytic dissociation and the upregulation of caveolae-Tie2 signaling that drives the CCM lesions? Authors could test whether the alpha-SMA signature of the pericytes at the lesion sites is rescued in the rescue experiments (e.g. Cav1 KO, Rebastinib treatment).

Third, some data provided in this manuscript demonstrate alterations in the expression of several molecular factors in the endothelial cells (i.e., VE-Cad, Claudin-5) and pericytes (i.e., alpha-SMA, NG2) in the CCM lesions of the Pdc10BECKO mutant animals. However, the authors based their conclusions mainly on the fluorescent immunostainings. Have the authors performed any other quantification analyses to support what they show in the IF images? WB, RT-PCR analyses, perhaps?

Also, the authors show that during CCM lesion formation the endothelial cells become leaky. What might cause the vascular hyper-permeability? Is it a result of disrupted tight junctions? Or due to increased transcytosis (aka increased caveolae vesicles). Or both? Authors could address this question by performing HRP injections into the mutant mice followed by EM ultrastructural analysis.

Finally, the authors propose that "caveolae mediates Tie2 intracellular trafficking and recycling, leading to hyper-activation of Tie2". However, while the levels of Tie2 are increased in the endothelial cells of the mutant mice, the data presented in this manuscript is not sufficient to prove that Tie2 receptor is indeed being recycled to the cell surface. In the figure 5f, authors show that both surface and endocytic Tie2 are elevated in the ccm3 knockdown cells. An alternative interpretation of the presented data might be is that Tie2 receptor is internalized via the caveolae-mediated endocytosis and translocated into the nucleus for further activation of the downstream signaling, rather than being recycled. Could the authors test whether the Tie2 receptor in the mutant mice colocalizes with the nuclei of the endothelial cells, perhaps by using the super resolution microscopy? Also, it would be interesting to test whether depletion of Cav1 results in more degradation of Tie2 protein, based on the clathrin-mediated Tie2 degradation mentioned in the main text (page 15, para 1). Authors could test this by examining the localization of Tie2 to late endosomes/lysosomes in the Cav1 depleted cells.

Specific comments:

1. P6, para 1: Authors claim that "splenectomy prolonged Pdc10BECKO mouse survival..." is that a new data? If yes, it should be shown here. If it was published, citation needed.
2. Could the authors provide a proof for successful Ccm3 depletion by Western or some other means?
3. Fig.1 C and K: The number of lesions is increased over time in both cerebrum and cerebellum, however, according to the presented data, the effect was more substantial in the cortex in the Pdc10BECKO mice. The authors don't relate to this at all in the text.
Also, according to the data presented here, in the cerebellum, there are larger lesions compared to the cerebrum at all time points tested. In the cerebrum, the majority of the lesions are smaller, and only at 6 months old mice, the size of most of the lesions is increased. Is there any physiological significance to the lesion size itself? Why the cerebrum vessels exhibit smaller lesion sizes in the earlier time points?
4. Supplementary Fig. 3: Is it possible to show that also in the adult stage (P60) the malformations remain specific to the veins by staining against endomucin?
5. Page 7, para 2 and Fig. 2: The authors claim there are no lesions when the ccm3 deletion is induced on day P21, however, they don't provide the data.
6. Page 7, Para 3: It should be explained in the main text why mG expressed mainly in the veins and capillaries but not in the arteries, perhaps referring to the Mfsd2a differentiated expression shown in previous studies.
7. Page 7, Para 3 last sentence: the authors suggest that "...as well as evident disruption of EC junctions" – however, they don't provide sufficient data, specific junctional staining, but only based on the mG expression.
8. Fig. 2g-j: In general, it is not clear from the main text or the Methods section how the

calculations were made. In particular, the quantification of perfused vs non perfused is not clear. Isn't the perfusion in WT vs the moderate should be mirrored? Why not all vessels in the WT are perfused? It is not clear from the text. Should be elaborated.

9. Fig. 3:

8.1 3a-c: The authors suggest that in the mutants, the pericytic processes are shortened. This is very clear when comparing the images of P5 del (moderate phenotype) vs WT. However, it's not that clear when comparing the P10 del (mild phenotype) with WT. Is the image of the P10 del is representative? Also, authors do not provide quantification for this statement, but instead refer in the main text to the brackets shown in the figure. However, in the P10 del (mild), shown in the middle column, the bottom panel in Fig. 3a, the brackets connect two different pericytes.

Furthermore, according to the image presented in Fig. 3a middle column, bottom panel - it seems that the length of the processes in the P10 is not actually shortened. In this image the processes of the detached pericyte looks comparable in the terms of length to the ones seen in the WT.

8.2 Furthermore, it is not clear why authors suggest "...compensatory effect for shortened processes..." (page 8, para 1) when referring to the increased density of pericytes in the mutant animals. They should elaborate on this statement.

8.3 Also, it is not well described neither in the main text or the Methods section, how the calculations made in the graphs 3b,c. Is it the number of caverns and pericytes per endothelial mm³?

8.4 3d-e: Authors say that the images are taken from animals age of P15, however, they haven't provided the exact experimental procedure paradigm they used in the experiment described in fig.3 – deletion at P1, P5, P10? They perhaps should explain why they chose P15 stage in these experiments.

8.5 Is the total measured length/area of the endothelia in the WT vs mutants equivalent?

8.6 3f, g and supplemental fig.5: Again, authors haven't provided the exact experimental procedure paradigm they used in the experiments described here – deletion at P5, P10? Is there an explanation why they have decided to image the tissues from pups at developmental stage P15? From the representative images in these figures, it is clear that alpha-SMA signal is increased in the CCMs lesions of the mutant animals. Have the authors performed any quantificational analysis to support the conclusion made from the provided images? Or perhaps performed WB or RT-PCR analyses apart from fluorescent immunostaining?

8.7 On the one hand, the authors claim that there's less pericytic coverage in the caverns (live imaging), on the other hand, here they focus on the caverns to show that pericytes are there (by showing NG2 expression). Is it an early preliminary (or intermediate) stage prior to the pericytic dissociation? Or perhaps at the P15 stage, the pericytes are still there and are dissociated later in adulthood? The authors should discuss this. Again, the age of the animals in the fig. 3a-c is not provided.

10. Supplemental Fig. 6: are all the TJs were weakened in the mutants? If not all, what is the percentage of the weakened TJs in the mutants? Again, what is the age of the animals? Is this following mild or severe ccm3 deletion?

11. Fig. 4f-j:

10.1 The authors demonstrate that the levels of the total and phospho- Tie2 receptor are increased in the mutant mice and in the samples of CCM3 lesions from humans. The upregulation of phospho-Tie2 might be due to the total upregulation of the Tie2 expression. Have the authors tested in a quantifiable manner whether the deletion of the ccm3 induced Tie2 receptor phosphorylation? For instance, measurement of the Phospho-Tie2 levels normalized by the total Tie2 protein (WB or IF).

10.2 Furthermore, in the supplemental fig. 8a authors demonstrate that the mRNA levels of the Tie2 are not altered following ccm3 knockdown. How do the authors explain the elevated levels of the Tie2 protein (supp. Fig. 8C) but not its mRNA?

12. Fig. 5a: The authors show co-localized expression of Cav1 and Tie2 by IF. Perhaps, a control with siRNA knockdown of Cav1 and Tie2 and IF on these cells would support the specificity of this staining.

13. Fig. 6f, h, k – The calculation for the pericytic coverage is not clear. Is the NG+ signal normalized to the EC signal? Have the authors quantified the NG+ along the same endothelial

length/area? In the images provided, it looks like there's more CD31+ signal in the Pcdcd10BECKO (due to the large lesion size) compared to the rest groups.

Minor comments:

1. Fig. 1f-g: Indicate in the figure legend what do the arrows stand for? Consider to include in the figure what for each of the dyes stand, similarly to the Collagen column. It would be easier for the audience that is not familiar with these dyes to follow.

2. Fig. 2:

2a: "P1 deltion" -> P1 deletion

2c: Not indicated in the figure legend what do the arrows stand for.

3. Pages 7-8, referral to the Nissl dye: The reference is not accurate for the labelling of pericytes by Nissl dye. The indicated reference doesn't include such a methodology. Authors should include the right citation: Damisah at al., Nature neurosci., 2017.

4. Fig. 3f, g: The changes in the colors of the CD31 and NG2 signals along the figure makes it difficult to follow. For instance, in Fig.3 f CD31 and NG2 are red and then on the right column NG2 appears as grey. In the Fig.3g it's the same for the CD31 and NG2 and then in the last column, CD31 is changed to blue. It would be helpful if each channel was differentially pseudo-colored and kept consistent along the figure.

5. Fig. 4e-h: Similarly, to my previous comment, it would be helpful if the Cav1 signal would be indicated by the same color along the figure, just as in fig.4i-j.

6. Page 9, para 2: "...We obtained paraffin blocks for 8 cases...", perhaps should be blocks of 8 cases.

7. Please, keep consistent either p-Tie2 or phospho-Tie2.

8. Page 11, para 3, "...intraperitoneally (i.p.) into Pcd10ECKO..." – Should be Pcdcd10BECKO.

9. Page 11, bottom: "To this end, we injected...subcutaneously injected..." – injected appear twice.

10. Supplemental Fig. 8b: check the figure legend. Instead of Cav1 should be UNC13B.

11. Supplemental Fig. 12a,b: "P1 deltion" -> P1 deletion

12. Page 14, para 2, the bottom of the page: "...Genetic rescue study with Cav1-deficient..." - Should "deficient".

General minor comments:

1. I found the paper reasonably easy to follow, but I think a non-specialist audience will struggle, as many of the terms are not defined. Some re-writing to address this – and some of the minor grammatical errors, typos and complex sentence construction throughout might be useful.

2. In general, authors should better explain elaborate more how they've performed image quantification analyses. For instance, for the EM analyses how many slices per each mouse? The explanation appears in the fig.4a legend, but not in the supplemental fig. 6.

3. "In vivo", "in vitro" in italics please.

References:

Jenny Zhou, H., L. Qin, H. Zhang, W. Tang, W. Ji, Y. He, X. Liang, Z. Wang, Q. Yuan, A. Vortmeyer, D. Toomre, G. Fuh, M. Yan, M. S. Kluger, D. Wu and W. Min (2016). "Endothelial exocytosis of angiopoietin-2 resulting from CCM3 deficiency contributes to cerebral cavernous malformation." Nat Med 22(9): 1033-1042.

Schulz, G. B., E. Wieland, J. Wüstehube-Lausch, G. Boulday, I. Moll, E. Tournier-Lasserre and A. Fischer (2015). "Cerebral Cavernous Malformation-1 Protein Controls DLL4-Notch3 Signaling Between the Endothelium and Pericytes." Stroke 46(5): 1337-1343.

Reviewer #3 (Remarks to the Author):

Zhou et al. describe a novel mouse model to study cerebral cavernous malformations (CCM),

where the *Ccm3* (*Pdcd10*) is deleted specifically in the brain and retinal vasculature using *Mfsd2a*-*CreERT2*. The authors observed increased caveolae and increased phospho-Tie2 in the lesions of *Pdcd10*BECKO mice. *Cav1*-deficiency and Tie2 inhibition in *Pdcd10*BECKO mice were able to reduce the CCM lesions. The authors also used two-photon microscopy to visualize the vasculature during CCM lesion progression.

Although a specific deletion of *Pdcd10* in the brain vasculature using the *Slco1c1*(BAC)-*CreERT2* was previously explored by Tang et al. (*Science Translational Medicine*, 2019), the disease phenotype was not characterized as carefully as in this manuscript by Zhou et al.

Notably, using two-photon, confocal and stimulated emission depletion (STED) super resolution microscopy, the authors provide high-resolution visualization of the vasculature and EC-pericyte interactions during CCM lesion progression. They show that CCM lesion was generated by initially bulging from capillary/post-capillary venule with a single dilated blinding end (cavern) that rapidly grew into disorganized venule beds with disrupted microvascular network.

The authors provide evidence for an intriguing model that increased caveolae vesicles in the brain microvascular ECs, and the increased caveolae augmented Tie2 receptor signaling contributed to CCM lesion progression. In summary, this is a very conclusive study, however, with few issues deserving additional consideration.

Major comments:

1. In contrast to *Cav1* genetically deficient mice, the role of Tie2 is demonstrated using rebastinib, a small molecular kinase inhibitor of Abl kinases, SRC, KDR, FLT3, and Tie-2. The reported IC50 values suggest that whereas the inhibitor has higher activity towards various Abl forms, it inhibits Tie2 and KDR at similar efficacies. The authors need to comment on the specificity of rebastinib, and the potential effects it may have via inhibition of Abl and Kdr, which contribute to vascular leakage and growth. The authors have also limited the analysis of the Tie2 pathway to analysis of phospho-Tie2, with no analysis of downstream signalling or gene expression changes.
2. Are the CCM lesions leaky in the *Pdcd10*BECKO mice? The imaging of EBD indicated that most caverns were perfused, but some of the newly formed caverns were non-perfused (Fig.2e with quantifications in 2h-i). Additionally, leakage was not observed (Figs 1-2). However, it is reported that *Cdh5* and *Cln5* levels are decreased, and that evident disruption of EC junctions in moderate lesions was observed (Fig.2f with quantification in 2j).

Minor comments:

3. Interestingly, *Pdcd10* deletion does not result in lesion formation when induced at p21. What makes the vessels different before and after this time point? Was a similar time dependence observed in the retina? As shown before in the retina, *Angpt2* expression seizes in the superficial vascular plexus after a short postnatal period (Hackett et al., 2000, Elamaa et al 2020). After this occurs, VEGF expression alone is not sufficient to cause angiogenic growth, unless *Angpt2* is ectopically expressed (Oshima, et al. 2004). Could the authors comment the expression of *Angpt2* in the postnatal brain vasculature and on the potential interplay of the *Angpt2* and VEGF pathways in CCM formation.
4. Are the non-perfused areas hypoxic? Regarding the potential role of Tie2, did the authors study if rebastinib increased perfusion in the lesions?
5. The authors should consider a recent publication of an in vitro vascular malformation model of activated Tie2 (Yuqi Cai et al., Constitutive Active Mutant TIE2 Induces Enlarged Vascular Lumen Formation with Loss of Apico-basal Polarity and Pericyte Recruitment, *Scientific Reports*, 2019).
6. The mechanism behind increased Tie2 activation remains open. The authors speculate "Therefore, it is conceivable that combined increases of ANGPT2 and Tie2 lead to highly activated Tie2 that could drive vascular malformation." Indeed, the activity of *Angpt2* is highly context-dependent. Ectopic expression of *Angpt2*, resulting in high levels of *Angpt2*, did not cause visible

changes in the brain vasculature (e.g. Li et al. JCI, 2020), although it induced the enlargement of post-capillary venules and capillaries in the trachea (Korhonen et al., Kim et al., JCI, 2016) in otherwise healthy mice. In the tracheal vascular bed, the activity of Angpt2 was dependent on Tie1. In the lymphatic vasculature, Angpt2 agonist activity has been attributed to low level of VE-PTP expression in the lymphatic compartment (Souma et al., PNAS, 2018). Have the authors explored Tie1 or VE-PTP expression in the postnatal brain/retina/CCM lesions?

7. Authors show that rebastinib hinders the disease formation when administered after tamoxifen. From a translational point of view, it would be of interest if Rebastinib can normalize the vasculature in already formed lesions.

8. Did the authors follow whether *Pdcd10*BECKO:*Cav1*^{-/-} (DKO; tamoxifen P1-3) prevented the development of lesions also when analysed at much later time points, e.g. at P60, or was the Caveolin-1 deficiency only delaying the disease progression?

9. In page 6, row 28: authors should remove 3 from sentence "...developed CCM3 lesions with dilated capillaries..."

10. In page 6, row 31: authors talk about body weight loss, however as the deletion is done postnatally before the mice have reached their adult weight, the correct term would be weight gain.

11. In page 6, row 34: the sentence "Venous endothelium and capillaries marker endomucin staining shown venous malformations at the periphery of the retinal vascular plexus" is unclear and should be rephrased.

12. In the figure legend of Supplementary Fig.4. *Pdcd10*ibbbKO should be corrected to *Pdcd10*BECKO

13. In Figure 2 in panel a P1 deltion should be corrected to P1 deletion

14. In Figure 5a authors should provide an image with more cells to make conclusions about the increased colocalization of TIE2 and Cav1.

15. In Figure 5b quantification of the western blot should be provided.

16. In figure 5 letters m and n indicating panels, are missing.

17. In page 12, row 4: Rabastinib should be corrected to Rebastinib.

18. Please correct ref 19. Detter, M.A., Snellings, D.A. & Marchuk, D.A. Cerebral Cavernous Malformations Develop Through Clonal Expansion of Mutant Endothelial Cells. *Cir Res* In press(2018).

RESPONSE TO REVIEWER COMMENTS

Response to Reviewer #1

This manuscript reports several novel discoveries related to the pathogenesis of Cerebral cavernous malformations.

1. First is their new model of disease, using a brain-specific endothelial cell Cre driver to delete the *Ccm3* gene. These animals live well past the former pre-weaning bottleneck and exhibit lesions in the cerebrum that are large, and show evidence of chronic hemorrhage. These are much more clinically relevant than the smaller and less hemorrhagic CCMs observed in the cerebellum in other mouse models of the disease. This itself is a major advance.
2. A second novel discovery is the connection between *Ccm3* gene/protein loss and increased caveolae vesicles, and with increased CAV1-Tie2 signaling. This data supporting this connection form a large part of the manuscript and figures.
3. Given activation of *tie2* as per the discovery discussed above, the authors show that Rebastinib, a selective inhibitor of *tie-2*, has a profound effect on reducing CCM lesions if given at the same time as lesion genesis begins (deletion of the *Ccm3* gene). (please note that the drug is misspelled in one place on page 12).
4. The use of transcranial two-photon microscopy shows that CCM lesions bulge from capillaries or post-capillary venues and pericytes become dissociated from the vessels.
5. The importance of the endothelial-mesenchymal transition (EndMT) in CCM pathogenesis has published evidence for and against. In this manuscript, using stimulated emission depletion (STED) microscopy, they show that alpha-smooth muscle actin expression occurs in vascular malformation associated (and aberrant) pericytes and not endothelial cells. The data looks relatively compelling, at least in this particular gene (*Ccm3*) mouse model of disease. This suggests that some of the data in support of an EndMT in CCM pathogenesis may not occur as such. Instead these changes may be due to pericytes that dissociate from ECs during CCM formation.

All of these are fundamental discoveries and elevate this paper to a high level.
- We highly appreciate your positive comments. We also thank you for your instructive suggestions. We have corrected the misspelling and we have addressed your concerns as follows.

The huge question, and one that has yet to be solved for any drug, is whether Rebastinib treatment given late in life can reverse / shrink pre-existing CCMs? I would not however insist that this experiment be done, given the current lockdown most universities are experiencing.
- Thank you for your great suggestions. We have tested the therapeutic effects of Rebastinib on pre-existing CCMs in a severe model (*Ccm3* deletion at P1 followed by administration of the inhibitor at P15 when the CCM lesion fully bloomed). Results showed that Rebastinib could prevent further CCM lesion progression and even shrink the pre-existing lesions compared with the vehicle-treated group. However, the inhibitor could not completely reverse the pre-existing lesions. More experiments are needed to test the therapeutic effects in less-progressive models (for example, the moderate and mild CCM lesion models with delayed injection of tamoxifen for *Ccm3* deletion described in our manuscript). We present the data as “Data for Reviewer only” and we feel that this should be incorporated into our future studies (Please see Fig.1 for Reviewer only).

Fig.1 for Reviewer only. Tie2 inhibition prevents progression of pre-existing lesions. a. A diagram for the protocol. WT and P1 deletion *Pdcd10*^{BECKO} mice were subcutaneously administrated with vehicle or Rebastinib daily at P16-P29 and brain tissues were harvested at P30. P16 KO mice were harvested as

lesion baseline. **b.** Images of fresh brain tissue and H&E staining of brain sections. **c.** The numbers of total lesions were quantified as # of lesions per 10 coronal sections, which were 200 μ m apart. $n = 6$ mice per group. *, $P < 0.05$; *** $P < 0.001$ (two-way ANOVA). Scale bars: 1 mm (fresh brain image); 25 μ m (H&E).

Questions:

1. Please discuss the link between the STRIPAK signaling complex and caveolae.

- Thank you for your suggestions. We have added the following paragraph in Discussion (page 18):

Interestingly, more recent studies suggest that CCM3 via the striatin-interacting phosphatase and kinase (STRIPAK) complex regulates Golgi stability and CDC42 or Rab11-dependent receptor endocytic recycling¹. The STRIPAK complex was identified by proteomic analysis which revealed that CCM3 and GCK kinases together with PP2A were present in a striatin family-associated protein complex². Interestingly, striatin family members in STRIPAK complex contain a caveolin scaffolding domain-interaction motif and have been shown to associate with Cav1³; and this association could localize the STRIPAK complex to caveolae to regulate membrane signaling, endocytosis and vesicle trafficking in cells¹. Given the dynamic association of CCM3 with the STRIPAK complex and that caveolae is dramatically increased upon CCM3 deletion, it worth exploring if the STRIPAK complex is enriched in caveolae in CCM3 defective ECs to mediate Tie2 cellular trafficking and signaling.

2, Since their data suggests that CCM3 regulates CAV1-Tie2 independently from UNC13B-mediated exocytosis and ANGPT2 secretion, please discuss how CCM3 loss separately effects these independent effects.

- Thank you for your suggestions. We have added discussion on page 18. *“Our study demonstrates CCM3 regulates Angpt2 and Tie2 through distinct mechanisms. We have previously shown that CCM3 tightly associates with GCK kinase STK24, and through this interaction CCM3 suppresses STK25-Unc13B-mediated exocytosis and Angpt2 secretion. Therefore, CCM3 loss release STK25 to facilitate Unc13-driven secretion of Angpt2 ligand ⁴. Our current study demonstrates that CCM3 deletion enhances the receptor Tie2 in ECs via caveolae-dependent but Unc13B-mediated exocytosis-independent mechanisms. On the other hand, we show that Cav1-caveolae regulates Tie2 signaling via modulating the Tie2 protein stability. How exactly CCM3 loss increases Cav1-caveolae is unclear.....”*

3. In light of the above, would a combination therapy of anti-ANGPT2 neutralizing antibody and Rebastinib have a more profound effect on CCM pathogenesis?

- Thank you for your suggestions. We have tested the combo vs single therapy with injection every day. Unfortunately, we did not see synergistic effects but observed an enhanced early lethality from the combo (Fig.2 for Reviewer only). We suspect that combination of anti-Angpt2 neutralizing antibody and Rebastinib could cause severe defects on vascular development or/and integrity and it needs further investigations. More experiments are needed to test the Combo therapeutic effects at lower doses in less-progressive models (for example, the moderate and mild CCM lesion models with delayed injection of tamoxifen for Ccm3 deletion described in our manuscript). We present the data as “Data for Reviewer only” and we feel that this should be incorporated into our future studies (Please see Fig.2 for Reviewer only). Nevertheless, we have discussed in page 19.

[REDACTED]

[REDACTED]

4. These newly discovered pathways appear to be unique to CCM3 pathogenesis, and possibly have no relevance to the other inherited forms, nor possibly to “sporadic” CCMs. The authors should discuss this limitation, or if it is not a limitation, discuss why not.

- Thank you for your suggestions. We agree with you and we have discussed the limitation of our study as follows (page 19):

Of note, CCM3, but not CCM1 or CCM2, specifically interacts with GCKs and the STRIPAK complex. Similarly, the Angpt2-Tie2 signaling is increased in CCM3-knockdown, but not in CCM1 or CCM2-deficient EC⁴. Therefore, our newly discovered Cav1-Tie2 pathways could be unique to CCM3 pathogenesis. It needs to explore whether or the Cav1-Tie2 pathways are related to the other inherited forms and “sporadic” CCMs.

5. Venous malformations are caused by specific somatic and germline mutations in tie-2 that have been shown to activate downstream signaling. In light of the current data on tie-2 signaling in CCM pathogenesis, the venous malformation studies would be quite informative to include in the Discussion, comparing and contrasting both the vascular anatomy and also the signaling.

- Thank you for your suggestions. We have discussed the differences between Cav1-Tie2 signaling in CCM3 and the Tie2 GOF-induced venous malformation at both anatomy and the signaling (pages 19-20).

6. The Discussion requires some drastic editing for style and sentence structure. On the bottom of page 14 deficient is misspelled.

- Thank you for your suggestions. We have edited and reconstructed the discussion. The changes in the text are marked in blue.

Response to Reviewer #2

The current manuscript describes a novel genetic mice model for studying cerebral cavernous malformations (CCMs), which are vascular abnormalities causing cerebral hemorrhage, stroke and seizures in adult humans. CCMs have been previously shown to be associated with loss-of-function mutations in one of the three Ccm genes. Here, the authors presented a novel mice model (i.e. Pdc10BECKO), in which the inducible Ccm3 gene deletion occurs specifically in the brain and retinal endothelial cells. The Pdc10BECKO mice, in contrast to the previously reported global Ccm3 knock-out mice, survive to adulthood and resemble human CCM pathogenesis. Moreover, unlike the global ccm3 KO mice, in the newly described Pdc10BECKO mice the CCM lesions occur in the cerebrum. Thus, the authors used this phenotype to their advantage and succeeded in visualizing and characterizing the vascular dynamics of the CCM lesions employing transcranial live imaging. This approach revealed that CCMs lesions are initiated mainly at the capillary/post-capillary venules and exhibit pericytes dissociation.

In their earlier work, the authors have demonstrated that high levels of Angpt2, resulting from the global ccm3 deletion, promote CCM pathogenesis in the brain. However, the role of the Angpt2 receptor Tie2 in the CCM pathogenesis was not clear thus far. In the current manuscript, the authors extended their analysis on the receptor Tie2 and its' role in the CCM pathogenesis. They have found that Tie2 protein is upregulated upon the induced ccm3 deletion in the brain and retinal endothelial cells. Furthermore, the authors have demonstrated by employing pharmacological and genetic tools that Tie2 upregulation is critical for the development of vascular malformations and is mediated via the caveolae-dependent mechanism.

The establishment of the novel Pdc10BECKO mice model, presented in this manuscript, definitely advances the field and allows further investigations of the CCM pathogenesis and would be of interest to others in the vascular biology community. That being said, the data presented here is not sufficient to support all conclusions the authors have made.

- We highly appreciate your critical comments and instructive suggestions. We have addressed your concerns as follows.

Major concerns:

1. The authors demonstrate that during CCM lesion formation, pericytes gain elevated expression levels of alpha-SMA and dissociate from the endothelial cells. However, the deficiency of pericytes at the CCM sites has been shown already by others (Schulz, Wieland et al. 2015). Furthermore, in their previous work (Jenny Zhou, Qin et al. 2016), authors already have shown similar pericytic dissociation in CCM lesions in the brain. The only novelty of this work is that there's an increase in alpha-SMA protein expression in the pericytes located in the CCM lesions of the mutant mice. The authors suggest that the increase in the alpha-SMA expression may contribute to pericytes dissociation from the endothelial cells; however, the data provided here is not sufficient to support this suggestion. It is not clear from the main text how the upregulation of alpha-SMA in the pericytes may contribute to the CCM progression. Are there any other studies supporting that idea, published examples in the field? If yes, the authors should discuss this. If not, authors should provide data, which would demonstrate that downregulation of alpha-SMA in pericytes can attenuate lesion progression.

- Thank you very much for your great questions. EndMT is not the focus in our current work but we have provided new observations. As Reviewer #1 pointed out: "The importance of the endothelial-mesenchymal transition (EndMT) in CCM pathogenesis has published evidence for and against. Using stimulated emission depletion (STED) microscopy, they show that alpha-smooth muscle actin expression occurs in vascular malformation associated (and aberrant) pericytes and not endothelial cells. The data looks relatively compelling, at least in this particular gene (Ccm3) mouse model of disease. This suggests that some of the data in support of an

EndMT in CCM pathogenesis may not occur as such. Instead these changes may be due to pericytes that dissociate from ECs during CCM formation”.

As you also pointed out, the novelty of this work is that there's an increase in alpha-SMA protein expression in the pericytes located in the CCM lesions of the mutant mice (Fig.3). Importantly, we showed that α -SMA expression in pericytes was diminished upon Tie2 inhibition (Supplementary Fig.10).

We agree with you that our data is not sufficient to support that the alpha-SMA expression contribute to pericytes dissociation from the endothelial cells. We have revised the text as follows on page 9: “*Taken together, these data indicate that pericytes may gain α -SMA expression during CCM lesion formation*”.

We agree with the editors' assessment that the in vivo experiments to show that downregulation of alpha-SMA in pericytes can attenuate lesion progression would be beyond the scope of our current study, and we would investigate in our future studies as we mention on page 17: “*The contribution of the SMA expression in pericytes and potentially in other cells to CCM pathogenesis is unknown, and further investigation is needed*”.

We appreciate your understanding.

2. The authors demonstrate elevated levels of both total Tie2 and phosphor-Tie2 protein in the brain and retinal endothelial cells in the Pdc10BECKO mice. They suggest that ccm3 deletion enhances the activation of the Tie2 receptor, which in turn, promotes vascular malformation. The quantification analyses provided here are not sufficient to support higher activity state of the Tie2, because high levels of the phospho-Tie2 might be a result of high levels of the total Tie2 expression (see specific comment 10.1).

- Thank you very much for your careful assessment on the p-Tie2 vs total Tie2. We have re-examined the quantifications and the results were consistent with your suggestions that the high levels of the p-Tie2 is a result of high levels of the total Tie2 expression. Upon normalization the ratios of p-Tie2/total Tie2 are similar between WT and Ccm3-deficient brain tissues or isolated ECs (**Fig.5**; Supplementary Fig.6). However, phosphorylation of Tie2 downstream Akt was significantly upregulated in Ccm3-deficient brain tissues and ECs which can be blocked by Tie2 inhibitor Rebastinib (Supplementary Fig.6; Supplementary Fig.11).

3. In the present work, authors demonstrate that ccm3 deletion in the brain and retinal vasculature affects both endothelial cells and pericytes. However, it is not clear from the manuscript how and if the changes in the molecular signature of both pericytes and endothelia are connected? Is there any connection between the pericytic dissociation and the upregulation of caveolae-Tie2 signaling that drives the CCM lesions? Authors could test whether the alpha-SMA signature of the pericytes at the lesion sites is rescued in the rescue experiments (e.g. Cav1 KO, Rebastinib treatment).

- Thank you very much for your concerns. The Angpt-Tie2 is well established signaling that regulates EC-pericyte interactions. Specifically, Angpt2 dissociates pericytes from ECs to destabilize vessels^{5,6}, and we have previously shown that CCM3-deficient ECs increases Angpt2 secretion which in turn enhances dissociations of pericyte whereas anti-Angpt2 blocked this dissociation in an in vitro EC-pericyte co-sprouting assay⁴.

In the present study, we show that Cav1 deletion or Tie2 inhibitor prevented pericyte dissociations from ECs, correlated with CCM lesion formation (**Fig.6 and Fig.7**). Moreover, we show that the alpha-SMA signature of the pericytes at the lesion sites is rescued by the Rebastinib treatment (Supplementary Fig.10).

4. Some data provided in this manuscript demonstrate alterations in the expression of several molecular factors in the endothelial cells (i.e., VE-Cad, Claudin-5) and pericytes (i.e., alpha-SMA, NG2) in the CCM lesions of the Pdc10BECKO mutant animals. However, the authors based

their conclusions mainly on the fluorescent immunostainings. Have the authors performed any other quantification analyses to support what they show in the IF images? WB, RT-PCR analyses, perhaps?

- Thank you for your concerns. We have provided a whole panel of Western blots for signaling proteins, EC and SMC/pericyte markers in WT and *Ccm3*ecKO at different stages (Supplementary Fig.6). Despite that Western blotting may not be cell-specific and temporal (lesion)-specific as the immunostaining, it has provided us important information related to distinct CCM lesion progression between cerebrum and cerebellum.

We found that CCM3 protein was expressed at a lower level in cerebellum than cerebrum (Note: Other brain cells also express CCM3 so overall CCM3 levels were not reduced in *Pdcd10*^{BECKO} mice unless we isolated brain ECs as presented in Supplementary Fig.2). Interestingly, Cav1, Tie2, p-Tie2 and downstream p-Akt were highly upregulated in cerebellum compared to cerebrum whereas other known CCM3-regulated signaling such as phosphor-MLC (indicative of the RhoA signaling) was increased more profoundly in cerebrum than cerebellum. These results further support a critical role of the Cav1-Tie2 signaling in CCM lesion progression.

In parallel, we also examined the vascular structural proteins such as EC junctional proteins (Cldn5, ZO1, VE-cadherin and N-cadherin), and SMC/pericyte marker proteins (PDGFR β and SMA). Of note, the overall levels of pericyte marker PDGFR β and EC junctional proteins were lower in cerebellum compared to cerebrum, reciprocal to the Cav1-Angpt2 levels.

5. The authors show that during CCM lesion formation the endothelial cells become leaky. What might cause the vascular hyper-permeability? Is it a result of disrupted tight junctions? Or due to increased transcytosis (aka increased caveolae vesicles). Or both? Authors could address this question by performing HRP injections into the mutant mice followed by EM ultrastructural analysis.

- Thank you very much for your great questions. We have performed the TMR-dextran perfusion assay, junctional analyses by EM and HRP/DAB-based transcytosis assays. Results show that increased transcytosis via caveolae was enhanced in *Pdcd10*^{BECKO} mice and this augment was diminished by the Rebastinib treatment (**Fig.8**).

Based on our analyses, we think both junctional disruption and enhanced transcytosis contribute to increased hyper-permeability.

6. The authors propose that “caveolae mediates Tie2 intracellular trafficking and recycling, leading to hyper-activation of Tie2”. However, while the levels of Tie2 are increased in the endothelial cells of the mutant mice, the data presented in this manuscript is not sufficient to prove that Tie2 receptor is indeed being recycled to the cell surface. In the figure 5f, authors show that both surface and endocytic Tie2 are elevated in the *ccm3* knockdown cells. An alternative interpretation of the presented data might be is that Tie2 receptor is internalized via the caveolae-mediated endocytosis and translocated into the nucleus for further activation of the downstream signaling, rather than being recycled. Could the authors test whether the Tie2 receptor in the mutant mice colocalizes with the nuclei of the endothelial cells, perhaps by using the super resolution microscopy?

- Thank you for your suggestion. We employed STED to visualize Tie2 localization and we found no/or very minimal nuclear translocation of total Tie2 or p-Tie2 in ECs of WT and *Ccm3*-KO brain tissues (**Fig.4h-i**). Based on this in vivo data (although we cannot exclude the Tie2 nuclear location in cultured ECs - see Fig.5), we didn't test the model that Tie2 is translocated into nucleus followed by further activation of downstream signaling. This could be incorporated into our future studies.

7. It would be interesting to test whether depletion of Cav1 results in more degradation of Tie2 protein, based on the clathrin-mediated Tie2 degradation mentioned in the main text (page 15,

para 1). Authors could test this by examining the localization of Tie2 to late endosomes/lysosomes in the Cav1 depleted cells.

- Thank you for your suggestion. To address your concern, we have performed extensive analyses to determine Tie2 intracellular trafficking, protein half-life and localization with lysosome markers. The results are presented in **Fig.5**, Supplementary Fig.7 with description on pages 10-11. In particular, we have shown that decreased Tie2 localization in the Lamp1⁺ lysosome was detected in Ccm3-deficient ECs. However, Cav1 deletion augmented localization of Tie2 with the lysosomal marker Lamp1 (**Fig.5h**).

Specific comments:

1. P6, para 1: Authors claim that “splenectomy prolonged Pdc10ECKO mouse survival...” is that a new data? If yes, it should be shown here. If it was published, citation needed.

- It is the new data. We have now described the detailed results in Supplementary Fig.1i-j. Specifically, splenectomy at P6 prolonged *Pdcd10*^{ECKO} mouse survival to 1 month, and these mice exhibited seizures with severe CCM lesions and brain hemorrhaging.

2. Could the authors provide a proof for successful Ccm3 depletion by Western or some other means?

- We have verified the Ccm3 deletion by isolating brain ECs followed by qRT-PCR and Western blot analyses. Results showed that Ccm3 was specifically deleted in brain ECs but not in pericytes (Supplementary Fig.2e-g).

3. Fig.1 C and K: The number of lesions is increased over time in both cerebrum and cerebellum, however, according to the presented data, the effect was more substantial in the cortex in the Pdc10BECKO mice. The authors don't relate to this at all in the text. Also, according to the data presented here, in the cerebellum, there are larger lesions compared to the cerebrum at all time points tested. In the cerebrum, the majority of the lesions are smaller, and only at 6 months old mice, the size of most of the lesions is increased. Is there any physiological significance to the lesion size itself? Why the cerebrum vessels exhibit smaller lesion sizes in the earlier time points?

- Thank you for your questions. Please see details below:

Page 7: *All previous CCM mouse models with whole-body EC-specific deletion induce CCM lesions only in the cerebellum as the mice died around P15⁷⁻¹². It is still unknown why CCM lesions progress more rapidly in cerebellum compared to cerebrum. This could be related to the unique structure and hemodynamics of cerebellum^{10,13} or spatial expression of the Angpt-Tie2 signaling as reported previously^{14,15}.*

In our current new model, lesions could be detected in both cerebrum and cerebellum even though lesions in cerebrum grow slower compared to cerebellum. We focus on cortex in our presentation because we used cortex for later two-photon microscopy analyses.

Page 9: *We found that CCM3 protein was expressed at a lower level in cerebellum than cerebrum (Supplementary Fig.6a) (Note: Other brain cells also expressed CCM3 so overall CCM3 levels were not reduced in Pdc10^{BECKO} mice unless we isolated brain ECs as presented in Supplementary Fig.2). Interestingly, Angpt2, Cav1, Tie2, p-Tie2 and downstream phosphor-Akt (p-Akt) were highly upregulated in cerebellum compared to cerebrum whereas other known CCM3-regulated signaling such as phosphor-MLC (indicative of the RhoA signaling) was increased more profoundly in cerebrum than cerebellum (Supplementary Fig.6a). These results further support a critical role of the Cav1-Tie2 signaling in CCM lesion progression.*

4. Supplementary Fig. 3: Is it possible to show that also in the adult stage (P60) the malformations remain specific to the veins by staining against endomucin?

- We have presented the staining for 2-month old lesions with CD31 and endomucin (Supplementary Fig.3b).

5. Page 7, para 2 and Fig. 2: The authors claim there are no lesions when the ccm3 deletion is induced on day P21, however, they don't provide the data.

- We have presented the data in Fig.2a-c.

6. Page 7, Para 3: It should be explained in the main text why mG expressed mainly in the veins and capillaries but not in the arteries, perhaps referring to the Mfsd2a differentiated expression shown in previous studies.

- We have referred to the Mfsd2a differentiated expression shown in previous studies.

Of note, mG expressed mainly in the veins and capillaries but not in the arteries, consistent with the Mfsd2a gene expression pattern¹⁶⁻¹⁸.

7. Page 7, Para 3 last sentence: the authors suggest that "...as well as evident disruption of EC junctions" – however, they don't provide sufficient data, specific junctional staining, but only based on the mG expression.

- We have removed this part from the sentence "...as well as evident disruption of EC junctions". We have provided the junctional immunostaining and EM with quantifications in Fig.3 and Supplementary Fig.5.

8. Fig. 2g-j: In general, it is not clear from the main text or the Methods section how the calculations were made. In particular, the quantification of perfused vs non perfused is not clear. Isn't the perfusion in WT vs the moderate should be mirrored? Why not all vessels in the WT are perfused? It is not clear from the text. Should be elaborated.

- We have added the details for quantifications of perfused and non-perfused vessels. Perfused and non-perfused area was defined by the overlapping and non-overlapping areas of EBD with the vasculature, respectively.

We have marked the typical non-perfused vessels in Fig.2d and all of them are newly formed caverns or sprouts. We detected ~3% non-perfused area/total vascular area in WT, and we think that these are newly formed vessels due to continuous brain vascular remodeling at this stage.

9. Fig. 3:

9.1 3a-c: The authors suggest that in the mutants, the pericytic processes are shortened. This is very clear when comparing the images of P5 del (moderate phenotype) vs WT. However, it's not that clear when comparing the P10 del (mild phenotype) with WT. Is the image of the P10 del is representative? Also, authors do not provide quantification for this statement, but instead refer in the main text to the brackets shown in the figure. However, in the P10 del (mild), shown in the middle column, the bottom panel in Fig. 3a, the brackets connect two different pericytes. Furthermore, according to the image presented in Fig.3a middle column, bottom panel - it seems that the length of the processes in the P10 is not actually shortened. In this image the processes of the detached pericyte looks comparable in the terms of length to the ones seen in the WT.

9.2 Furthermore, it is not clear why authors suggest "...compensatory effect for shortened processes..." (page 8, para 1) when referring to the increased density of pericytes in the mutant animals. They should elaborate on this statement.

9.3 Also, it is not well described neither in the main text or the Methods section, how the calculations made in the graphs 3b,c. Is it the number of caverns and pericytes per endothelial mm³?

9.5 Is the total measured length/area of the endothelia in the WT vs mutants equivalent?

- We address these concerns together since they are related.

We agree with you that pericyte length was not significantly increased in mild lesions and we have revised the text. We have re-labeled the figure too.

We have added the quantification methods as follows (page 21): Quantifications of cavern number, vascular area and pericyte number. High-resolution fluorescence images at 1024 pixels x 1024 pixels from regions of interest (ROIs) which equals 738.10 μm x 738.10 μm were acquired. Cavern number, vascular areas and pericyte number were quantified from four separate ROI images from Z-projections of 60 μm depth from the cerebrum cortical surface for each mouse group captured from the cerebrum cortex. For imaged volume (mm^3), microns were converted to mm which is 0.7381 mm x 0.7381 mm. To calculate volume we multiplied 0.7381 x 0.7381 by the depth (e.g. 60 μm therefore 0.06 mm). i.e. $\text{mm}^3 = 0.7381 \times 0.7381 \times 0.06$. Cavern numbers were calculated for total numbers per mm^3 imaged volume. For the imaged vasculature, we quantified vascular areas instead of vascular volume, and GFP⁺ vascular areas were calculated after merging 60-layer images (1 μm per layer). (Note: It is difficult to calculate vascular volume due to complexed vascular network and various vessel diameters). Subsequently, number of pericytes per 10,000 μm^2 vascular area was quantified.

We have realized that our initial quantification for pericyte density (pericyte number/total area) was not accurate. We have quantified as pericytes/vascular area which is more accurate. Based on the new analyses, PC density in mild lesion was not significantly altered compared to WT. However, PC density was drastically reduced in moderate lesions compared to WT and mild lesions (in revised Fig.2k-m).

As mentioned above, both PC density and PC length in mild lesion were not significantly altered compared to WT. Therefore, there is no compensatory effect. We have revised the text (page 8).

9.4 3d-e: Authors say that the images are taken from animals age of P15, however, they haven't provided the exact experimental procedure paradigm they used in the experiment described in fig.3 – deletion at P1, P5, P10? They perhaps should explain why they chose P15 stage in these experiments.

9.6 3f, g and Supplementary fig.5 (revised Fig.3 c-f): Again, authors haven't provided the exact experimental procedure paradigm they used in the experiments described here – deletion at P5, P10? Is there an explanation why they have decided to image the tissues from pups at developmental stage P15? From the representative images in these figures, it is clear that alpha-SMA signal is increased in the CCMs lesions of the mutant animals. Have the authors performed any quantificational analysis to support the conclusion made from the provided images? Or perhaps performed WB or RT-PCR analyses apart from fluorescent immunostaining?

- We address these two concerns together since they are related to each other.

We have now provided protocol diagrams and description of ages for Ccm3 deletion and analyses in text/legends for each experiment. Specifically, Fig.3c-f were P1 deletion/P15 analyses.

In general, we used P1 deletion/P15 early severe lesion model for most of our studies, and the moderate lesion model (P5 deletion/P60 analyses) for two-photon microscopy imaging.

We have provided WB data for α -SMA (Supplementary Fig.6) and page 10.

9.7 On the one hand, the authors claim that there's less pericytic coverage in the caverns (live imaging), on the other hand, here they focus on the caverns to show that pericytes are there (by showing NG2 expression). Is it an early preliminary (or intermediate) stage prior to the pericytic dissociation? Or perhaps at the P15 stage, the pericytes are still there and are dissociated later in adulthood? The authors should discuss this. Again, the age of the animals in the fig. 3a-c is not provided.

- We have now provided protocol diagrams and description of ages for Ccm3 deletion and analyses in text/legends for each experiment. Specifically, the original Fig.3a-c were a moderate lesion model with P5 deletion and analyses at P60, therefore some pericytes were detected. Yes, we agree with you that at an early or intermediate stages pericytes were still attached to vessels but were dissociated at later lesions.

10. Supplementary Fig. 6 (now Supplementary Fig.5): are all the TJs were weakened in the mutants? If not all, what is the percentage of the weakened TJs in the mutants? Again, what is the age of the animals? Is this following mild or severe ccm3 deletion?

- We apologize for lacking clear description. It was a severe lesion model with P1 deletion and P15 analyses. In this model, it was around 65% tight junctions were weakened, and this was based on quantifications from 20 EM section per mouse and n=5 (Supplementary Fig.5a-b).

11. Fig. 4f-j:

11.1 The authors demonstrate that the levels of the total and phospho-Tie2 receptor are increased in the mutant mice and in the samples of CCM3 lesions from humans. The upregulation of phospho-Tie2 might be due to the total upregulation of the Tie2 expression. Have the authors tested in a quantifiable manner whether the deletion of the ccm3 induced Tie2 receptor phosphorylation? For instance, measurement of the Phospho-Tie2 levels normalized by the total Tie2 protein (WB or IF).

- We have analyzed total and phospho-Tie2 (p-Tie2) by immunostaining and Western blotting. The results are consistent with your suggestion that both total and p-Tie2 are increased to the same extent, and the ratios of p-Tie2/total Tie2 are similar between WT and the mutant mice (Fig.4, Fig.5 and Supplementary Fig.6 and 11).

11.2 Furthermore, in the Supplementary fig. 8a (now Supplementary fig. 7d) now authors demonstrate that the mRNA levels of the Tie2 are not altered following ccm3 knockdown. How do the authors explain the elevated levels of the Tie2 protein (supp. Fig. 8C; now Supplementary fig. 7e) but not its mRNA?

- We have determined the protein half-life and we showed that CCM3 and Cav1 regulate Tie2 at a post-translational level, specifically the Tie2 protein stability (revised Fig.5 and Supplementary Fig.7). We have added the new data on page 11.

12. Fig. 5a: The authors show co-localized expression of Cav1 and Tie2 by IF. Perhaps, a control with siRNA knockdown of Cav1 and Tie2 and IF on these cells would support the specificity of this staining.

- Thank you for your suggestions and we have added new data on Fig.5 and page 10. Cav1 siRNA diminished the surface Tie2 whereas Tie2 siRNA removed both surface and intracellular Tie2 staining (**Fig.5a**).

13. Fig. 6f , h, k – The calculation for the pericytic coverage is not clear. Is the NG+ signal normalized to the EC signal? Have the authors quantified the NG+ along the same endothelial length/area? In the images provided, it looks like there's more CD31+ signal in the Pcdcd10BECKO (due to the large lesion size) compared to the rest groups.

- We have added the quantification methods. Yes, the NG+ signal normalized to the EC signal. Specifically, quantification of coverage of pericytes (NG2), adherens junction (VE-cad) and tight junctions (Cldn5) on CD31 vessels were measured by % of the marker coverage on CD31 vessels, i.e., % green (CD13)-conjugated area/total red (CD31) areas was quantified by Image J.

Minor comments:

1. Fig. 1f-g: Indicate in the figure legend what do the arrows stand for? Consider to include in the figure what for each of the dyes stand, similarly to the Collagen column. It would be easier for the audience that is not familiar with these dyes to follow.

- Arrows indicate positive staining. We have included in the figure what for each of the dyes stand: Prussian blue (Hemosiderin), Sirius red (Collagen) and Ki67 (Proliferation).

2. Fig. 2:

2a: "P1 deltion" -> P1 deletion:

- We have corrected it.

2c: Not indicated in the figure legend what do the arrows stand for. :

- In Fig.2c, arrowheads and arrows indicate normal capillary and cavern, respectively.

In Fig.2d, Arrowheads indicate normal capillaries/postcapillary venules in WT and *Pdcd10*^{BECKO} mice, whereas arrows indicate non-perfused newly formed sprouts in *Pdcd10*^{BECKO} mice.

3. Pages 7-8, referral to the Nissl dye: The reference is not accurate for the labelling of pericytes by Nissl dye. The indicated reference doesn't include such a methodology. Authors should include the right citation: Damisah at al., Nature neurosci., 2017.

- We have corrected the reference.

4. Fig. 3f, g: The changes in the colors of the CD31 and NG2 signals along the figure makes it difficult to follow. For instance, in Fig.3 f CD31 and NG2 are red and then on the right column NG2 appears as grey. In the Fig.3g it's the same for the CD31 and NG2 and then in the last column, CD31 is changed to blue. It would be helpful if each channel was differentially pseudo-colored and kept consistent along the figure.

- Based on your suggestion, we have switched to their original color in revised Fig.3d-f.

5. Fig. 4e-h: Similarly, to my previous comment, it would be helpful if the Cav1 signal would be indicated by the same color along the figure, just as in fig.4i-j.

- We apologize not to switch since we try to be consistent in our manuscript and keep Tie2 and p-Tie2 in green when it was co-stained with other markers (CD31, Cav1 or NG2).

6. Page 9, para 2: "...We obtained paraffin blocks for 8 cases...", perhaps should be blocks of 8 cases.

- We have made the change.

7. Please, keep consistent either p-Tie2 or phospho-Tie2.

- We keep p-Tie2 throughout the text.

8. Page 11, para 3, "...intraperitoneally (i.p.) into *Pdcd10ECKO*..." – Should be *Pdcd10BECKO*.

- We have corrected it.

9. Page 11, bottom: "To this end, we injected...subcutaneously injected..." – injected appear twice.

- We have corrected it.

10. Supplementary Fig. 8b: check the figure legend. Instead of Cav1 should be UNC13B.

- We have corrected it.

11. Supplementary Fig. 12a,b: "P1 deltion" -> P1 deletion

- We have corrected it.

12. Page 14, para 2, the bottom of the page: "...Genetic rescue study with Cav1-deficient..." - Should "deficient".

- We have corrected it.

General minor comments:

1. I found the paper reasonably easy to follow, but I think a non-specialist audience will struggle, as many of the terms are not defined. Some re-writing to address this – and some of the minor grammatical errors, typos and complex sentence construction throughout might be useful.

- We have defined the terms such as VM, GOF, and etc; and correct the grammatical errors.

2. In general, authors should better explain elaborate more how they've performed image quantification analyses. For instance, for the EM analyses how many slices per each mouse? The explanation appears in the fig.4a legend, but not in the Supplementary fig. 6.

- We have added more details on image quantification analyses, including EM and IF analyses, in the legends.

3. "In vivo", "in vitro" in italics please.

- We have made the changes.

Response to Reviewer #3

Zhou et al. describe a novel mouse model to study cerebral cavernous malformations (CCM), where the *Ccm3* (*Pdcd10*) is deleted specifically in the brain and retinal vasculature using *Mfsd2a*-*CreERT2*. The authors observed increased caveolae and increased phospho-Tie2 in the lesions of *Pdcd10*BECKO mice. *Cav1*-deficiency and Tie2 inhibition in *Pdcd10*BECKO mice were able to reduce the CCM lesions. The authors also used two-photon microscopy to visualize the vasculature during CCM lesion progression.

Although a specific deletion of *Pdcd10* in the brain vasculature using the *Slco1c1*(BAC)-*CreERT2* was previously explored by Tang et al. (*Science Translational Medicine*, 2019), the disease phenotype was not characterized as carefully as in this manuscript by Zhou et al. Notably, using two-photon, confocal and stimulated emission depletion (STED) super resolution microscopy, the authors provide high-resolution visualization of the vasculature and EC-pericyte interactions during CCM lesion progression. They show that CCM lesion was generated by initially bulging from capillary/post-capillary venule with a single dilated blinding end (cavern) that rapidly grew into disorganized venule beds with disrupted microvascular network. The authors provide evidence for an intriguing model that increased caveolae vesicles in the brain microvascular ECs, and the increased caveolae augmented Tie2 receptor signaling contributed to CCM lesion progression. In summary, this is a very conclusive study, however, with few issues deserving additional consideration.

- We highly appreciate your positive comments. We also thank you for your instructive suggestions. We have addressed your concerns as follows.

Major comments:

1. In contrast to *Cav1* genetically deficient mice, the role of Tie2 is demonstrated using rebastinib, a small molecular kinase inhibitor of Abl kinases, SRC, KDR, FLT3, and Tie-2. The reported IC50 values suggest that whereas the inhibitor has higher activity towards various Abl forms, it inhibits Tie2 and KDR at similar efficacies. The authors need to comment on the specificity of rebastinib, and the potential effects it may have via inhibition of Abl and Kdr, which contribute to vascular leakage and growth. The authors have also limited the analysis of the Tie2 pathway to analysis of phospho-Tie2, with no analysis of downstream signalling or gene expression changes.

- Thank you for your suggestions. We have addressed your concern and examined the specific target of Rebastinib in CCM3-deficient brain and ECs. Please find the result section (Supplementary Fig.11 and page 14):

For the signaling, we showed that phosphorylation of Tie2 downstream Akt was significantly upregulated in *Ccm3*-deficient brain tissues and ECs which can be blocked by Tie2 inhibitor Rebastinib (Supplementary Fig.11). Interestingly, *Angpt2*, *Cav1*, Tie2, p-Tie2 and downstream phosphor-Akt (p-Akt) were highly upregulated in cerebellum compared to cerebrum whereas other known CCM3-regulated signaling such as phosphor-MLC (indicative of the RhoA signaling) was increased more profoundly in cerebrum than cerebellum (Supplementary Fig.6). These results further support a critical role of the *Cav1*-Tie2 signaling in CCM lesion progression. (also see Response # 3 below).

We showed that both p-Tie2 and p-Abl were increased in *Pdcd10*^{BECKO} brain, and both increases could be blocked by Rebastinib. Interestingly, we showed that Abl siRNA (but not Kdr siRNA), similar to *Cav1* siRNA, attenuated increased total (and p-Tie2) in CCM3-deficient ECs. However, Abl expression was not regulated by CCM3 or *Cav1*. These data suggest that Abl, together with *Cav1*, regulate Tie2 protein expression and activation in EC. Our study provides a novel mechanism for Tie2 regulation. The contribution of Abl to CCM lesion progression need further investigations.

2. Are the CCM lesions leaky in the *Pdcd10*BECKO mice? The imaging of EBD indicated that most caverns were perfused, but some of the newly formed caverns were non-perfused (Fig.2e with quantifications in 2h-i). Additionally, leakage was not observed (Figs 1-2). However, it is reported that *Cdh5* and *Cln5* levels are decreased, and that evident disruption of EC junctions in moderate lesions was observed (Fig.2f with quantification in 2j).

- Based on your suggestion, we have performed TMR-dextran perfusion assay and HRP/DAB-based transcytosis assay. We detected vascular leakage in CCM lesions, consistent with the disruption of EC junctions. Moreover, Tie2 inhibition could attenuate vascular leakage in the mouse models.

We have added new data into a new figure (Fig.8) and new result section “Tie2 inhibition normalizes EC barrier function” on pages 14-15.

Minor comments:

3. Interestingly, *Pdcd10* deletion does not result in lesion formation when induced at p21. What makes the vessels different before and after this time point? Was a similar time dependence observed in the retina? As shown before in the retina, *Angpt2* expression seizes in the superficial vascular plexus after a short postnatal period (Hackett et al., 2000, Elamaa et al 2020). After this occurs, VEGF expression alone is not sufficient to cause angiogenic growth, unless *Angpt2* is ectopically expressed (Oshima, et al. 2004). Could the authors comment the expression of *Angpt2* in the postnatal brain vasculature and on the potential interplay of the *Angpt2* and VEGF pathways in CCM formation.

- We have examined *Angpt2* and Tie2 expression in the postnatal brain by Western blot, and presented the data in Supplementary Fig.6a and pages 9-10.

4. Are the non-perfused areas hypoxic? Regarding the potential role of Tie2, did the authors study if rebastinib increased perfusion in the lesions?

- We have not measured the hypoxia, and we expect the area might be hypoxic in the mutant mice. Specifically, the non-perfused area quantified in WT mice was ~3% relative to vascular area, and was increased 2.7-4-fold (~8-12% in the mutant mice). This needs further investigations.

Rebastinib normalized the vascular structure and function (Fig.7 and Fig.8). Although we did not perform EBD perfusion, we expect that Rebastinib would increase perfusion in the brain vasculature.

5. The authors should consider a recent publication of an *in vitro* vascular malformation model of activated Tie2 (Yuqi Cai et al., Constitutive Active Mutant TIE2 Induces Enlarged Vascular Lumen Formation with Loss of Apico-basal Polarity and Pericyte Recruitment, Scientific Reports, 2019).

- We have added the reference and stated as follows:

Moreover, mutant ECs expressing Tie2-L914F, the most frequent mutation identified in venule malformations patients, form enlarged lumens with the paucity of pericytes, mimicking vascular lesions present in an *in vitro* model¹⁹.

Furthermore, we have discussed the differences between Cav1-Tie2 signaling in CCM3 and the Tie2 GOF-induced venous malformation at both anatomy and the signaling (pages 19-20).

6. The mechanism behind increased Tie2 activation remains open. The authors speculate “Therefore, it is conceivable that combined increases of ANGPT2 and Tie2 lead to highly activated Tie2 that could drive vascular malformation.” Indeed, the activity of *Angpt2* is highly context-dependent. Ectopic expression of *Angpt2*, resulting in high levels of *Angpt2*, did not cause visible changes in the brain vasculature (e.g. Li et al. JCI, 2020), although it induced the enlargement of post-capillary venules and capillaries in the trachea (Korhonen et al., Kim et al., JCI, 2016) in otherwise healthy mice. In the tracheal vascular bed, the activity of *Angpt2* was dependent on

Tie1. In the lymphatic vasculature, Angpt2 agonist activity has been attributed to low level of VE-PTP expression in the lymphatic compartment (Souma et al., PNAS, 2018). Have the authors explored Tie1 or VE-PTP expression in the postnatal brain/retina/CCM lesions?

- Thank you for your suggestions and advice. We have examined both VE-PTP and Tie1 in Ccm3-deficient ECs. Our data suggest that increased Tie1 in CCM3-KO ECs may contribute to Angpt2 autocrine-induced Tie2 activity (see Fig.1 for Reviewer only). However, it needs more studies to define the mechanism by which CCM3 regulates Tie1 (for example, Tie1 mRNA, Tie1 surface expression and protein stability), and how increased Tie1 contributes to Angpt2-Tie2 signaling and CCM lesion formation using in vitro and in vivo CCM models. This is beyond the scope of current study and we would like to incorporate into our future studies. Nevertheless, we have added in our discussion (page 19).

Data for Reviewer only:

In the lymphatic vasculature, Angpt2 agonist activity has been attributed to a low level of VE-PTP expression in the lymphatic compartment²⁰. We did not detect any changes in the mRNA expression of VE-PTP between WT and Ccm3KO ECs, although the protein expression and more importantly the phosphatase activity towards Tie2 need to be determined. Interestingly, it has been reported that endothelial Tie1 is essential for the agonist activity of autocrine Angpt2 by directly interacting with Tie2 to promote Angpt2-induced p-Tie2²¹⁻²⁴, promoting Angpt2-induced the enlargement of post-capillary venules and capillaries in the trachea^{21,22}. These studies prompted us to examine Tie1 expression in CCM lesions and CCM3-KO ECs. We observed highly upregulated Tie1 expression in CCM lesions (**a-d**). Moreover, Tie1 was upregulated in cultured

ECs by CCM3 deletion and co-silencing Tie1 attenuated p-Tie2 in CCM3-KO ECs (**e**). These data suggest that increased Tie1 in CCM3-KO EC may attribute to Angpt2-induced Tie2 activation.

Fig.1 for Reviewer only. CCM lesions exhibit increased Tie1 expression correlated with increased Angpt2-Tie2 signaling. a-d. Angpt2, Tie1 and p-Tie2 were upregulated in mouse CCM lesions. Cerebral sections from P15 old WT and P1 deletion *Pdc10*^{BECKO} mice were immunostained for Angpt2, Tie1 and p-Tie2 with CD31. Representative merged images of normal vessels and CCM lesions (asterisks) are shown. n=6. ****, P<0.0001. **e.** Tie1 was increased in CCM3-KO ECs. Various siRNA-transfected hBMVECs cells were

subjected to Western blotting for Tie1 and p-Tie2. Relative protein levels were quantified and fold changes are presented by taking Ctrl siRNA as 1.0. n=3.

7. Authors show that rebastinib hinders the disease formation when administered after tamoxifen. From a translational point of view, it would be of interest if Rebastinib can normalize the vasculature in already formed lesions.

- Thank you for your great suggestions. We have tested the therapeutic effects of Rebastinib on pre-existing CCMs in a severe model (Ccm3 deletion at P1 followed by administration of the inhibitor at P15 when the CCM lesion fully bloomed). Results showed that Rebastinib could

prevent further CCM lesion progression and even shrink the pre-existing lesions compared with the vehicle-treated group. However, the inhibitor could not completely reverse the pre-existing lesions. More experiments need to test the therapeutic effects in less-progressive models (for example, the moderate and mild CCM lesion models with delayed injection of tamoxifen for Ccm3 deletion described in our manuscript). We present the data as “Data for Reviewer only” and we feel that this should be incorporated into our future studies (Please see Fig.2 for Reviewer only).

Fig.2 for Reviewer only. Tie2 inhibition prevents progression of pre-existing lesions. a. A diagram for the protocol. WT and P1 *Pdcd10*^{BECKO} mice were subcutaneously administered with vehicle or Rebastinib daily at P16-P29 and brain tissues were harvested at P30. P16 KO mice were harvested as lesion baseline. b. Images of fresh brain tissue and H&E staining of brain sections. c. The numbers of total lesions were quantified as # of lesions per 10 coronal sections, which were 200 μ m apart. n = 6

mice per group. *, $P < 0.05$; *** $P < 0.001$ (two-way ANOVA). Scale bars: 1 mm (fresh brain image); 25 μ m (H&E).

8. Did the authors follow whether *Pdcd10*^{BECKO}:*Cav1*^{-/-} (DKO; tamoxifen P1-3) prevented the development of lesions also when analysed at much later time points, e.g. at P60, or was the Caveolin-1 deficiency only delaying the disease progression?

- Yes, we have presented the data for p60 in the original Fig.6i-k. *Cav1* deficiency dramatically reduced lesion sizes and numbers even at 2-month age as visualized by whole brain imaging (Fig.6i). We have now indicated the ages of mice in the figures.

9. In page 6, row 28: authors should remove 3 from sentence “...developed CCM3 lesions with dilated capillaries...”

- We have corrected to CCM lesions.

10. In page 6, row 31: authors talk about body weight loss, however as the deletion is done postnatally before the mice have reached their adult weight, the correct term would be weight gain.

- We have corrected to body weight gain.

11. In page 6, row 34: the sentence “Venous endothelium and capillaries marker endomucin staining shown venous malformations at the periphery of the retinal vascular plexus” is unclear and should be rephrased.

- We have rephrased the sentence as “Staining with venous endothelium and capillary marker endomucin indicated that venous malformations were detected at the periphery of the retinal vascular plexus”.

12. In the figure legend of Supplementary Fig.4. Pcd10ibbbKO should be corrected to Pcd10BECKO.

- We have corrected it. We have also corrected it in the Figure 4c.

13. In Figure 2 in panel a P1 deltion should be corrected to P1 deletion.

- We have corrected to P1 deletion.

14. In Figure 5a authors should provide an image with more cells to make conclusions about the increased colocalization of TIE2 and Cav1.

- We have provided more cells in Fig.5a.

15. In Figure 5b quantification of the western blot should be provided.

- We have provided the quantification for Fig.5b.

16. In figure 5 letters m and n indicating panels, are missing.

- We have added the labeling.

17. In page 12, row 4: Rabastinib should be corrected to Rebastinib.

- We have corrected it.

18. Please correct ref 19. Detter, M.A., Snellings, D.A. & Marchuk, D.A. Cerebral Cavernous Malformations Develop Through Clonal Expansion of Mutant Endothelial Cells. Cir Res In press (2018).

- We have updated the reference ²⁵.

REFERENCES

1. Lant, B., *et al.* CCM-3/STRIPAK promotes seamless tube extension through endocytic recycling. *Nat Commun* **6**, 6449 (2015).
2. Goudreault, M., *et al.* A PP2A phosphatase high density interaction network identifies a novel striatin-interacting phosphatase and kinase complex linked to the cerebral cavernous malformation 3 (CCM3) protein. *Mol Cell Proteomics* **8**, 157-171 (2009).
3. Gaillard, S., Bartoli, M., Castets, F. & Monneron, A. Striatin, a calmodulin-dependent scaffolding protein, directly binds caveolin-1. *FEBS Lett* **508**, 49-52 (2001).
4. Zhou, H.J., *et al.* Augmented endothelial exocytosis of angiopoietin-2 resulting from CCM3-deficiency contributes to the progression of cerebral cavernous malformation. *Nat Med* **22**, 1033-1042 (2016).
5. Eklund, L. & Olsen, B.R. Tie receptors and their angiopoietin ligands are context-dependent regulators of vascular remodeling. *Exp Cell Res* **312**, 630-641 (2006).
6. Maisonpierre, P.C., *et al.* Angiopoietin-2, a natural antagonist for Tie2 that disrupts in vivo angiogenesis. *Science* **277**, 55-60 (1997).
7. McDonald, D.A., *et al.* A novel mouse model of cerebral cavernous malformations based on the two-hit mutation hypothesis recapitulates the human disease. *Hum Mol Genet* **20**, 211-222 (2011).
8. Chan, A.C., *et al.* Mutations in 2 distinct genetic pathways result in cerebral cavernous malformations in mice. *J Clin Invest* **121**, 1871-1881 (2011).
9. Cunningham, K., *et al.* Conditional deletion of Ccm2 causes hemorrhage in the adult brain: a mouse model of human cerebral cavernous malformations. *Hum Mol Genet* **20**, 3198-3206 (2011).
10. Boulday, G., *et al.* Tissue-specific conditional CCM2 knockout mice establish the essential role of endothelial CCM2 in angiogenesis: implications for human cerebral cavernous malformations. *Dis Model Mech* **2**, 168-177 (2009).
11. Maddaluno, L., *et al.* EndMT contributes to the onset and progression of cerebral cavernous malformations. *Nature* **498**, 492-496 (2013).
12. Zhou, Z., *et al.* Cerebral cavernous malformations arise from endothelial gain of MEKK3-KLF2/4 signalling. *Nature* **532**, 122-126 (2016).
13. Plate, K.H. Mechanisms of angiogenesis in the brain. *J Neuropathol Exp Neurol* **58**, 313-320 (1999).
14. Hackett, S.F., *et al.* Angiopoietin 2 expression in the retina: upregulation during physiologic and pathologic neovascularization. *J Cell Physiol* **184**, 275-284 (2000).
15. Elamaa, H., *et al.* Angiopoietin-4-dependent venous maturation and fluid drainage in the peripheral retina. *Elife* **7**(2018).
16. Ben-Zvi, A., *et al.* Mfsd2a is critical for the formation and function of the blood-brain barrier. *Nature* **509**, 507-511 (2014).
17. Andreone, B.J., *et al.* Blood-Brain Barrier Permeability Is Regulated by Lipid Transport-Dependent Suppression of Caveolae-Mediated Transcytosis. *Neuron* **94**, 581-594 e585 (2017).
18. Chow, B.W., *et al.* Caveolae in CNS arterioles mediate neurovascular coupling. *Nature* **579**, 106-110 (2020).
19. Cai, Y., Schrenk, S., Goines, J., Davis, G.E. & Boscolo, E. Constitutive Active Mutant TIE2 Induces Enlarged Vascular Lumen Formation with Loss of Apico-basal Polarity and Pericyte Recruitment. *Sci Rep* **9**, 12352 (2019).
20. Souma, T., *et al.* Context-dependent functions of angiopoietin 2 are determined by the endothelial phosphatase VEPTP. *Proc Natl Acad Sci U S A* **115**, 1298-1303 (2018).
21. Korhonen, E.A., *et al.* Tie1 controls angiopoietin function in vascular remodeling and inflammation. *J Clin Invest* **126**, 3495-3510 (2016).

22. Kim, M., *et al.* Opposing actions of angiopoietin-2 on Tie2 signaling and FOXO1 activation. *J Clin Invest* **126**, 3511-3525 (2016).
23. Leppanen, V.M., Saharinen, P. & Alitalo, K. Structural basis of Tie2 activation and Tie2/Tie1 heterodimerization. *Proc Natl Acad Sci U S A* **114**, 4376-4381 (2017).
24. Savant, S., *et al.* The Orphan Receptor Tie1 Controls Angiogenesis and Vascular Remodeling by Differentially Regulating Tie2 in Tip and Stalk Cells. *Cell Rep* **12**, 1761-1773 (2015).
25. Detter, M.A., Snellings, D.A. & Marchuk, D.A. Cerebral Cavernous Malformations Develop Through Clonal Expansion of Mutant Endothelial Cells. *Cir Res* **123**, 1143-1151 (2018).

REVIEWER COMMENTS

Reviewer #1 (Remarks to the Author):

The authors have been very responsive to my questions and concerns. I have no further comments or concerns.

Reviewer #2 (Remarks to the Author):

The authors addressed all of our concerns. There is a minor typo line 120 should be "cavin proteins" instead of "calvin proteins".

Reviewer #3 (Remarks to the Author):

The authors have responded to all comments. There are few issues that remain to be considered.

Specific Comments:

1. P. 6, row 137. Please correct the phrasing, since Tang et al have reported on a brain EC specific deletion of Pcd10 (STM, 2019), in addition to global-EC deletion.
2. Fig 5a. It remains unclear how the authors determined the intracellular vs cell surface localization of Tie2 in the immunofluorescence stainings of cells. In fact, the method is not described in the manuscript.
3. P. 16, row 493. The authors should also refer to Tang et al (STM 2019) who developed a brain EC specific Pcd10 deletion. See #1.
4. P. 16, row 505. Did the authors investigate rebastinib-induced EC junction normalization in the Pcd10BECKO mice, as stated in the discussion?

IN RESPONSE TO RESPONSE TO REVIEWERS

Response to Reviewer #1

The authors have been very responsive to my questions and concerns. I have no further comments or concerns.

- Thank you very much.

Response to Reviewer #2

The authors addressed all of our concerns. There is a minor typo line 120 should be “cavin proteins” instead of “calvin proteins”.

- Thank you very much. We have corrected it.

Response to Reviewer #3

The authors have responded to all comments. There are few issues that remain to be considered.

- Thank you very much. We have addressed your concerns as follows.

Specific Comments:

1. P. 6, row 137. Please correct the phrasing, since Tang et al have reported on a brain EC specific deletion of *Pdcd10* (STM, 2019), in addition to global-EC deletion.

-Thank you for your suggestion. We have rephrased as “Current CCM models have employed *Ccm* gene deletions with either a global EC- or a brain EC-specific promoter” (Monvoisin et al., 2006; Tang et al., 2019; Wang et al., 2010).

2. Fig 5a. It remains unclear how the authors determined the intracellular vs cell surface localization of Tie2 in the immunofluorescence stainings of cells. In fact, the method is not described in the manuscript.

- We have revised our text as follows: As previously reported (Parton et al., 2020), Cav1 was restricted to the membrane proximal region in control EC where Tie2 was localized with Cav1 (arrow) as determined by immunofluorescence staining. Silencing of CCM3 in HBMVEC significantly increased Cav1 and Tie2 co-localization both in the membrane proximal region (arrow) and in intracellular vesicles (arrowhead).

Importantly, we have determined the surface expression of Tie2 by biotinylation assays (Fig. 5d-e).

3. P. 16, row 493. The authors should also refer to Tang et al (STM 2019) who developed a brain EC specific *Pdcd10* deletion. See #1.

-We have rephrased the sentence: A similar model has been established for a *Ccm2* or *Ccm3* deletion with the *Slcoc1-CreERT2* system (Cardoso et al., 2020; Tang et al., 2019).

4. P. 16, row 505. Did the authors investigate rebastinib-induced EC junction normalization in the *Pdcd10*BECKO mice, as stated in the discussion?

-Yes, we have determined the tight junctions by EM and have now included the data in Fig.8f-g.

References:

- Cardoso, C., M. Arnould, C. De Luca, C. Otten, S. Abdelilah-Seyfried, A. Heredia, A.L. Leutenegger, M. Schwaninger, E. Tournier-Lasserre, and G. Boulday. 2020. Novel Chronic Mouse Model of Cerebral Cavernous Malformations. *Stroke* 51:1272-1278.
- Monvoisin, A., J.A. Alva, J.J. Hofmann, A.C. Zovein, T.F. Lane, and M.L. Iruela-Arispe. 2006. VE-cadherin-CreERT2 transgenic mouse: a model for inducible recombination in the endothelium. *Dev Dyn* 235:3413-3422.
- Parton, R.G., M.A. Del Pozo, S. Vassilopoulos, I.R. Nabi, S. Le Lay, R. Lundmark, A.K. Kenworthy, A. Camus, C.M. Blouin, W.C. Sessa, and C. Lamaze. 2020. Caveolae: The FAQs. *Traffic* 21:181-185.
- Tang, A.T., K.R. Sullivan, C.C. Hong, L.M. Goddard, A. Mahadevan, A. Ren, H. Pardo, A. Peiper, E. Griffin, C. Tanes, L.M. Mattei, J. Yang, L. Li, P. Mericko-Ishizuka, L. Shen, N. Hobson, R. Girard, R. Lightle, T. Moore, R. Shenkar, S.P. Polster, C.J. Roedel, N. Li, Q. Zhu, K.J. Whitehead, X. Zheng, A. Akers, L. Morrison, H. Kim, K. Bittinger, C.J. Lengner, M. Schwaninger, A. Velcich, L. Augenlicht, S. Abdelilah-Seyfried, W. Min, D.A. Marchuk, I.A. Awad, and M.L. Kahn. 2019. Distinct cellular roles for PDCD10 define a gut-brain axis in cerebral cavernous malformation. *Sci Transl Med* 11:
- Wang, Y., M. Nakayama, M.E. Pitulescu, T.S. Schmidt, M.L. Bochenek, A. Sakakibara, S. Adams, A. Davy, U. Deutsch, U. Luthi, A. Barberis, L.E. Benjamin, T. Makinen, C.D. Nobes, and R.H. Adams. 2010. Ephrin-B2 controls VEGF-induced angiogenesis and lymphangiogenesis. *Nature* 465:483-486.

REVIEWERS' COMMENTS

Reviewer #3 (Remarks to the Author):

The authors have responded to all my comments.

Additionally, they have included new data and text. The text on p. 13 should be checked for English language, including the sentence on row 405, which starts in the middle of a sentence.